



# Stratified suppression of turbulence in an ice shelf basal melt parameterisation

Claire K. Yung[1,2,3], Madelaine G. Rosevear[2,4], Adele K. Morrison[1,2], Andrew McC. Hogg[1,3], and Yoshihiro Nakayama[5]

[1]Research School of Earth Sciences, Australian National University, Canberra, Australia
[2]Australian Centre for Excellence in Antarctic Science, University of Tasmania, Hobart, Australia
[3]Australian Centre of Excellence for Climate Extremes, Australian National University, Canberra, Australia
[4]Department of Mechanical Engineering, University of Melbourne, Melbourne, Australia
[5]Institute of Low Temperature Science, Hokkaido University, Hokkaido, Japan

**Correspondence:** Claire K. Yung (claire.yung@anu.edu.au)

**Abstract.**

Ocean-driven basal melting of Antarctic ice shelves is an important process that affects the Antarctic Ice Sheet, global climate and sea level. Basal melting occurs within ice shelf cavities, which are not represented in most global ocean or climate models. Models targeted for studying ice-ocean interactions include ice shelf cavities and are critical tools for understanding basal melt and the ocean circulation beneath ice shelves but rely on parameterisations to predict basal melt. Most currently used basal melt

parameterisations best represent shear-driven melting occurring in a limited parameter space of ice shelf cavity conditions. In other conditions, stratification of buoyant meltwater against the ice interface suppresses melt and diffusive convection plays a role, both processes that are not adequately included in existing melt parameterisations. We implement an improved three-equation melt parameterisation in two ocean models, which accounts for stratification suppressing the turbulence that drives basal melting. This stratification feedback parameterisation is based on the results of LES studies, which suggest a functional

dependence of heat and salt transfer coefficients on the viscous Obukhov scale. Changes in melting and circulation due to the stratification feedback are regime-dependent: melt rates in idealised, quiescent simulations decrease by 80% in warm cavity conditions and 50% in cold conditions. The stratification feedback also suppresses melt rates in a high-resolution regional Pine Island Glacier simulation by 60%, suggesting that much of the ice shelf boundary layer is affected by stratification. However,

unconstrained boundary layer parameters, inter-model differences and unresolved processes continue to present challenges for accurately modelling basal melt in ocean models.

## 1   Introduction

Ice loss from the Antarctic Ice Sheet will have profound effects on global sea level (Fretwell et al., 2013; Seroussi et al., 2020), the global thermohaline circulation (Jacobs, 2004; Li et al., 2023) and therefore global climate. Antarctic ice shelves, the

floating extensions of the Antarctic Ice Sheet, buttress the ice sheet and slow its flow towards the ocean. However, ice shelves melt from underneath where they are in contact with the ocean; this basal melting contributes half of Antarctica's total mass



loss (Rignot et al., 2013), and has been accelerating in recent decades (Rignot and Jacobs, 2002; Rignot et al., 2013, 2014; Pritchard et al., 2012). A lack of observations beneath ice shelves (Malyarenko et al., 2020; Rosevear et al., 2022a) has led to a reliance on ocean models to understand ice-ocean interactions and predict future Antarctic melt (Dinniman et al., 2016).

However, there remain large uncertainties in melt rate projections and feedback mechanisms within the ice-ocean system, associated with poorly understood and insufficiently constrained physical processes (IPCC, 2023; Bennetts et al., 2024).

Antarctic ice shelf melting is controlled by ice shelf–ocean boundary layer processes, which occur on scales that are too small to resolve in large-scale ocean, climate and earth system models (Rosevear et al., 2024). We thus rely on basal melt parameterisations designed to represent the observed melting process (e.g. Hellmer and Olbers, 1989; Holland and Jenkins,

1999). However, existing widely employed parameterisations applied to borehole ocean data overestimate melt compared to co-located radar-based observations (Kimura et al., 2015; Begeman et al., 2018; Middleton et al., 2022; Rosevear et al., 2022a; Schmidt et al., 2023; Davis et al., 2023) in many Antarctic ice shelves. This overestimation can be attributed to an oversimplification of the processes that drive ice shelf cavity melt. It is therefore critical to better represent these processes in basal melt parameterisations for accurate sea level and climate projections (Rosevear et al., 2024). Improvements to basal

melting parameterisations are also motivated by the strongly coupled relationship between melting and buoyancy-generated ice shelf cavity circulation (e.g. MacAyeal, 1984; Jenkins, 1991; Jacobs et al., 1992; Jourdain et al., 2017). Feedbacks between melt and circulation are seen both within ice shelf cavities and on the Antarctic margins (Little et al., 2009; Jacobs et al., 2011; Mathiot et al., 2017; Jourdain et al., 2017; Si et al., 2024).

Multiple physical processes contribute to melting beneath ice shelves. These include the diffusion of heat and salt, turbulence

generated by ocean currents interacting with the ice, and convective flows driven by buoyant meltwater (Malyarenko et al., 2020; Rosevear et al., 2024). Various parameterisations (e.g. McPhee et al., 1987; Hellmer and Olbers, 1989; Holland and Jenkins, 1999; Kerr and McConnochie, 2015; McConnochie and Kerr, 2017; Schulz et al., 2022; Zhao et al., 2024) exist to account for these processes where they cannot be resolved, usually by quantifying an efficiency of heat transport across the ice shelf-ocean boundary layer. In ocean models, this efficiency is often taken to be a constant transfer coefficient multiplied by

the velocity of the far-field flow (e.g. Asay-Davis et al., 2016). This functional form assumes a current-driven shear that creates turbulent mixing and heat transport. This assumption is reasonable in some ice shelf cavity conditions, such as the tidally driven, cold Filchner-Ronne Ice Shelf cavity (Jenkins et al., 2010). In such cold cavities, temperatures are generally less than 0.5 °C warmer than the local freezing point (Jenkins et al., 2010). However, in some ice shelf cavities (such as in the Amundsen Sea, e.g. Jacobs et al., 2012), ocean temperatures can be greater than 2 °C warmer than the local freezing point. In these warmer

conditions, the effect of buoyant meltwater stratifying the ice shelf-ocean boundary layer, suppressing turbulence and therefore creating feedback on heat and salt transport is important and not captured by a constant transfer coefficient (Vreugdenhil and Taylor, 2019; Rosevear et al., 2022b). Nor does the parameterisation account for buoyancy-driven convection that may enhance melt (e.g. McConnochie and Kerr, 2017) or the effect of diffusive convection (e.g. Rosevear et al., 2021). These three additional processes are all relevant beneath Antarctic ice shelves (Rosevear et al., 2022b).

Ocean model simulations can address the challenge of inaccurate basal melting parameterisations using model tuning (Asay-Davis et al., 2016). By tuning the transfer or drag coefficients in the melt parameterisation (e.g. Nakayama et al., 2017, 2018;



Hoffman et al., 2024), integrated melt rates within the range of satellite-derived estimates (e.g. Depoorter et al., 2013; Rignot et al., 2013; Liu et al., 2015; Adusumilli et al., 2020) can be achieved. Other simulations use varying choices of basal melt parameterisations (e.g., the forms of Jenkins, 1991; Holland and Jenkins, 1999; Jenkins et al., 2010, used directly from the references or with some tuning), and note biases in ice shelf cavity-integrated melt rates, which are often attributed to biases in water masses, possibly due to low horizontal resolution and a lack of associated eddy transport onto the continental shelf, biased forcing products, or the absence of tides in the simulation (e.g. Timmermann et al., 2012; Kusahara and Hasumi, 2013; Nakayama et al., 2014; Schodlok et al., 2016; Mathiot et al., 2017; Jourdain et al., 2017; Naughten et al., 2018; Nakayama et al., 2019; Richter et al., 2022; Hyogo et al., 2024), and could also be related to the vertical discretisation of the basal melt parameterisation (Gwyther et al., 2020). Therefore, there exist other biases in current ocean models that may compensate for the biases in the melt parameterisation across ice shelf cavity regimes that are expected from *in situ* ice shelf observations (Rosevear et al., 2022a), which lead to integrated ocean model melt rates within or close to the range of satellite-derived melt estimates. However, inaccuracies in the basal melt parameterisation may contribute to simulated regional biases in melt rate and ocean conditions (Mathiot et al., 2017; Naughten et al., 2018; Richter et al., 2022), noting that spatial variation in melt rate within each ice shelf is significant (Adusumilli et al., 2020; Zinck et al., in review), and an accurate basal melt parameterisation will be particularly important when considering possible future ice shelf regime changes (Hellmer et al., 2012; Naughten et al., 2021; Nakayama et al., 2022; Mathiot and Jourdain, 2023; Haid et al., 2023) and future sea level contributions (Goldberg et al., 2019; Morlighem et al., 2021). These limitations motivate the development of more accurate basal parameterisations that encompass the physical processes occurring across a wider range of ice shelf cavity conditions.

Large eddy simulations, direct numerical simulations and laboratory studies have been used to model and understand small-scale turbulent processes that control basal melting of ice shelves. For instance, idealised simulations and laboratory studies have demonstrated the presence of double-diffusive convection (Rosevear et al., 2021; Middleton et al., 2021), as well as convective plumes (Gayen et al., 2016; Mondal et al., 2019; Zhao et al., 2024; Anselin et al., 2024; Kerr and McConnochie, 2015; McConnochie and Kerr, 2018), the effect of stratification of melting (Vreugdenhil and Taylor, 2019; Rosevear et al., 2022b; Begeman et al., 2022), the geometric feedback of ice ablation (Wilson et al., 2023; Sweetman et al., 2024) and the effect of vertical resolution on boundary layer structure in turbulence-permitting ice-ocean melt simulations (Patmore et al., 2023; Burchard et al., 2022). Many of these idealised studies propose modifications or alternatives to the existing melt parameterisations to account for the physical processes occurring in the more quiescent and warmer ice shelf cavity conditions where current parameterisations do poorly (Rosevear et al., 2022a, b). Some of these parameterisations match well with *in situ* observations, such as the Kerr and McConnochie (2015) convective melt rates at vertical ice faces in Greenland (Schulz et al., 2022; Zhao et al., 2024) and beneath the Ross Ice Shelf (Malyarenko et al., 2020). The latter is notable since the Kerr and McConnochie (2015) laboratory study uses vertical ice faces whereas Antarctic ice shelves are generally weakly sloped ($< 1°$) from the horizontal. However, thus far, these idealised parameterisation modifications have not been implemented nor tested in realistic, large-scale models. In this work, we aim to bridge this gap between the insights created by idealised process studies, and the large-scale ocean models used in climate and sea level projections.





We focus on incorporating the effect of stratification due to meltwater on ice shelf–ocean boundary layer turbulence in basal melt parameterisations. The importance of stratification near the ice–ocean boundary has been known for decades: McPhee (1981) proposed an analytic theory derived from Monin-Obukhov boundary layer theory (Monin and Obukhov, 1954) to explain how stabilising surface buoyancy fluxes, such as the melting of sea ice, impact the structure of the water column. McPhee (1981) defined a stability parameter, $\eta_*$ that scales as a function of the mixed layer depth, velocity and eddy diffusivity. Holland and Jenkins (1999) formalised this stability parameter in the three-equation melt parameterisation, to account for the feedback of stratification suppressing turbulence and therefore melt. However, in ocean models, this stability parameter is often ignored (and set to 1 for simplicity, representing neutral conditions, e.g. Losch, 2008; Dansereau et al., 2014). Furthermore, the stability parameter relies on the assumption of the Monin-Obukhov similarity scaling, which has been shown to break down in strongly stratified conditions (Vreugdenhil and Taylor, 2019). Recent Large Eddy Simulation studies have enabled insights into an improved functional form for the stratification feedback on basal melt (Vreugdenhil and Taylor, 2019; Rosevear et al., 2022b). Transfer coefficients, representing the efficiency of heat and salt transport by turbulence across the ice-ocean boundary layer, decrease as the ice shelf cavity conditions become warmer and more quiescent. The commonly used three-equation parameterisation (Jenkins et al., 2010) can therefore be modified to empirically account for the unresolved feedback between stratification and basal melting in large-scale ocean models.

In this study, we present a modified basal melt parameterisation which we then implement into two ocean models, MOM6 and MITgcm. The parameterisation incorporates the feedback effect of stratification on shear-driven melting based on Large Eddy Simulation experiments. We use the ocean models in idealised ice shelf cavity configurations, spanning a spread of ice shelf cavity regimes, to determine how the stratification feedback affects melt rates and ice shelf cavity ocean circulation compared to the existing constant transfer coefficient parameterisation. We also employ a high-resolution MITgcm simulation of Pine Island Glacier to assess the parameterisation in a realistic configuration. Section 2 describes the parameterisation and its implementation. Section 3 describes the ocean models and the idealised and realistic model configurations. We present the ocean model results in Section 4, before summarising the results and discussing the ongoing challenges in parameterising and predicting basal melt in Section 5 and providing concluding remarks in Section 6.

## 2 Melt Parameterisation Design and Validation

### 2.1 The Three-Equation Melt Parameterisation and Transfer Coefficients

Ice shelf cavity-scale ocean models cannot resolve the turbulent fluxes within the ice shelf-ocean boundary layer. Models generally employ the three-equation basal melt parameterisation (Hellmer and Olbers, 1989; Holland and Jenkins, 1999). This parameterisation consists of three equations solved at the ice shelf-ocean interface. The linear freezing point equation of state,

$$T_b = aS_b + b + cp_b \,, \tag{1}$$

describes the variation of the temperature $T_b$ at the ice-ocean interface with pressure $p_b$ and salinity $S_b$, where subscripts $b$ indicate the ice-ocean boundary layer and the values of constants $a$, $b$ and $c$ are presented in Table 1. The heat conservation



equation,

$$\rho_I L_f m = \rho_I c_{p,I} \kappa_I^T \left.\frac{\partial T_I}{\partial z}\right|_b - \rho_M c_{p,M} \gamma_T (T_b - T_M)\,, \tag{2}$$

describes the balance of heat transport between the ocean mixed layer (sometimes referred to as the far-field, denoted $M$), ice-ocean boundary ($b$) and ice ($I$), and the latent heat required by melting, with $m$ the melt rate. Parameters and constants are presented in Table 1. The key unknown here is the transfer velocity for heat, $\gamma_T$, describing the efficiency of heat transport within the boundary layer. The salt conservation equation,

$$\rho_I m S_b = -\rho_M \gamma_S (S_b - S_M) \tag{3}$$

is similar to the heat equation, where $\gamma_S$ is the transfer velocity for salt. We assume there is no salt flux within the ice and that the salinity of the ice is zero. These three equations (1-3) are solved to obtain the three unknowns; the salinity $S_b$ and temperature $T_b$ at the ice-ocean interface, and the melt rate $m$.

Within the three-equation parameterisation (Eqns. 1-3), different parameter choices can be made. Firstly, the transfer velocities $\gamma_T$ and $\gamma_S$ are important controls of the melt rate. Typically, these transfer velocities are assumed to be proportional to 135 the friction velocity $u_*$, which is a measure of the shear stress on the boundary. In ocean models, $u_*$ is usually taken to be linearly proportional to a far-field velocity as $u_* = C_d^{1/2} U_M$ with $C_d$ the drag coefficient. Proportionality constants $\Gamma_T$ and $\Gamma_S$ are called transfer coefficients, defined by

$$\gamma_T = \Gamma_T u_*\,, \qquad \gamma_S = \Gamma_S u_*. \tag{4}$$

The values of these transfer coefficients are not well known: they can be tuned to observed estimates, as Jenkins et al. (2010) 140 (hereafter J10) did at the Filchner-Ronne ice shelf using co-located borehole ocean measurements and radar-derived melt rates, or tuned in an ocean model to give a desired melt rate (Asay-Davis et al., 2016; Nakayama et al., 2018; Hyogo et al., 2024). Alternatively, transfer coefficients could vary according to theoretical scaling (Kader and Yaglom, 1972; McPhee et al., 1987; Jenkins, 1991) and may also include a Monin-Obukhov scaling in the case of stabilising buoyancy forcing (McPhee, 1981; McPhee et al., 1987; Holland and Jenkins, 1999) (hereafter the HJ99-M81 formulation, Appendix A1). Malyarenko et al. 145 (2020) reviews ocean-driven ice ablation and the development of these parameterisations.

However, the J10 and HJ99-M81 parameterisations overestimate melt in many Antarctic ice shelves, particularly warmer and quiescent ice shelves (Rosevear et al., 2022a). Here, co-located borehole and radar-derived melt rates suggest different, smaller transfer coefficient values than J10. Rosevear et al. (2022b) explain how the J10 and HJ99-M81 parameterisations only do well in specific ice shelf regimes that align with the well-mixed, shear-driven flow. At warmer and more quiescent condi-150 tions, stratification and diffusive-convection physics become more relevant. Even though HJ99-M81 is designed to account for stabilisation due to stratification, its effect on melting in the parameterisation is modest (Appendix A1, Fig. A1).

The drag coefficient is also a large factor in the uncertainty of basal melt predictions (e.g. Dansereau et al., 2014; Walker et al., 2013; Gwyther et al., 2015; Jourdain et al., 2017; Zhao et al., 2024). Suggested values range from 0.0015 (Holland and Jenkins, 1999) to 0.0097 (Jenkins et al., 2010), with a value of 0.0022 estimated from turbulence measurements beneath





the smooth underside of the Filchner-Ronne Ice Shelf (Davis and Nicholls, 2019). However, the drag coefficient beneath ice is expected to vary spatially: sea ice studies suggest dependence on ice roughness (Robinson et al., 2017) and stratification (Kawaguchi et al., 2024), and the boundary layer flow profile has also been shown to affect the drag coefficient at vertical glacial ice faces (Zhao et al., 2024). In ocean models, the drag coefficient has often been used in conjunction with the transfer coefficients as tuning factors to obtain desired melt rates (via the product $\Gamma_T \sqrt{C_d}$, the thermal Stanton number, e.g. Jourdain

et al., 2017), though modifying the drag coefficient in an ocean model may also affect the simulated upper layer velocity.

## 2.2    Stratification Feedback on Turbulence – Insights from Large Eddy Simulations

Stratification due to buoyant meltwater has two distinct effects on the melt rate. One is the effect of meltwater to cool the uppermost part of the boundary layer, which decreases the thermal driving, the difference between $T_M$ and the local freezing point, relative to the far-field thermal driving that parameterisations generally consider (Rosevear et al., 2022b). The other

is the ability of stratification to suppress boundary layer turbulence, which we focus on. Vreugdenhil and Taylor (2019) and Rosevear et al. (2022b) use Large Eddy Simulations (LES) to diagnose regimes of Antarctic ice shelf melt based on the viscous Obukhov scale $L^+$, a non-dimensional variable defined as the ratio of the Obukhov length $L$ and a viscous length scale $\delta_\nu$:

$$L^+ = \frac{L}{\delta_\nu} = \frac{\frac{-u_*^3}{k B_b}}{\nu / u_*} = \frac{-u_*^4}{\nu k B_b} \, , \tag{5}$$

where $\nu$ is the molecular viscosity, $k$ the von Kármán constant (Table 1), and $B_b$ the surface buoyancy flux. Assuming transfer

velocities given by the three-equation melt parameterisation (Eqns. 1-4), the surface buoyancy flux can be written as

$$B_b = g \left( \beta (S_b - S_M) u_* \Gamma_S - \alpha (T_b - T_M) u_* \Gamma_T \right) \, , \tag{6}$$

with $\alpha$ and $\beta$ the linear thermal expansion and haline contraction coefficients and $g$ the gravitational acceleration (Table 1). A small $L^+$ means the flow is affected everywhere by either stratification or viscosity, which both suppress turbulence. Alternatively, $L^+$ can be thought of as measuring the relative importance of shear currents (represented by $u_*$) to buoyancy and

stratification on the flow.

     $L^+$ can be used to distinguish regimes of ice shelf melting. Rosevear et al. (2022a) and Vreugdenhil and Taylor (2019) use LES simulations beneath horizontal ice and find that at large $L^+ \geq \mathcal{O}(10^4)$, corresponding to cold temperatures (small $T_M - T_b$) and fast flows (large $u_*$), the ice shelf cavity is in a well-mixed regime. In this regime, melting is controlled by velocity shear and the transfer coefficients are similar to J10 and HJ99-M81, therefore existing parameterisations perform well

(Rosevear et al., 2022a) (Fig. 1a,b). At smaller viscous Obukhov scales, $\mathcal{O}(10^4) > L^+ > \mathcal{O}(10^3)$, corresponding to warmer and more quiescent flows, the ice shelf cavity enters a stratified regime where buoyant meltwater acts to suppress turbulence but melting is still shear-driven, thus effectively decreasing the transfer coefficients. Finally, at low $L^+ < \mathcal{O}(10^3)$ (we use $L^+ < 2500$ as the cut-off, following Rosevear et al., 2022b), corresponding to the warmest and most quiescent flows, the ice shelf cavity enters the diffusive-convective regime where the difference between the salt and heat diffusivities results in

diffusive convection and melt rates are transient and dependent on a diffusive-convective timescale (Rosevear et al., 2022b; Middleton et al., 2021). Due to its transient nature, this regime is inherently difficult to parameterise and is not the focus of our



**Table 1.** Table of constants, variables and parameters in the basal melt parameterisations

| Symbol | Description | Value |
| --- | --- | --- |
| a | Liquidus gradient for salinity | $-0.0573\ ^{\circ}\mathrm{C\ kg\,g^{-1}}$ |
| b | Liquidus constant offset | $0.0826\ ^{\circ}\mathrm{C}$ |
| c | Liquidus gradient for pressure | $-7.53 \times 10^{-1}\ ^{\circ}\mathrm{C\ dbar^{-1}}$ |
| $T_b, S_b, p_b, \rho_b$ | Ice-ocean boundary layer temperature, salinity, pressure, density | |
| $T_M, S_M, \rho_M$ | Far-field temperature, salinity and density | |
| $\rho_I$ | Ice density | $918\ \mathrm{kg\ m^{-3}}$ |
| $c_{p,I}, \kappa_I^T$ | Heat capacity and conduction parameters in ice, not used | |
| $c_{p,M}$ | Heat capacity in seawater | $3974\ \mathrm{J\ K^{-1}\ kg^{-1}}$ |
| $L_f$ | Latent heat of fusion | $3.34 \times 10^5\ \mathrm{J\ kg^{-1}}$ |
| $\rho_0$ | Eqn. of state reference density | $1027.51\ \mathrm{kg\ m^{-3}}$ |
| $\alpha$ | Eqn. of state thermal expansion coefficient | $3.733 \times 10^{-5}\,^{\circ}\mathrm{C}{-}1$ |
| $\beta$ | Eqn. of state haline contraction coefficient | $7.843 \times 10^{-4}\ \mathrm{psu^{-1}}$ |
| $g$ | Gravitational acceleration | $9.80\ \mathrm{m\,s^{-2}}$ |
| $\gamma_T, \gamma_S$ | Transfer velocities for heat and salt | |
| $L^+$ | viscous Obukhov scale | |
| $\nu$ | molecular viscosity | $1.95 \times 10^{-6}\ \mathrm{m^2 s^{-1}}$ |
| $k$ | von Kármán constant | 0.40 |
| $B_b$ | surface buoyancy flux | |
| $\Gamma_T, \Gamma_S$ | Transfer coefficients for heat and salt | |
| $A_T$ | StratFeedback heat transfer coefficient constant of proportionality | -7.39 |
| $n_T$ | StratFeedback heat transfer coefficient $L^+$ scaling factor | 0.322 |
| $A_S$ | StratFeedback salt transfer coefficient constant of proportionality | -9.90 |
| $n_S$ | StratFeedback salt transfer coefficient $L^+$ scaling factor | 0.223 |
| $\Gamma_{T,CC}$ | ConstCoeff parameters and upper-limit of StratFeedback parameterisation | 0.012 |
| $\Gamma_{S,CC}$ | ConstCoeff parameters and upper-limit of StratFeedback parameterisation | $3.9 \times 10^{-4}$ |
| $u_*$ | Friction velocity | |
| $C_d$ | Drag coefficient | 0.0025 |
| $U$ | Far-field velocity | |
| $U_t$ | Prescribed tidal velocity | $0.01\ \mathrm{m\,s^{-1}}$ |
| $T^*$ | Thermal driving | |

work. Note that these ice shelf cavity regime definitions, defined by $L^+$ values, differ slightly from Rosevear et al. (2022b) in that we only describe the stratified regime as the ice shelf cavity conditions with suppressed turbulence due to stratification. In





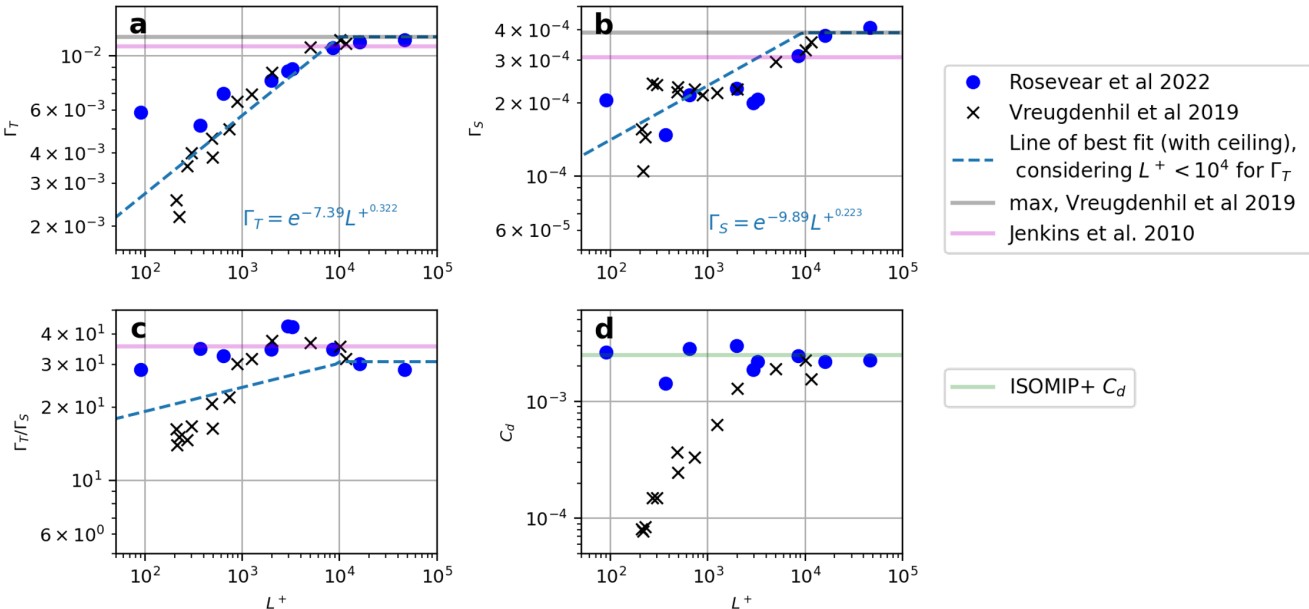

**Figure 1.** Large Eddy Simulation data, with Vreugdenhil and Taylor (2019) in the black crosses and Rosevear et al. (2022b) in blue dots, indicating the relationship between transfer coefficients (a) $\Gamma_T$, (b) $\Gamma_S$, their ratio (c) $\Gamma_T/\Gamma_S$ and (d) drag coefficient $C_d$ against viscous Obhukov scale $L^+$. The maximum Vreugdenhil and Taylor (2019) (ConstCoeff) values of the transfer coefficients are included (grey lines), which are similar to the Jenkins et al. (2010) values (pink lines). The blue dashed line indicates the choice of fit of transfer coefficients as a function of viscous Obukhov scale for our stratification feedback parameterisation.

contrast, the Rosevear et al. (2022b) stratified regime definition includes the effect of stratification to cool the boundary layer
as mentioned earlier, which is not captured by $L^+$.

## 2.3   Stratification Feedback Parameterisation Design

The stratification feedback (StratFeedback) basal melt parameterisation explored in this study is based on the results of Rosevear et al. (2022b) and Vreugdenhil and Taylor (2019), to incorporate the unresolved suppression of turbulence by buoyant meltwater. Both studies suggest an increase of heat and salt transfer coefficients (calculated from heat and salt gradients) with
the viscous Obukhov scale, up to a constant value (Fig. 1a,b). Assuming a power-law relationship, we calculate a line of best fit through the log-log representation of the $\Gamma$–$L^+$ data where $L^+ < 1 \times 10^4$, but enforce a maximum of $\Gamma_{T,S}$ to be the maximal limits from Vreugdenhil and Taylor (2019) (which is slightly greater than J10). The Vreugdenhil and Taylor (2019) maxima are also our 'control' parameterisation with constant transfer coefficients (ConstCoeff or CC in equations). We force the line to reach these ConstCoeff parameters at $L^+ = 1 \times 10^4$ so that the transition point from the well-mixed to stratified regimes is
uniform between temperature and salinity, and therefore that the ratio $\Gamma_T/\Gamma_S$ is monotonic. This fit is chosen for simplicity since regime transitions of heat and salt transport may occur at different $L^+$, and the exact transitions are uncertain (Fig. 1).



The resultant StratFeedback parameterisation is

$$\Gamma_T = \min \left\{ \exp\left(A_T\right)\left(L^+\right)^{n_T}, \Gamma_{T,\mathrm{CC}} \right\} , \tag{7}$$

$$\Gamma_S = \min \left\{ \exp\left(A_S\right)\left(L^+\right)^{n_S}, \Gamma_{S,\mathrm{CC}} \right\} , \tag{8}$$

where the values of the constants are presented in Table 1. When the melt rate is negative, that is, the ice-ocean boundary layer is freezing, the viscous Obukhov scale becomes negative so our scaling does not make sense, therefore we use the ConstCoeff

(CC) transfer coefficients. Note we also neglect the conductive heat flux term of Eqn. 2. Since the transfer coefficients depend on $L^+$, which in turn depends on melt rate via surface buoyancy forcing, iteration is required for convergence of the three-equation parameterisation solution. We note that other functional forms of a variable transfer coefficient would fit the data of Fig. 1a,b (e.g.Rosevear et al. (2022b) consider a logarithmic fit). To briefly explore the sensitivity to our choice, we also tested steeper and shallower gradient power laws (Appendix A2).

We could also consider an alternative parameterisation where the drag coefficient, as well as the transfer coefficients, is varied. Monin-Obukhov theory expects that under a stabilising buoyancy flux, the drag coefficient is also reduced as the turbulent boundary layer velocity is suppressed relative to the far-field velocity strength. Indeed, Vreugdenhil and Taylor (2019) see this reduction drag coefficient in LES experiments with smaller $L^+$. However, Rosevear et al. (2022b) see only a small change in drag coefficient, and it is not captured by $L^+$ (Fig. 1d). The difference in the behaviour of the drag coefficients between

the LES studies, which otherwise agree strongly, is likely due to the different methods of forcing the current beneath the ice. In the approach of Vreugdenhil and Taylor (2019), strong near-ice stratification leads to the entire model domain becoming laminar. This, in turn, leads to acceleration of the far-field flow and very low drag coefficients. Conversely, the approach taken by Rosevear et al. (2022b) involves a much deeper model domain and constant far-field flow, and this cycle of laminarisation and acceleration does not occur. We find the approach of Rosevear et al. (2022b) to be more realistic and therefore choose to

follow their results in this study. A further alternative formulation of a stratification feedback parameterisation would be to vary the thermal Stanton number ($\Gamma_T C_d^{1/2}$), or use the mixed layer stability parameter $\mu$ (Rosevear et al., 2022b). However, these alternative formulations are beyond the scope of this study.

To illustrate the effect of the StratFeedback parameterisation, we solve the three-equation parameterisation (Eqns. 1-3) with several transfer coefficient choices across different ice shelf cavity regimes. The thermal driving,

$$T^* = T_M - T_{fr}(S_M) , \tag{9}$$

which quantifies the maximum heat available for melting (where $T_{fr}(S) = aS + b + cp$ is the local freezing point as in Eqn. 1), and friction velocity, $u_*$, are used to compute melt rates with the StratFeedback, ConstCoeff, J10 and HJ99-M81 transfer coefficients, assuming $S_M = 34.5$ g/kg and a pressure of 500dbar ($\sim 500$m depth). Fig. 2a demonstrates that the ConstCoeff, J10 and HJ99-M81 transfer coefficients have similar melt rate contours in the thermal driving–friction velocity parameter

space; the ratio of HJ99 and ConstCoeff is relatively uniform except at very low velocities where the McPhee (1981) stability parameter becomes relevant (Fig. A1, and recall that many ocean models set $\eta_* = 1$ for simplicity and therefore do not account for this stratification parameter). However, the magnitude of the ratio is closer to 1 than for the StratFeedback parameterisation,





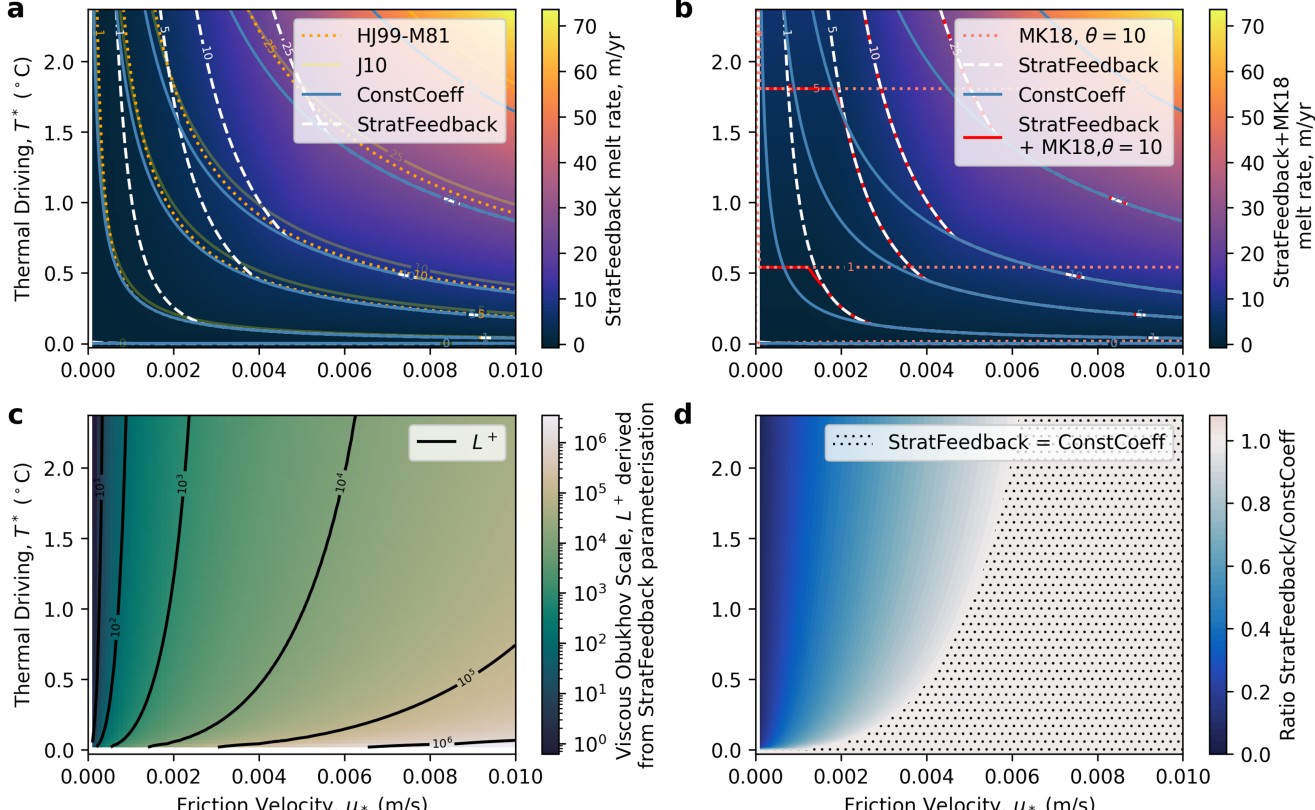

**Figure 2.** Thermal driving – friction velocity phase diagram indicating melt rates calculated as a function of far-field temperature, salinity and pressure (which are set to $S = 34.5$ psu and $p = 500$ dbar) and friction velocity. The melt rates are solved for a variety of parameterisation options in (a) and (b). The orange, dotted lines in panel (a) are the Holland and Jenkins (1999) formulation with the $\eta_*$ stratification parameter (McPhee, 1981). A constant transfer coefficient formulation (ConstCoeff) is in the blue solid lines (using the maximal values of Vreugdenhil and Taylor (2019), which is similar to Jenkins et al. (2010) in the light yellow solid lines), and the stratification feedback (StratFeedback) parameterisation we develop here is shown in white solid lines. In panel (b), we add in the constant melt rates obtained from McConnochie and Kerr (2018) with a slope angle of $\theta = 10°$ from the horizontal in the pink dotted line, and the combination of the StratFeedback+MK18 limit in the red dash-dot line. Panel (c) shows the viscous Obukhov scale $L^+$ derived from the stratification feedback parameterisation, and where $L^+ = 1 \times 10^4$ indicates where the white dashed lines (StratFeedback) and blue line (ConstCoeff) transition from having the same melt rate to the right and different to the left. Panel (d) also shows this in the ratio of the StratFeedback to ConstCoeff melt rates, with stippling indicating where they are equal.

indicating that the McPhee (1981) $\eta_*$ term does not capture the full extent of the stratification feedback on melt seen in the LES simulations. StratFeedback limits to ConstCoeff at high friction velocities and lower thermal driving ($L^+ > 10^4$, Fig. 2c), but changes gradient and has relatively less melting in warmer and more quiescent conditions (the diffusive-convective and stratified regimes), also indicated by the melt rate ratio (Fig. 2d). At a thermal driving of $T^* = 2°C$ and $u_* = 0.001$ m s$^{-1}$,





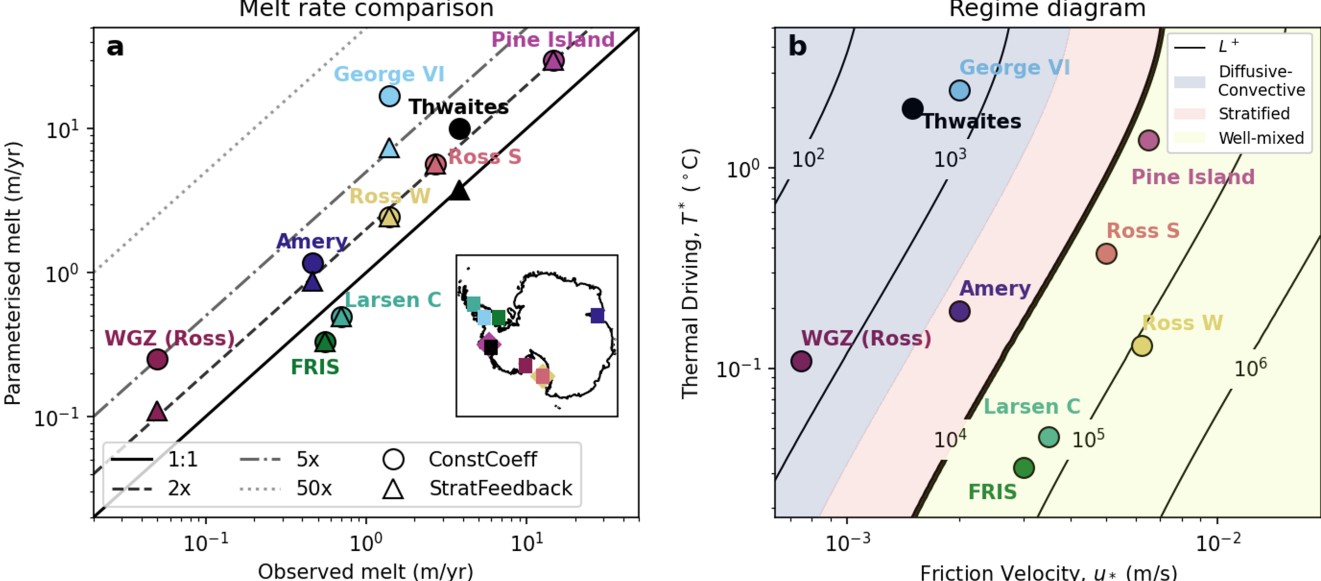

**Figure 3.** Parameterised melt against observed melt rate (a), for borehole observational data updated from Rosevear et al. (2022a), with the ConstCoeff parameterisation in circles and StratFeedback in triangles. Thermal driving – friction velocity regime (b) updated from Rosevear et al. (2022b), where the thick $L^+$ line of $1 \times 10^4$ divides where the StratFeedback parameterisation diverges (to the left) and where the transfer coefficients are constant and equal to ConstCoeff (to the right). The diffusive-convective ($L^+ < 2500$), stratified ($2500 < L^+ < 10^4$) and well-mixed, shear-driven ($L^+ > 10^4$) regimes are shaded. Data is obtained from the Filchner-Ronne Ice Shelf (FRIS, Jenkins et al., 2010), Larsen C Ice Shelf (Davis and Nicholls, 2019), Amery Ice Shelf (Rosevear et al., 2022a), Ross Ice Shelf (Ross S for summer and Ross W for winter data, Stewart, 2018), WISSARD Grounding Zone of the Ross Ice Shelf (WGZ (Ross), Begeman et al., 2018), George VI ice shelf (Kimura et al., 2015; Middleton et al., 2022), Pine Island Glacier (Stanton et al., 2013) and Thwaites Glacier (Davis et al., 2023). Further computation details are supplied in Table B1.

StratFeedback predicts 30% of the ConstCoeff melt, indicating that the StratFeedback parameterisation significantly modifies melt rates in some ice shelf cavity regimes. Fig. 2b uses alternative parameterisation choices in this low-velocity regime that are explained in Section 2.5.

## 2.4 Comparison to Observations

Following Rosevear et al. (2022a), we compare the melt rate produced by the StratFeedback and ConstCoeff melt parameterisations with observed melt rates at limited direct observations of boreholes in Antarctic ice shelves (Fig. 3; data presented in Appendix B). If the parameterisations accurately predicted melt from temperature, salinity, pressure and velocity observations, we would expect the points to lie on the solid 1:1 line of Fig. 3a. However, we find that in general, the ConstCoeff parameterisation overestimates the melt, except for at the Larsen C Ice Shelf and Filchner-Ronne Ice Shelf (FRIS). Note that the studies that originally presented this data may have used slightly different melt parameterisations in their comparisons (e.g. Jenkins





et al., 2010; Davis and Nicholls, 2019, where different drag coefficients and transfer coefficients were used), and recall we ignore heat conduction into the ice, but these choices make no qualitative difference to the overestimation of melt rates.

For five of the ice shelf borehole locations, the melt rate does not change between the ConstCoeff and StratFeedback param-
eterisations (Fig. 3a, co-located circles and triangles). This is because the viscous Obukhov scale $L^+$ is greater than $1 \times 10^4$, indicating these ice shelves are in the well-mixed melt regime (Fig. 3b). However, several of these high $L^+$ locations still have overestimated parameterised melt by a factor of $\sim 2$, for both the ConstCoeff and StratFeedback parameterisations. A likely explanation is a second impact of stratification on melt rate: even in high $L^+$ conditions, the development of a cold meltwater layer can decrease the thermal driving relative to that expected by the far-field temperature, which our StratFeedback parame-
terisation does not address. However, parameterised melt rates decrease when using the StratFeedback rather than ConstCoeff parameterisation for the borehole observations at the Amery, George VI, Thwaites and grounding zone of the Ross Ice Shelf (WGZ) (Fig. 3a, triangles lower than circles, and therefore parameterised melt is closer to matching observed melt rates at the 1:1 line). Of these locations, George VI, Thwaites and WGZ have small $L^+$ (Fig. 3b) suggesting they lie in the diffusive-convective melt regime and that the StratFeedback parameterisation is still missing physics for these ice shelves. We discuss a
possible approach for bridging the transition between stratified and diffusive-convective regimes in section 2.5. Nevertheless, the improvement in overestimation of melt rates for the ice shelves in the stratified regime, such as the Amery ice shelf (as well as the benefit seen in the diffusive-convective regime) motivates us to implement and test the StratFeedback parameterisation in ocean models. Given the lack of *in situ* observational data, there may be many Antarctic ice shelves and sub-sections of ice shelves in this stratified regime.

## 2.5 Limiting to a Velocity-Independent Parameterisation

There is both a numerical and physical reason for the low-velocity ice shelf cavity regime to be specially treated with the three-equation parameterisation (where this regime is characterised by low velocities, but has considerable overlap with the $L^+ < 2500$ diffusive-convective regime). Numerically, a friction velocity of zero (perhaps created by initialising the model at rest) will result in identically zero melt according to Eqns. (2-4), which may lead to numerical problems while solving
for the melt rate. Physically, in the diffusive-convective regime with $L^+ < \mathcal{O}(10^3)$, the StratFeedback parameterisation is an extrapolation. When the friction velocity is $5 \times 10^{-4}$ m s$^{-1}$, Fig. 2d shows that the StratFeedback parameterisation can have ten times less melt than the ConstCoeff formulation. Indeed, at very low velocities the melt rate with the stratification feedback could become arbitrarily small, when in reality we always expect some melt in the presence of a thermal or haline driving even without currents due to the effect of diffusive-convection beneath horizontal ice (Rosevear et al., 2021; Middleton et al., 2021)
or buoyant convection beneath sloping ice (McConnochie and Kerr, 2018; Mondal et al., 2019).

To address this limit, we also implement a transition between the shear-driven parameterisation to a velocity-independent parameterisation based on laboratory studies of sloped ice (McConnochie and Kerr, 2018, hereafter MK18) and similar direct numerical simulations (Mondal et al., 2019). Similarly to the method of Schulz et al. (2022) and Zhao et al. (2024) for vertical ice melt parameterisations, we transition to alternative transfer velocities for heat and salt at low velocities. The effective
transfer velocities are determined by the slope of the ice base, $\theta$ from the horizontal, and other thermodynamic variables



(derivation in Appendix A3):

$$\gamma_{T,\text{eff}} = \chi \sin^{2/3}\theta \left( \frac{g(\beta(S_M - S_b) - \alpha(T_M - T_b))}{\nu} \right)^{1/3} \kappa_s^{1/6} \kappa_T^{1/2} \,, \qquad (10)$$

$$\gamma_{S,\text{eff}} = \gamma_{T,\text{eff}} \frac{\rho_I S_b}{\rho_0 S_M} \sqrt{\frac{\kappa_S}{\kappa_T}} \,, \qquad (11)$$

with $\chi$, an experimentally derived non-dimensional constant, and other parameters defined in Appendix A3 and Table A1.

We choose to transition between regimes by computing the maximum of the velocity-dependent (Eqn. 4) and MK18 velocity-independent transfer velocities (Fig. 2b; red dash-dot line connects the white dashed line, the StratFeedback shear-driven parameterisation, and the orange dotted line, the MK18 limit at a given ice base angle of $10°$). That is,

$$\gamma_T = \max\left( \Gamma_T u_*, \gamma_{T,\text{eff}} \right) \,, \qquad (12)$$

$$\gamma_S = \max\left( \Gamma_S u_*, \gamma_{S,\text{eff}} \right) \,. \qquad (13)$$

Therefore at low velocities, the melt rate is independent of velocity. This formulation differs from McConnochie and Kerr (2017), who propose a transition between shear-driven and convective melt regimes at a critical velocity, noting that the transition conditions are still poorly constrained (Rosevear et al., 2024). The dependence on ice base slope means that beneath horizontal ice shelves ($\theta = 0°$) the MK18 melt rate will still be zero. Furthermore, MK18 and Mondal et al. (2019) do not recommend using the parameterisation on gently sloped ice with angle less than $2°$, therefore the MK18 limit applied to gently 295 sloped Antarctic ice shelves is still an extrapolation into a poorly explored ice shelf regime.

We also explore other alternatives for a velocity-independent parameterisation: a minimum friction velocity and a prescribed tidal velocity, created by altering the definition of the friction velocity. The minimum friction velocity method is expressed as:

$$u_* = \max\left\{ C_d^{1/2} U, u_{*,\text{min}} \right\} \,, \qquad (14)$$

where the minimum velocity is supposed to represent heat transport occurring through diffusion even at very low current 300 speeds (Gwyther et al., 2016). Alternatively, one could consider unresolved, high-frequency tidal velocities increasing the mean friction velocity. The ISOMIP+ (Asay-Davis et al., 2016) protocol calculates friction velocity by adding a tidal velocity $u_t$ in quadrature with the far-field or mixed layer velocity $U$, scaled by the square root of the drag coefficient:

$$u_* = C_d^{1/2} \sqrt{U^2 + u_t^2} \,, \qquad (15)$$

Consequently, the formulation also enforces a minimum friction velocity.

The choices of transfer coefficients and low-velocity limits used in the three-equation basal melt parameterisations that are discussed in this study are summarised in Table 2.

## 3  Model Configurations

To test the performance of the stratification feedback parameterisation in a three-dimensional ice shelf cavity scale model, we implement the parameterisation in two widely used ocean models, MOM6 and MITgcm. We use the Second Ice Shelf-Ocean





**Table 2.** Summary of three-equation basal melt parameterisation transfer coefficients $\Gamma_T$, $\Gamma_S$ used in this study. The five transfer coefficient parameterisations assume a friction velocity calculated from the drag coefficient $C_d$. When implemented in the idealised models, we also explore alternative low-velocity limit choices combined with the ConstCoeff and StratFeedback transfer coefficients which modify the parameterisation at low friction velocities.

| Name | Description | Parameterisation |
|------|-------------|------------------|
| J10 | Jenkins et al. (2010) observation-derived | $\Gamma_T = 0.011$; $\Gamma_S = 3.1 \times 10^{-4}$; $C_d = 0.0097$ |
| HJ99-M81 | Holland and Jenkins (1999)+McPhee (1981) | Variable $\Gamma_T$, $\Gamma_S = f(u_*, \mathrm{buoyancy})$ |
| | with stability parameter $\eta_*$ | (Appendix A1), $C_d = 0.0015$ |
| HJ99-neutral | Holland and Jenkins (1999), $\eta_* = 1$ | Variable $\Gamma_T$, $\Gamma_S = f(u_*)$ |
| | | (Appendix A1), $C_d = 0.0015$ |
| ConstCoeff | Maximum of Vreugdenhil and Taylor (2019) | $\Gamma_T = 0.012$; $\Gamma_S = 3.9 \times 10^{-4}$; $C_d = 0.0025$ |
| StratFeedback | Empirical fit of Vreugdenhil and Taylor (2019) | Variable $\Gamma_T$, $\Gamma_S$ (Eqns. 7,8); maxima match ConstCoeff; |
| | & Rosevear et al. (2022b) | $C_d = 0.0025$ |
| *Low-velocity limit* | | |
| Prescribed $U_t$ | Friction velocity prescribed tidal velocity, | $U_t = 0.01 \mathrm{m\,s^{-1}}$ added to $u_*$, Eqn. 15, |
| | Asay-Davis et al. (2016) | effective min. $u_* = 5 \times 10^{-4} \mathrm{m\,s^{-1}}$ |
| Minimum $u_*$ | Friction velocity minimum | Minimum $u_* = 1 \times 10^{-4} \mathrm{m\,s^{-1}}$, Eqn. 14 |
| MK18 | Convective, velocity-independent parameterisation, | $\gamma_T = \max(\Gamma_T u_*, \gamma_{T,\mathrm{eff}})$ and similar for $\gamma_S$ |
| | McConnochie and Kerr (2018) & Mondal et al. (2019) | Eqn. 10/Sec. A3, [a] additional min $u_* = 1 \times 10^{-4} \mathrm{m\,s^{-1}}$ |

[a] Required for numerical stability to avoid zero transfer velocities in the case of horizontal slopes ($0°$) and no flow, with little effect on the results as the MK18 effective transfer velocities are generally larger than $\Gamma_{T,S} u_{*,\mathrm{min}}$, except when $\theta < 0.05°$ and $u_* < 1 \times 10^4 \mathrm{m\,s^{-1}}$.

Model Intercomparison Project (ISOMIP+) configuration (Asay-Davis et al., 2016) to assess the effect of the parameterisation in an idealised configuration. Then, to explore different regimes of Antarctic ice shelf cavities, the MOM6 ISOMIP+ experiments are modified to include idealised barotropic tides of varying amplitude. Finally, the parameterisation is tested in a high-resolution simulation of Pine Island Glacier (Nakayama et al., 2021). In this section, we briefly describe the ISOMIP+ experiment (and refer the reader to Asay-Davis et al., 2016, for further details) followed by each of the ocean models used in 315 this study.

### 3.1 ISOMIP+ Setup and Modifications

We use the idealised ISOMIP+ Ocean0 model configuration (Asay-Davis et al., 2016) in both MOM6 and MITgcm, which are also submissions to the ISOMIP+ intercomparison project (Yung et al., in prep). The Ocean0 ice shelf draft and bathymetry represent an idealised ice shelf cavity with walls at either side and a grounding line. To assess different regimes of ice shelf 320 cavities, we use both the warm and cold ISOMIP+ temperature and salinity distributions as initial conditions and restoring boundary forcing, all linear as a function of depth. In this way, our warm test cases are effectively the Ocean0 experiment of





ISOMIP+, and our cold test case is a static, cold version of Ocean1, also used in Gwyther et al. (2020). The warm configuration has a temperature of 1°C and salinity of 34.55 g/kg at the bottom of the cavity, aiming to simulate the presence of warm water intrusions, varying linearly to -1.9°C and 33.8 g/kg at the surface. The cold configuration has a spatially uniform temperature
of -1.9°C and a salinity range of 33.8 to 34.7 g/kg. The salinity, temperature and layer interfaces are restored at the northern boundary using a sponge with a restoring timescale of 0.1 days. We use 36 vertical layers, though note the difference in vertical coordinates between MOM6 and MITgcm described below. Unless specified, we follow the mixing, viscosity and equation of state protocols of ISOMIP+ (Asay-Davis et al., 2016).

To simulate basal melt, we use the three-equation parameterisation (Eqns. 1-3) without the ice heat conduction term. We
perform experiments with the StratFeedback and ConstCoeff transfer coefficients and each of the three low-velocity limit choices (Section 2; summarised in Table 2). In all ISOMIP+ simulations, the drag coefficient $C_d = 0.0025$ is used for the melt parameterisation and top and bottom boundary conditions for momentum.

All idealised experiments are run for 730 days. By this time, models are spun up to an equilibrium state.

### 3.1.1 Idealised MOM6 Configuration

The Modular Ocean Model 6 (MOM6; Adcroft et al., 2019) is a finite volume, hydrostatic ocean model which has been used for idealised simulations of ice shelf cavities (Stern et al., 2019). MOM6 is configured on an Arakawa C grid with a generalised vertical coordinate, though here we employ the isopycnal layered version of the model rather than its Arbitrary Lagrangian Eulerian vertical coordinate capabilities (Griffies et al., 2020). We use a bulk mixed layer parameterisation for the surface boundary layer (Hallberg, 2003) and the Jackson et al. (2008) vertical mixing parameterisation with critical Richardson
number 0.25.

The MOM6 ice shelf thermodynamics code numerically solves the three-equation parameterisation using an iterative loop, and both a constant transfer coefficient (e.g. Jenkins et al., 2010) as well as the variable formulation (Holland and Jenkins, 1999; McPhee et al., 1987) can be used. The new stratification feedback parameterisation is implemented with an additional iterative loop to solve for the melt rate, buoyancy forcing and viscous Obukhov scale as described in Section 2. The model
samples temperature, salinity and velocity over the bulk mixed layer in the melt parameterisation, then inserts freshwater in the bulk mixed layer as a volume flux (which can later be entrained in the interior ocean layers, Hallberg, 2003). The magnitude of melting is likely to be sensitive to these choices, as well as vertical resolution (Gwyther et al., 2020). Melting is set to zero when the ocean column is less than 10 m thick. The friction velocity $u_*$ is calculated from the velocities in the uppermost model layer.

### 350 3.1.2 Idealised MITgcm Configuration

The Massachusetts Institute of Technology general circulation model (MITgcm; Marshall et al., 1997) is a finite-volume ocean model that can simulate ice shelf cavities (Losch, 2008). MITgcm uses z (depth) coordinates and is built on an Arakawa C grid. Partial cells are included, with a minimum thickness of 25% of the normal cell thickness of 20 m. Melt rate is calculated using a quadratic equation (Losch, 2008), therefore we implement an additional iterative loop that solves the three-equation





system with the modified and varying transfer coefficients until the solution converges. Tracers and the velocities for the friction velocity and melt parameterisation are sampled over a 20 m layer (Losch, 2008). Meltwater is represented as a virtual salt flux rather than a volume flux, distributed over the same 20 m layer. Unstable vertical mixing is parameterised with a convection scheme (Cessi and Young, 1996).

### 3.1.3 Idealised Explicit Tidal Forcing

In addition to the ISOMIP+ experiments, we run an additional MOM6 case where we add an idealised barotropic tide as an open boundary condition to inject more kinetic energy into the otherwise relatively quiescent cavity. This method differs from the prescribed tidal forcing in the melt parameterisation, where the effect of tides is artificially added to the friction velocity (Section 2.5). By forcing tides explicitly, we capture the direct effect of tides on melting and also the indirect effects due to tidal advection, mixing and residual circulation within the cavity. Here, the sponge boundary is replaced by a Flather-Orlanski
(Flather, 1976; Orlanski, 1976) open boundary, nudged to the values of the sponge configuration output, with an additional sinusoidal tidal velocity and sea surface height forcing at the M2 frequency of 2 cycles per 24 hours and 50 minutes. The amplitude of the tidal velocity and sea surface height are calculated by considering the volume change within the cavity as a result of the tides (with an assumption of linearity, as MOM6 permits grounding line movement), with velocity amplitudes of $0.2 \, \mathrm{m\,s^{-1}}$, $0.1 \, \mathrm{m\,s^{-1}}$, $0.05 \, \mathrm{m\,s^{-1}}$ and $0.01 \, \mathrm{m\,s^{-1}}$ matching sea surface height amplitudes at the boundary of 6.4 m, 3.2 m, 1.6 m
and 0.32 m respectively. However, the resulting tidal velocity at the ice-ocean interface is significantly less (Fig. S1), as seen by Gwyther et al. (2016) and Jourdain et al. (2019). Note we use the minimum friction velocity limit of $u_* = 1 \times 10^{-4} \, \mathrm{m\,s^{-1}}$ discussed in Section 2.5 for numerical stability.

### 3.2 Pine Island Glacier Configuration

For our realistic test, we use the Nakayama et al. (2021) MITgcm Pine Island Glacier configuration. This model configuration
uses MITgcm with the hydrostatic approximation, and has a high spatial resolution of 200 m in the horizontal and 10 m in the vertical, and has been evaluated against satellite observations (Shean et al., 2019; Adusumilli et al., 2020) to have a realistic representation of melt (Nakayama et al., 2019). Although the model can include subglacial discharge, we use the model without this additional flux, noting that the changes in melt rate seen by adding realistic subglacial discharge are modest compared to that expected by adding the StratFeedback parameterisation (Nakayama et al., 2021). Dynamic and thermodynamic sea ice are
included (Losch et al., 2010). The density equation of state is from Jackett and Mcdougall (1995) and the same linear freezing point equation of state as ISOMIP+ is used.

Nakayama et al. (2021) use the Holland and Jenkins (1999) velocity-dependent parameterisation with transfer coefficients dependent on the Prandtl and Schmidt numbers but where the McPhee (1981) $\eta_*$ stability parameter is set to 1, and a drag coefficient $C_d = 0.0015$ (i.e. the MITgcm default values and parameterisation; Losch (2008); named HJ99-neutral in Table 2).
The ocean bathymetry and static ice draft are based on BedMachine-Antarctica (Morlighem et al., 2020). Tides are not included. The model is forced at the boundaries by the Nakayama et al. (2019) model output, which is in turn forced by the ECCO LLC270 optimisation (Zhang et al., 2018) of the ECCO reanalysis (Dee et al., 2011).





We run the Pine Island simulation for 50 days, starting from January 30 2010 conditions, for each basal melt parameterisa-
tion: the original Holland and Jenkins (1999) formulation, the constant coefficient parameterisation, the stratification feedback
parameterisation, and the stratification feedback parameterisation with a tuned drag coefficient.

## 4 Results

### 4.1 Idealised ISOMIP+ Results

We can compare the effect of the stratification feedback (StratFeedback) parameterisation against the more commonly used
constant transfer coefficient (ConstCoeff) method using the ISOMIP+ Ocean0 warm and cold experiments in both MOM6
and MITgcm. The StratFeedback parameterisation greatly affects the melt rates, hydrography and circulation within the ide-
alised ice shelf cavity. Additionally, there are differences between the two models. Fig. 4a-h demonstrates that the temperature
stratification is markedly different in the warm experiments compared to the cold experiments, but all simulations have colder
temperatures near the ice-ocean boundary layer due to the cold meltwater. The size of the meltwater plume varies between
models, appearing thicker in MITgcm than in MOM6. Comparing the melt parameterisations, the constant coefficient experi-
ments (columns 1 and 3) have colder meltwater layers than those with the StratFeedback parameterisation (columns 2 and 4).
Figures 4i-p explain why: melt rates are significantly greater in the experiments with constant transfer coefficients compared
to the same experiments with the StratFeedback parameterisation (note the different colourmap axes between the warm and
cold simulations). Indeed, the warm experiments with the StratFeedback parameterisation (Fig. 4j,l) have melt rates more sim-
ilar to the cold experiments (Fig. 4m-p) than their warm, constant coefficient counterparts (Fig. 4i,k). Comparing MOM6 and
MITgcm, we see that MITgcm has larger melt rates, particularly in the warm experiments (Fig. 5). Stronger melting may be
associated with the $z$-level coordinates in MITgcm versus the higher resolution layer coordinate below the ice shelf in MOM6,
and the different choices of thermal driving sampling depth (Gwyther et al., 2020).

The strong reduction in melt when the StratFeedback parameterisation is included corresponds to the design of the parame-
terisation, which suppresses the transfer coefficients in response to a low viscous Obukhov scale, $L^+$ (Section 2). $L^+$ is smaller
in the warm experiments due to greater thermal driving, therefore the transfer coefficients are more reduced in the warm ex-
periments than the cold. This explains why the decrease in melt rate from the constant transfer coefficient experiments to the
StratFeedback parameterisation experiments is far greater for the warm experiments than the cold, where $L^+$ is larger. Still,
$L^+$ is not large enough in the cold experiments for the cavity to be entirely in the shear-driven regime where $L^+ > 1 \times 10^4$;
otherwise, the parameterisations would have identical melt rates.
The melt rate reduction due to the stratification feedback leads to a change in ice shelf cavity circulation. Fig. 6 shows
the overturning streamfunction (calculated in density space, where streamlines indicate the overturning circulation) and the
depth-averaged total kinetic energy over the whole domain. In all experiments, there is an overturning circulation with buoyant
water travelling up the ice-ocean boundary. In some cold cavity cases, there is an opposing overturning cell at lighter densities,
created by the conservation of volume as the lower overturning water reaches its neutral density and flows toward the boundary
at $x$= 800 km. The kinetic energy is strongest near the positive $y$ boundary of the ice shelf cavity, created as the buoyant plume





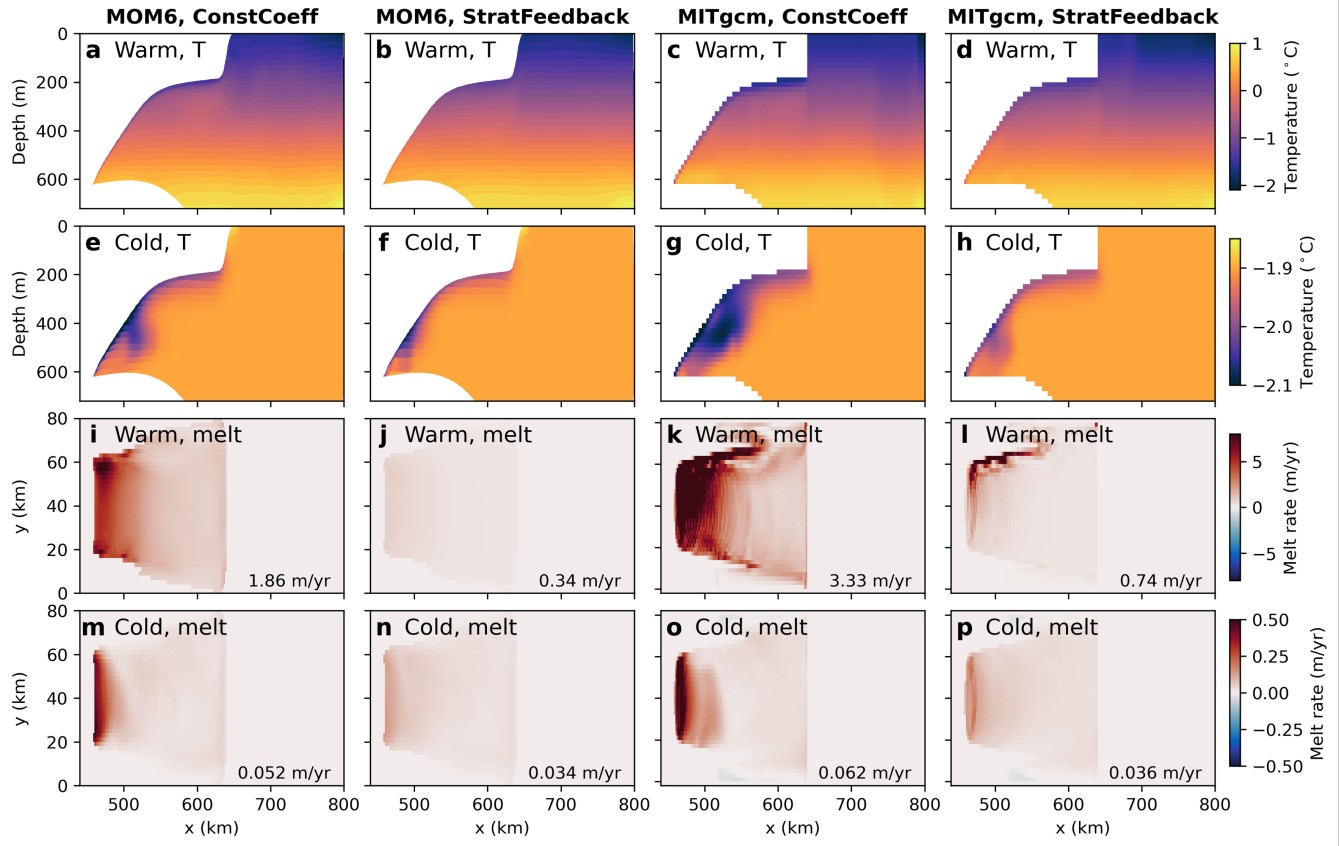

**Figure 4.** Temperature stratification and melt rates for MOM6 (left two columns) and MITgcm (right two columns). All experiments use the ISOMIP+ protocol-specified tidal velocity $U_t = 0.01\,\mathrm{m\,s^{-1}}$ in the melt rate calculation. Variables are averaged over the last 180 days of the simulation, with the temperature profile taken at the y=40 km transect. Columns 1 and 3 show the constant coefficient melt parameterisation results, and columns 2 and 4 contain the stratification feedback parameterisation. Rows 1 and 3 are for the warm experiment, and rows 2 and 4 for the cold case. Melt rates averaged over the ice shelf are listed. Note the different colourbar ranges between the warm and cold simulations.

is pushed to its left (i.e. positive $y$) by the Coriolis force. There is a clear relationship between the magnitude of the melt rates in Fig. 4 and the magnitude of the overturning circulation and kinetic energy, which are both weaker in the cold experiments than the warm (note the different colourbar scales) and weaker in the StratFeedback parameterisation experiments compared to the constant coefficient experiments. This coupled feedback between ice shelf basal melt and cavity circulation is expected because of the buoyancy-driven flow (e.g. Holland et al., 2008; Jourdain et al., 2017).


Additionally, the overturning circulation and kinetic energy are greater in MITgcm than in MOM6 for most experiments (Fig. 6. In the warm experiments, this may be explained by greater melt rates (Fig. 5) and therefore a stronger buoyant meltwater





**Figure 5.** Area-averaged melt rates in the final 180 days of the ISOMIP+ simulations for MOM6 (a, c) and MITgcm (b, d). Warm experiments are in the top row, and cold experiments in the bottom. Hatched bars are experiments with a constant transfer coefficient and solid colours are with the stratification feedback parameterisation, and percentages indicate the ratios of the StratFeedback to ConstCoeff melt rates. Each of the three columns within panels show the results for different choices of lower velocity limit, either a minimum friction velocity of $u_{*,min} = 10^{-4}$ m s$^{-1}$, a prescribed tidal velocity of $u_t = 0.01$ m s$^{-1}$ (which implies a minimum friction velocity of $u_{*,min} = 5 \times 10^{-4}$ m s$^{-1}$) or transitioning smoothly to the McConnochie and Kerr (2018) parameterisation with the local basal slope angle $\theta$ (which for ISOMIP+ ranges between $0°$ and $2°$).

plume but also occurs in the cold experiments where the melting was of similar magnitude. Model choices thus affect both the magnitude of melt and the resultant ice shelf cavity circulation (Yung et al., in prep).

**4.2 Sensitivity to the Low-Velocity Limit**

Thus far, we have investigated the hydrography, melt rate and circulation for the ISOMIP+ warm and cold experiments, using a prescribed, additional tidal velocity in the calculation of friction velocity for the melt parameterisation. Melt rates strongly



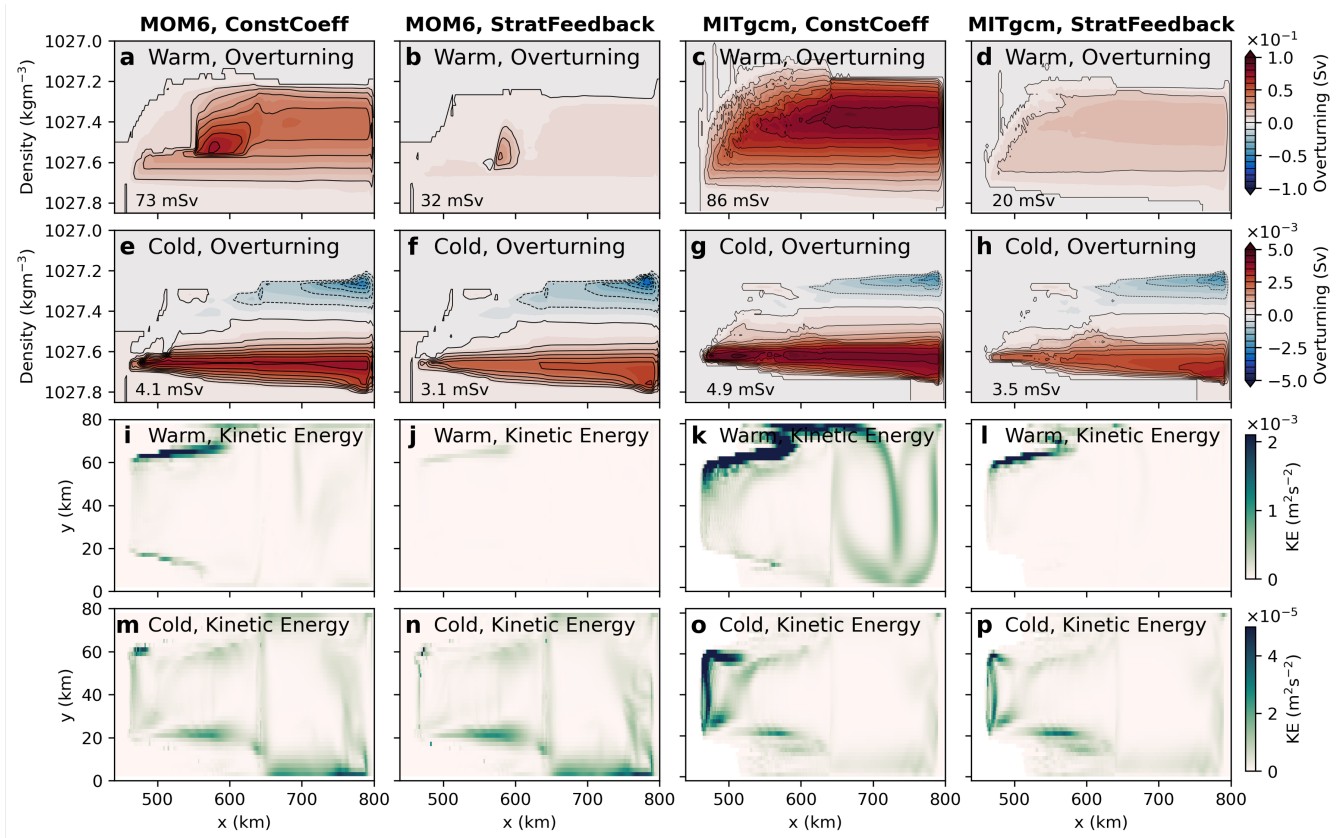

**Figure 6.** Zonally-averaged overturning streamfunction (a-h) in density coordinates, and the depth-averaged kinetic energy (KE) $\frac{1}{2}|\mathbf{u}|^2$ (i-p). All experiments use the ISOMIP+ protocol-specified tidal velocity $U_t = 0.01\,\mathrm{m\,s}^{-1}$ in the melt rate calculation. Data is averaged over the last 180 days of the simulation, binned online in MOM6 and binned using daily averaged output for MITgcm. Columns 1 and 3 show the constant coefficient melt parameterisation results, and columns 2 and 4 contain the stratification feedback parameterisation. Rows 1 and 3 are for the warm experiment, and rows 2 and 4 for the cold case. Black contours are spaced by $10\,\mathrm{mSv}$ in panels (a-d) and $0.5\,\mathrm{mSv}$ in panels (e-h), and the text lists the maximum value of the overturning streamfunction in the domain.

decrease with the incorporation of the StratFeedback parameterisation. However, thermal driving – friction velocity regime diagrams for these experiments indicate that this choice of prescribed tidal velocity directly affects the results (Fig. 7). Almost

all gridboxes in the ice shelf cavity for the prescribed tidal velocity experiments (purple colours) have their friction velocity approximately equal to $C_d^{1/2} u_t = 5 \times 10^{-4}\,\mathrm{m\,s}^{-1}$; that is, the minimum value of Eqn. 15, indicating that model velocities are too weak to contribute significantly to the melting. Low velocities in the idealised configuration can be explained by the smooth topography and ice draft and the lack of a boundary forcing that produces momentum: circulation in the cavity is driven entirely by meltwater buoyancy (apart from restoring at the open boundary).

We explore alternative low-velocity limits in the melt parameterisation (Fig. 5, compare the three columns in each panel), noting that the optimal choice for these is unknown. Different low-velocity limit choices lead to different melt rates, indicating



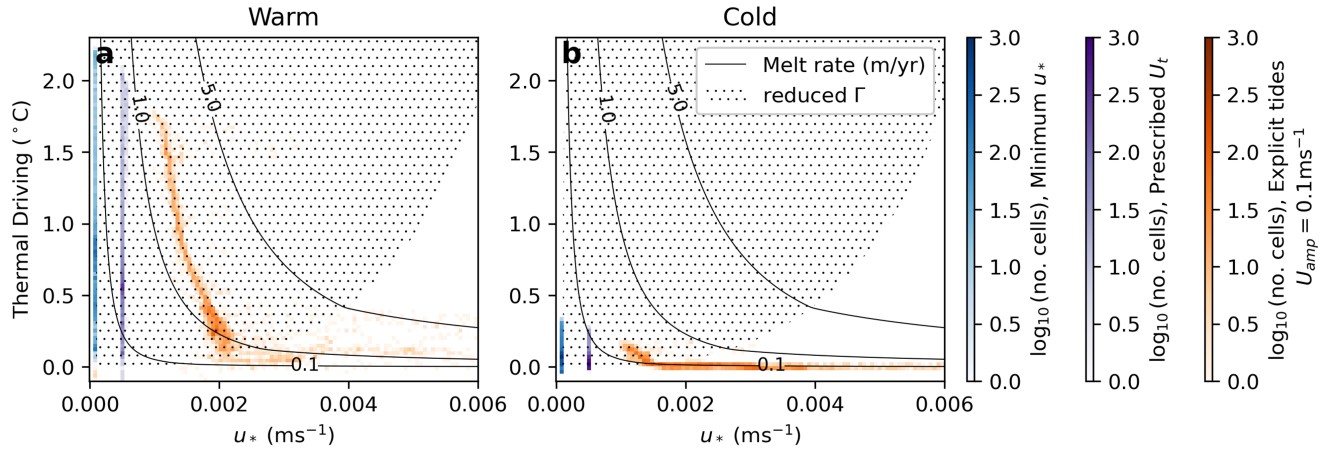

**Figure 7.** Thermal driving – friction velocity regime diagrams for selected MOM6 StratFeedback experiments, indicating the number of grid cells in each regime time-averaged over the final 180 days of the simulation. Panel (a) shows warm experiments and (b) cold. The minimum friction velocity $1 \times 10^{-4}\,\mathrm{m\,s^{-1}}$ experiments are shown in blue (leftmost vertical line), prescribed tidal velocity $U_t = 0.01\,\mathrm{m\,s^{-1}}$ in purple (middle vertical line) and explicit tidal forcing with amplitude $0.1\,\mathrm{m\,s^{-1}}$ in orange colours to the right. StratFeedback melt rates are shown in the solid contours and stippling shows where transfer coefficients are reduced from the ConstCoeff values, both calculated assuming a salinity $S_M = 34.05\,\mathrm{g/kg}$ and pressure $300\,\mathrm{dbar}$, which are representative values for the ISOMIP+ cavity.

that some or all of the ISOMIP+ ice shelf boundary is in this low-velocity regime (Fig. 7). Removing the prescribed tidal velocity and replacing it with a smaller minimum friction velocity (blue colours in Fig. 7) causes melt rates to be nearly zero in both warm and cold experiments with the stratification feedback, where small friction velocities at initialisation (due to the zero flow initial conditions) lead to positive feedback between weak melting and weak cavity circulation (leftmost columns in Fig. 5). Another approach, transitioning to the McConnochie and Kerr (2018) melt rates (section 2.5), resulted in similar melt rates to the minimum friction velocity experiment with the ConstCoeff parameterisation (compare first and third hatched columns in Fig. 5), but larger with the StratFeedback parameterisation (rightmost columns in Fig. 5). This increase for the StratFeedback cases occurs because the MK18 limit enforces a larger minimum melt rate than that created by the minimum friction velocity. However, the shallow slopes of the ISOMIP+ experiment limit the reliability of the MK18 parameterisation.

Between MOM6 and MITgcm, the behaviour of the StratFeedback parameterisation under each of the warm, cold, and alternative low-velocity limits is consistent, despite melt rates being larger in MITgcm. There are similar percentage decreases in melt rate between the ConstCoeff and StratFeedback experiments despite the variation in the magnitude of melt (Fig. 5). The different magnitude of melt between models may be explained chiefly by the different vertical coordinates (Gwyther et al., 2020), where the $z$-level coordinates of MITgcm result in a coarser vertical resolution near the ice, and therefore a stronger thermal driving, but may also be associated with other model choices such as the vertical mixing scheme and thermal driving sampling depth. Nonetheless, the similar behaviour between the models with and without the stratification feedback parameter gives us confidence in our simulated melt rates, circulation and their feedback.





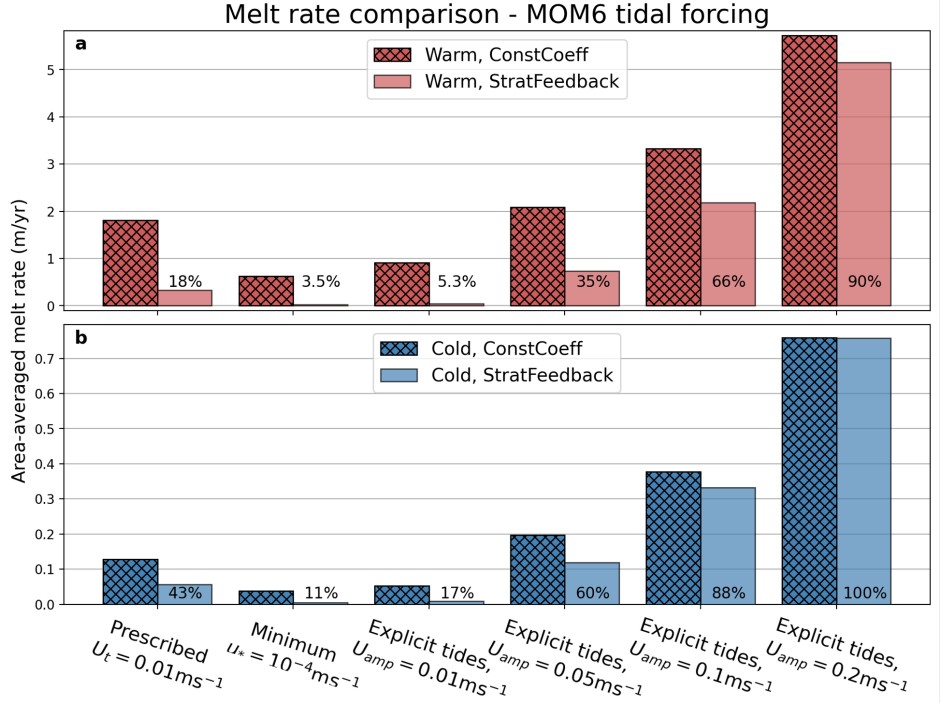

**Figure 8.** Area-averaged melt rates for MOM6 experiments averaged over the last 180 days of the simulation, with either a prescribed tidal velocity of $0.01\,\mathrm{m\,s^{-1}}$, a minimum friction velocity of $1 \times 10^{-4}\,\mathrm{m\,s^{-1}}$, or both the minimum friction velocity and idealised barotropic tides applied at the open-ocean boundary with velocity amplitudes of $0.01\,\mathrm{m\,s^{-1}}$, $0.05\,\mathrm{m\,s^{-1}}$, $0.1\,\mathrm{m\,s^{-1}}$ and $0.2\,\mathrm{m\,s^{-1}}$. Warm experiments are shown in panel (a) and cold in panel (b). Hatched bars are experiments with a constant transfer coefficient and solid colours are with the stratification feedback parameterisation.

## 4.3 Energetic Ice Shelf Cavity Regimes

Motivated by the low melt rates in the idealised ISOMIP+ test cases, we replace the prescribed tidal velocities in Figs. 4-6 with explicit simulation of idealised tides in MOM6. Explicit tides move the experiments to more energetic (and realistic) ice shelf cavity regimes (from the blue and purple to the orange colours in Fig. 7). The cavity circulation is therefore no longer only driven by meltwater buoyancy, and this results in increased melting in both the cold and warm experiments relative to the minimum $u_*$ experiment, which is the control for the tide experiments (Fig. 8, second columns from the left). The magnitude of

melting depends on the amplitude of the tidal forcing (Fig. 8). We see that a $0.05\,\mathrm{m\,s^{-1}}$ amplitude tide gives similar melt rates (within a factor of 1-2) to the prescribed tidal velocity experiment despite the tide amplitude at the boundary being five times greater. This occurs because the tidal velocity amplitude adjacent to the ice-ocean interface is smaller than the forced tide at the open boundary (Fig. S1). In this experiment, the root mean square tidal velocity simulated within the cavity is approximately




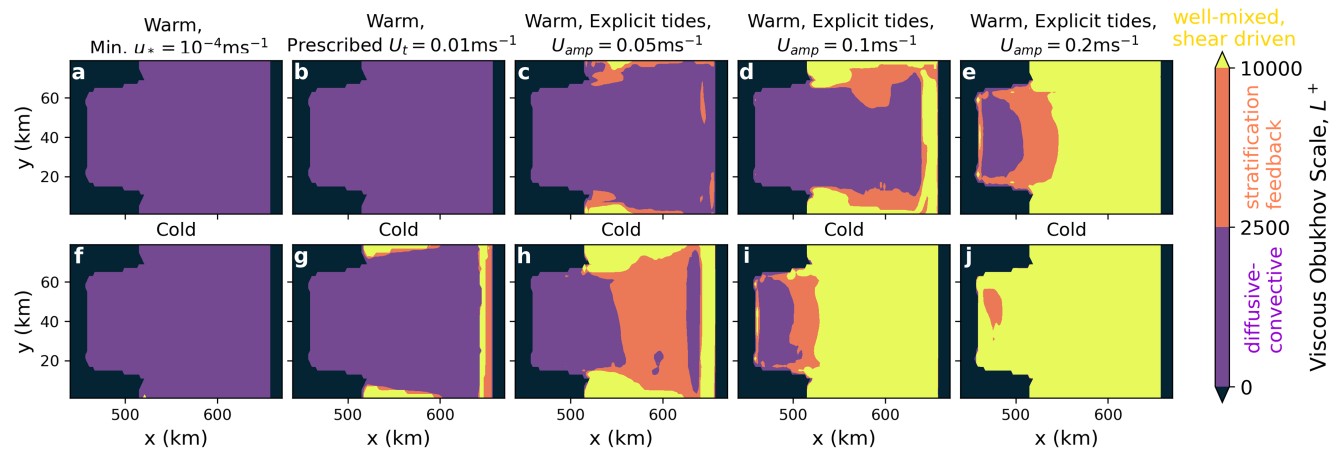

**Figure 9.** Viscous Obukhov Scale, $L^+$, regimes for the MOM6 ice shelf region averaged over the last 180 days of the StratFeedback simulations. For each experiment, the diffusive-convective regime is shown in purple and corresponds to $L^+ < 2500$. The stratification feedback regime is shown in orange, with $2500 < L^+ < 1 \times 10^4$, indicating the region where melting is expected to be steady but the transfer coefficients are suppressed due to the effect of stratification. The well-mixed shear-driven regime with $L^+ > 1 \times 10^4$ and constant (maximal) transfer coefficients is indicated in yellow. Warm experiments are shown in the top row and cold in the bottom row.

$0.02\,\mathrm{m\,s^{-1}}$ (see e.g. Anselin et al., 2023, for a derivation of the tidal velocity contribution to $u_*$), similar to the $0.01\,\mathrm{m\,s^{-1}}$
prescribed tidal velocity.

The largest warm and cold tidal amplitude ConstCoeff experiments ($0.2\,\mathrm{m\,s^{-1}}$) have at least three and six times the magnitude of melt, respectively, compared to the warm and cold control cases with prescribed tidal velocity (Fig. 8, compare leftmost and rightmost hatched columns). The proportional increase in melt rate is even greater for the StratFeedback experiments. This can be explained by a shift in the ice shelf cavity regime with the addition of a strong external velocity. Fig. 7 shows a shift

towards higher friction velocities in the explicit tide experiments (compare the purple and orange colours, which is for the $0.1\,\mathrm{m\,s^{-1}}$ experiment), and a shift out of the stratified regime (indicated by the stippling, assuming a salinity of $S = 34.05$) and into the well-mixed region of $T^* - u^*$ parameter space where the StratFeedback and ConstCoeff parameterisations are equal (Fig. 2d). Increased melt in the highest tide amplitude experiment also leads to cooling and weaker thermal driving (Fig. 7), further shifting the cavity regime to well-mixed conditions.

The behaviour of the StratFeedback parameterisation as the cavity environment becomes progressively more energetic is shown in Fig. 9. With low or no tidal forcing, much of the cavity sits within the diffusive-convective melt regime (purple colours in Fig. 9). As the tidal amplitude is increased, the stratified (orange colours) and well-mixed (yellow colours) shear-driven melt regimes begin to dominate. For the experiment with the largest tidal forcing, only small regions of the ice shelf cavity are within the stratified or diffusive-convective regimes, and the StratFeedback and ConstCoeff parameterisations give

similar melt rates (Fig. 8 ).





There are rectified impacts of the tidal forcing on the circulation and hydrography which are beyond the scope of this paper. However, the idealised tidal simulations demonstrate the difficulty in achieving realistic ice shelf cavity regimes in idealised models. Even with a large tidal forcing of $0.2\,\mathrm{m\,s^{-1}}$ amplitude velocity (corresponding to a $6.4\,\mathrm{m}$ sea level anomaly forcing in this idealised cavity), the warm cavity is not entirely in the well-mixed regime. Idealised ocean models should therefore be

used with caution when assessing melt parameterisations or other ice shelf boundary layer physics, or indeed other aspects of ice shelf cavity circulation.

### 4.4 Realistic Pine Island Glacier Simulation

To assess the parameterisation in a realistic situation where circulation is more complex and the results can be compared to observations, we use the MITgcm Pine Island Glacier setup of Nakayama et al. (2021) (model details in Section 3.2). After

20 days of simulation, melt rates are approximately equilibrated, and we compare the melt rate distributions for three different parameterisation choices averaged over days 20-50. The Holland and Jenkins (1999) parameterisation with the McPhee (1981) $\eta_*$ stability parameter set to 1, hereafter HJ99-neutral, has a mean melt rate nearly identical to the ConstCoeff parameterisation (11.3 m/yr and 11.0 m/yr respectively) due to the similar magnitude transfer coefficients (Table 2, Appendix A1). Melt is enhanced near the grounding line (Fig. 10a), but does not reach observed melt rates of up to 200 m/yr in this region (Shean

et al., 2019; Zinck et al., in review, Fig. S5). When the StratFeedback parameterisation is applied, melt rates decrease as in the idealised experiments, with the average melt rate 40% of the original (Fig. 10b). The reduction indicates that much of the simulated ice shelf is in the stratified melt regime, implying stratification feedback on the melt rate. Furthermore, when we include the MK18 low-velocity limit, the melt rates increase to an average of 8 m/yr (not shown). This melt rate is larger than the StratFeedback simulation (4 m/yr) because the relatively large ice base slopes (up to 30°) contribute substantial melting via

the MK18 parameterisation, but is still weaker than the HJ99-neutral and ConstCoeff parameterisations of ~11 m/yr. All the simulations have mean melt rates less than satellite observations of 12-18 m/yr (Rignot et al., 2013; Depoorter et al., 2013; Liu et al., 2015; Adusumilli et al., 2020; Zinck et al., in review).

Despite this reduction in melt away from observational estimates, the unconstrained drag coefficient allows us to tune the overall melt rate to match the original HJ99-neutral experiment. This requires a drag coefficient of $C_d = 0.0042$, which lies

between the value $C_d = 0.0015$ used in the original simulation and the value $C_d = 0.0097$ suggested by Jenkins et al. (2010). In this tuned StratFeedback simulation (Fig. 10c), the melt rates are stronger near the Pine Island Glacier grounding line and in the ice shelf channels but weaker in more quiescent regions of the ice shelf compared to the HJ99-neutral experiment (Fig. 10d). The tuned StratFeedback simulation enhances melt in regions with large friction velocities (Fig. 10e). Here, the large $L^+$ indicates well-mixed regimes, whereas regions with lower friction velocities have lower $L^+$ and are simulated to be

in the stratified and diffusive-convective regimes (Fig. 10f). The StratFeedback parameterisation therefore enhances the spatial variability in melt beneath Pine Island Glacier. Ocean hydrography and circulation respond to this modified melt rate (see Supplementary Figs. S2,S3,S4).

The difference in the spatial distribution of melt rates between the original simulation and that with the StratFeedback parameterisation highlights the spatial heterogeneity in melt rate regimes within individual ice shelves. However, this Pine



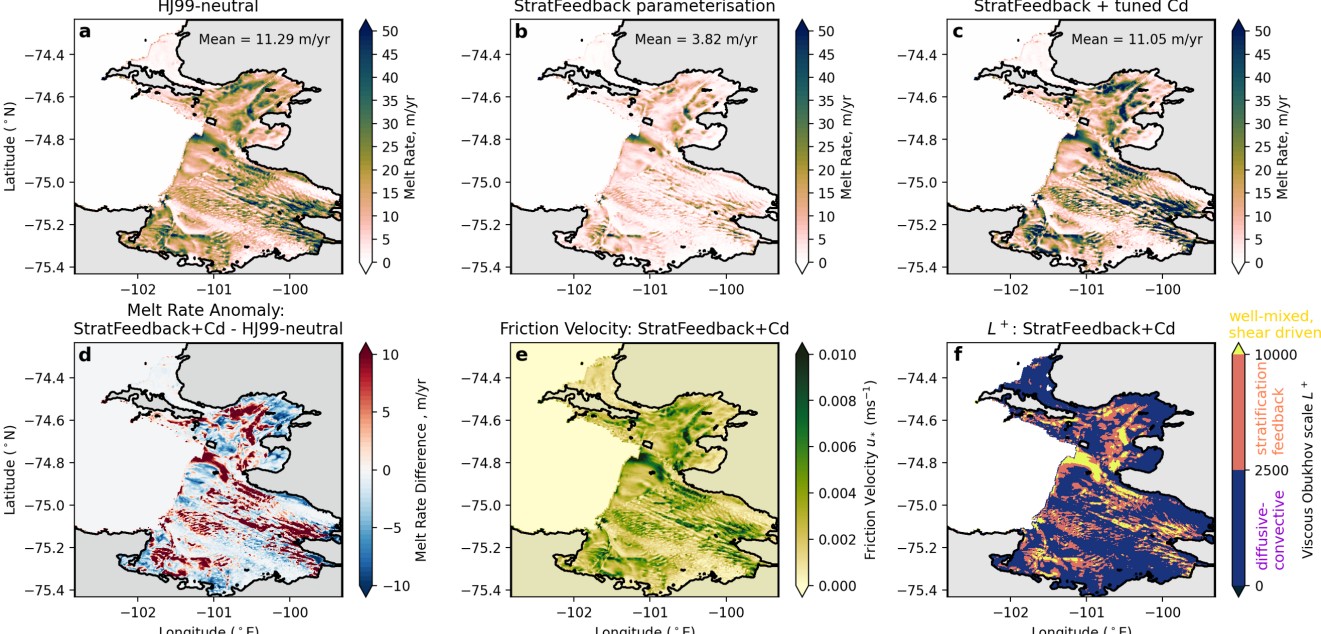

**Figure 10.** MITgcm Pine Island Glacier melt rates averaged over days 20-50 of the simulation, for (a) the HJ99-neutral basal melt parameterisation used in Nakayama et al. (2021); (b) the stratification feedback (StratFeedback) parameterisation (with the same drag coefficient $C_d = 0.0015$) and (c) the stratification feedback parameterisation (with a drag coefficient of $C_d = 0.0042$ to match the average melt rate of (a)), denoted StratFeedback+Cd. Panel (d) compares (c)-(a). The friction velocity and viscous Obukhov scales of the stratification feedback parameterisation with tuned drag coefficient (c) are shown in panels (e) and (f).

Island Glacier simulation also highlights a difference between modelled ocean conditions and *in situ* observations. Without tuning the drag coefficient, the majority of the Pine Island Glacier cavity was simulated to be in a stratified or diffusive-convective regime ($L^+ < 10^4$), rather than the well-mixed shear-driven regime predicted by the large $L^+$ of $1.1 \times 10^4$ from the borehole observation (Fig. 3b). This may be due to the specific location of the borehole (which was in one of the channels approximately halfway between the ice front and grounding zone (Stanton et al., 2013), which in Fig. 10f may be in the well-525 mixed regime), or the representation of bathymetry and ice shelf draft in models. It is also possible that by incorporating more physics into the melt parameterisation, we unveil compensating model biases in realistic Antarctic ice shelf cavity simulations. If models are too smooth (e.g. ISOMIP+), or too coarse to resolve small-scale currents, tides (though models have shown that tides have limited effect at Pine Island Glacier Nakayama et al., 2019), topography, or boundary-intensified flows such as plumes, then the simulated velocities being smaller than reality may have compensated and masked the overestimation 530 of melt rates by shear-driven melt parameterisations. Therefore, it is perhaps not surprising that without further tuning, melt rates with the StratFeedback parameterisation are further from observations than the simulation with the existing HJ99-neutral parameterisation.





## 5 Discussion

In this study, we developed and implemented a basal melt parameterisation that includes the feedback effect of stratification
suppressing turbulence at the ice-ocean boundary layer in ocean models. The parameterisation modifies the three-equation melt
parameterisation (Hellmer and Olbers, 1989; Holland and Jenkins, 1999) transfer coefficients using empirical functional forms
based on Large Eddy Simulations (e.g. Vreugdenhil and Taylor, 2019; Rosevear et al., 2022b, Fig. 1)). Compared to a constant
transfer coefficient formulation, the melt rates are suppressed at low velocity and warm ice shelf cavity conditions (Fig. 2),
where stratification feedback effects on melt are expected to be stronger, due to the relatively higher melt (strengthening the
stratification) in low turbulence conditions. This results in improved melt rates compared to observations for ice shelf cavities
in the stratified but still shear-driven or diffusive-convective ice shelf cavity regimes, particularly the Amery Ice Shelf (Fig.
3). The parameterisation therefore decreases the disagreement between direct melt rate observations and parameterisations in
some ice shelf cavity regimes and leaves them unchanged in others.

We tested the stratification feedback melt parameterisation in model configurations of varying complexity and ice shelf
cavity regimes. Compared to the constant coefficient formulation, idealised ISOMIP+ experiments in MOM6 and MITgcm
saw significant reductions in melt in both the warm and cold simulations (Fig. 5). Circulation, hydrography and kinetic energy
were influenced by the change in melt rate (Fig. 4, 6), with a positive feedback between melt rates and circulation strength.
This melt rate reduction demonstrated that the ISOMIP+ experiments are simulated to be in a quiescent, warm regime with
a low viscous Obukhov scale (Figs. 7,9), and therefore that the stratification feedback parameterisation suppresses melt. This
can be explained by the buoyancy-only ISOMIP+ setup, where there are no external forcings (except for a restoring sponge
layer), and therefore that circulation is driven only by the buoyant meltwater plume. When an idealised external forcing was
applied, with a barotropic tidal forcing in MOM6, melt rates increased as in Gwyther et al. (2016) and regions of the ice shelf
cavity moved into the more energetic, shear-driven regime (Figs. 7,8,9) where the stratification feedback was less pronounced,
per the parameterisation design. However, stratification feedback effects were seen even in the coldest, most energetic cavities,
suggesting a broad relevance of the stratified regime across Antarctica.

However, the melt rate (and therefore ocean circulation) is extremely sensitive to melt parameterisation choices at the low-
velocity limit, primarily because of the low velocities in the smooth, idealised ISOMIP+ cavity. Attempts to connect the
shear-driven melt parameterisation (with the stratification feedback included) and previously suggested velocity-independent
convective melt scaling (Kerr and McConnochie, 2015; McConnochie and Kerr, 2018; Mondal et al., 2019; Schulz et al.,
2022) led to higher melt rates (Fig. 5) than with the chosen minimum friction velocity, but the idealised model framework
made assessing the accuracy of the parameterisation in the context of realistic observations and models difficult. The optimum
low-velocity limit for melt parameterisations is still unknown, and awareness of this sensitivity to parameterisation choices is
important when simulating cavities with low velocities.

A realistic simulation of Pine Island Glacier also showed a significant reduction in melt rate when the stratification feedback
parameterisation was employed (Fig. 10). However, the same average melt rate as the original HJ99-neutral parameterisation
can be achieved with the StratFeedback parameterisation by tuning the drag coefficient, a significant unconstrained parameter.





The ConstCoeff and tuned StratFeedback parameterisation had distinct spatial patterns of melt rate, with the stratification feedback parameterisation having stronger peaks of melt rates in the Pine Island channels and near the grounding line, moving the simulated melt rate pattern closer to observations (Shean et al., 2019; Adusumilli et al., 2020; Zinck et al., in review).

However, assessing the accuracy of the basal melt parameterisations in a regional model compared to observations is difficult because of a lack of direct observations and knowledge about the local ice shelf regimes. Additionally, choices made with the unconstrained drag coefficient can significantly affect conclusions.

Our results demonstrate the importance of testing basal melt parameterisations across various ice shelf cavity regimes. The basal melt-ocean circulation positive feedback makes idealised models extremely sensitive to specific choices in the parame-

terisation, possibly more so than realistic models. However, across all the model experiments, $L^+$ values were low compared to that suggested by *in situ* borehole observations, where many locations had conditions with $L^+ > 10^4$ (Fig. 3b). The inability of models to give as high $L^+$ as observations may be explained by ocean models lacking the small-scale flow variability observed at high frequencies beneath ice shelves, either through not resolving these scales of motion, or due to anomalously smooth topography and ice shape. The results suggest that one reason current parameterisations in ocean models do achieve

relatively good agreement with satellite-derived melt rates (Richter et al., 2022; Galton-Fenzi et al., in prep.) may be due to a compensation of biases: if simulated velocities are weaker than reality and constant transfer coefficients larger than reality, the basal melt transfer velocities would be of the correct magnitude. This compensation, however, would be expected to depend on resolution. Unravelling this compensation led to low melt rates in our StratFeedback parameterisation, highlighting the need for more high-resolution ice shelf cavity simulations as well as for the community to exercise caution around the simulated

velocities in ice shelf cavities.

Furthermore, the difference between MOM6 and MITgcm ISOMIP+ experiments demonstrates the ongoing challenge in unifying parameterisations when ocean models simulate the ice shelf ocean boundary layer differently. Specifically, the vertical resolution and coordinate are important factors, where appropriate choices (using high vertical resolution and appropriately sampling the far-field ocean properties and distributing meltwater) can reduce the sensitivity of simulated melt to model choices

(Gwyther et al., 2020; Scott et al., 2023).

There remain questions around formulating a regime-aware, physically accurate basal melt parameterisation. Future work should further explore the transition between the shear-controlled stratified regime and the transient diffusive-convective regime (e.g. Rosevear et al., 2022b). We have proposed one option for a transition to a velocity-independent convective parameterisation at low velocities, but other physical processes such as diffusive convection are currently not included. Developing a truly

regime-aware parameterisation likely requires further understanding of the physics governing each regime and the transitions, through more high-resolution numerical simulations, laboratory experiments and *in situ* ice shelf-ocean boundary layer observations. For example, conducting similar experiments to Rosevear et al. (2022b) that resolve the boundary layer with shallow ice slopes would fill a currently undersampled regime. Additionally, the transfer and drag coefficients for refreezing (Galton-Fenzi et al., 2012; Gwyther et al., 2016; McPhee et al., 2016) and the effect of heat conduction into the ice shelf (Holland and

Jenkins, 1999; Wiskandt and Jourdain, in review) are also important data-poor factors we have not explored. Finally, our Pine Island Glacier simulation experiments highlight an outstanding unknown in basal melt parameterisations: the drag coefficient





(Gwyther et al., 2016). Further work is required to understand the spatial distribution of drag coefficients around Antarctic ice shelves, and how we can model the momentum boundary layer and its effect on melting in ocean models with varying vertical resolution and coordinates because as demonstrated, regime-aware parameterisations still require tuning in ocean models to obtain realistic melt rates.

This work has focused on basal melt parameterisations in ocean models, but there is also scope to translate its findings into basal melt parameterisations for ice sheet models (e.g. Burgard et al., 2022). Improving the accuracy of our climate and earth system models will require ongoing development of regime-aware basal melt parameterisations and implementation in large-scale models.

## 6 Conclusions

We implemented a basal melt parameterisation which accounts for turbulence suppression due to stratification in the ice shelf-ocean boundary layer in two ocean models. Our main findings discussed in this article are that

- the stratification feedback suppresses melt rates in the idealised ISOMIP+ model experiments compared to the control, constant coefficient melt parameterisation,

- because stratification effects were seen even in the coldest and most energetic cavities, we expect that the stratified regime will have broad relevance to Antarctic ice shelves

- the change in melt rate between the constant coefficient and stratification feedback parameterisations depends on the cavity conditions, including the temperature and presence of external tidal forcing, as well as the choice of low-velocity limit in the melt parameterisation, and is sensitive to the strong feedback with ice shelf cavity circulation

- when tested in a Pine Island Glacier simulation, the spatial distribution of melt was modified by the stratification feedback parameterisation and the melt rate was decreased to 40% of the original parameterisation without further tuning of the drag coefficient. With drag coefficient tuning, the melt rate becomes intensified in glacial channels and near the grounding line, with higher peak melt rates moving closer to satellite observational estimates.

Accurate simulation of Antarctic ice shelf basal melting will require further work to optimise basal melt parameterisations across ice shelf cavities in different thermal and energetic conditions. This will be particularly important in warm cavities with weak flows, where there are large uncertainties in the transitions between ice shelf cavity regimes. Future work should also aim to identify biases in ice shelf cavity regimes in realistic ocean model simulations and explore how the effect of these biases on melt rate may be addressed.

*Code and data availability.* A zenodo repository with processed model output, model configurations as well as the code used to generate figures (the latter which can also be found at https://github.com/claireyung/stratification-feedback-param-paper) will be provided upon





acceptance. The MOM6 and MITgcm melt parameterisation modifications can be found at https://github.com/claireyung/MOM6/tree/dev/gfdl_31May24+iceparam and https://github.com/claireyung/MITgcm/tree/iceparam respectively, and will be copied to the zenodo repository. Barnes et al. (2024) contains a pathway to generate open boundary conditions for MOM6, the output of which is relied on for the file structure of tidal and open boundary conditions.

**Appendix A: Melt Rate Parameterisation Design**

Here, we provide further details on the melt parameterisation formulations. Table A1 presents relevant constants, variables and parameters in addition to Table 1.

### A1 Holland and Jenkins (1999) parameterisation and McPhee (1981) stability parameter

We review the Holland and Jenkins (1999) (HJ99) parameterisation and McPhee (1981) stability parameter, referred to in

this study as HJ99-M81. These studies use both thermodynamics and Monin-Obukhov boundary-layer scalings (Monin and Obukhov, 1954) to quantify momentum, salt and heat transport over the boundary layer. Building on Kader and Yaglom (1972), McPhee et al. (1987) and others, HJ99 express the transfer velocities used in the three-equation parameterisation (Eqns. 1-3) as

$$\gamma_{T,S} = \frac{u_*}{\Lambda_{\text{Turb}} + \Lambda_{\text{Mole}}^{T,S}} \,, \tag{A1}$$

where we use $\Lambda$ to denote the dimensionless factors rather than the original HJ99 $\Gamma$ notation to avoid confusion with Eqn. 4. Here, $\Lambda_{\text{Mole}}^{T,S}$ represents heat and salt transfer associated with molecular diffusion;

$$\Lambda_{\text{Mole}}^{T} = 12.5(\text{Pr})^{2/3} - 6 \,, \tag{A2}$$

$$\Lambda_{\text{Mole}}^{S} = 12.5(\text{Sc})^{2/3} - 6 \,, \tag{A3}$$

with Pr the Prandtl number $\nu/\kappa_T$ and Sc the Schmidt number $\nu/\kappa_S$ (refer to Table A1 for constants and values). $\Lambda_{\text{Turb}}$ represents

transfer associated with turbulence,

$$\Lambda_{\text{Turb}} = \frac{1}{k} \ln\left(\frac{u_* \xi_N \eta_*^2}{f h_\nu}\right) + \frac{1}{2\xi_N \eta_*} - \frac{1}{k} \,. \tag{A4}$$

Here, $\xi_N$ is a dimensionless constant, $h_\nu$ the viscous sublayer thickness, estimated as $h_\nu = 5\nu/u_*$ and $\eta_*$ is the McPhee (1981) stability parameter designed to account for stabilising buoyancy fluxes

$$\eta_* = \left(1 + \frac{\xi_N u_*}{|f| L R_c}\right)^{-1/2} \,. \tag{A5}$$

Refer to Table A1 for definitions. The melt parameterisation contains constant parameters, $\eta_*$ (a function of friction velocity and buoyancy) and the friction velocity, so it is a variable transfer coefficient. In most ocean models that use the HJ99 parameterisation (Losch, 2008; Dansereau et al., 2014), $\eta_*$ is set to 1, representing neutral conditions (which we refer to as



**Table A1.** Table of constants, variables and parameters for Appendix A, in addition to Table 1.

| Symbol | Description | Value |
| --- | --- | --- |
| $\kappa_T$ | Heat diffusivity | $1.41 \times 10^{-7} \text{ m}^2 \text{ s}^{-1}$ |
| $\kappa_S$ | Salt diffusivity | $8.07 \times 10^{-10} \text{ m}^2 \text{ s}^{-1}$ |
| ***HJ99-neutral and HJ99-M81*** | | |
| $\Lambda_{\text{Turb}}$ | Dimensionless turbulent transfer factor | |
| $\Lambda_{\text{Mole}}^{T,S}$ | Dimensionless heat and salt molecular diffusion factor | |
| $\eta_*$ | Stability parameter | |
| Pr | Prandtl number | 13.8 |
| Sc | Schmidt number | 2432 |
| $\xi_N$ | Dimensionless stability constant in $\Lambda_{\text{Turb}}$ | 0.052 |
| $|f|$ | Coriolis parameter (assume a latitude of $75°$) | $1.41 \times 10^{-4} \text{ s}^{-1}$ |
| $h_\nu$ | Viscous sublayer thickness | |
| $L$ | Obukhov length, Eqn. 5 | |
| $R_c$ | Critical Richardson number | 0.2 |
| ***MK18 limit*** | | |
| $\theta$ | Angle from the horizontal | |
| $\chi$ | Scaling factor in McConnochie and Kerr (2018) | 0.086 |
| ***StratFeedback sensitivity*** | | |
| $A_{T,\max}$ | StratFeedback heat transfer coefficient constant of proportionality, max | -10.173 |
| $n_{T,\max}$ | StratFeedback heat transfer coefficient $L^+$ scaling factor, max | 0.273 |
| $A_{S,\max}$ | StratFeedback salt transfer coefficient constant of proportionality, max | -7.903 |
| $n_{S,\max}$ | StratFeedback salt transfer coefficient $L^+$ scaling factor, max | 0.409 |
| $A_{T,\min}$ | StratFeedback heat transfer coefficient constant of proportionality, min | -9.456 |
| $n_{T,\min}$ | StratFeedback heat transfer coefficient $L^+$ scaling factor, min | 0.148 |
| $A_{S,\min}$ | StratFeedback salt transfer coefficient constant of proportionality, min | -6.66 |
| $n_{S,\min}$ | StratFeedback salt transfer coefficient $L^+$ scaling factor, min | 0.206 |

HJ99-neutral parameterisation). HJ99-neutral has varying transfer coefficients due to the dependence on friction velocity, but with much less variability than the StratFeedback parameterisation (within 10% of ConstCoeff melt rates in most relevant con-

ditions). Fig. A1a shows this reduced variability, by plotting the ratio of melt rate calculated with HJ99-neutral across different thermal driving and friction velocity conditions to that from the ConstCoeff parameterisation, as a comparison to Fig. 2d. If we allow $\eta_*$ to vary (the HJ99-M81 form) and compare melt rates to the ConstCoeff parameterisation (Fig. A1c), we can see that melt rates are suppressed under very low friction velocity and high thermal driving conditions (also shown in Fig. A1b, comparing to HJ99-neutral). However, this suppression is less extreme and far less extensive in regime-space than the empirically

derived StratFeedback parameterisation (Fig. 2d).



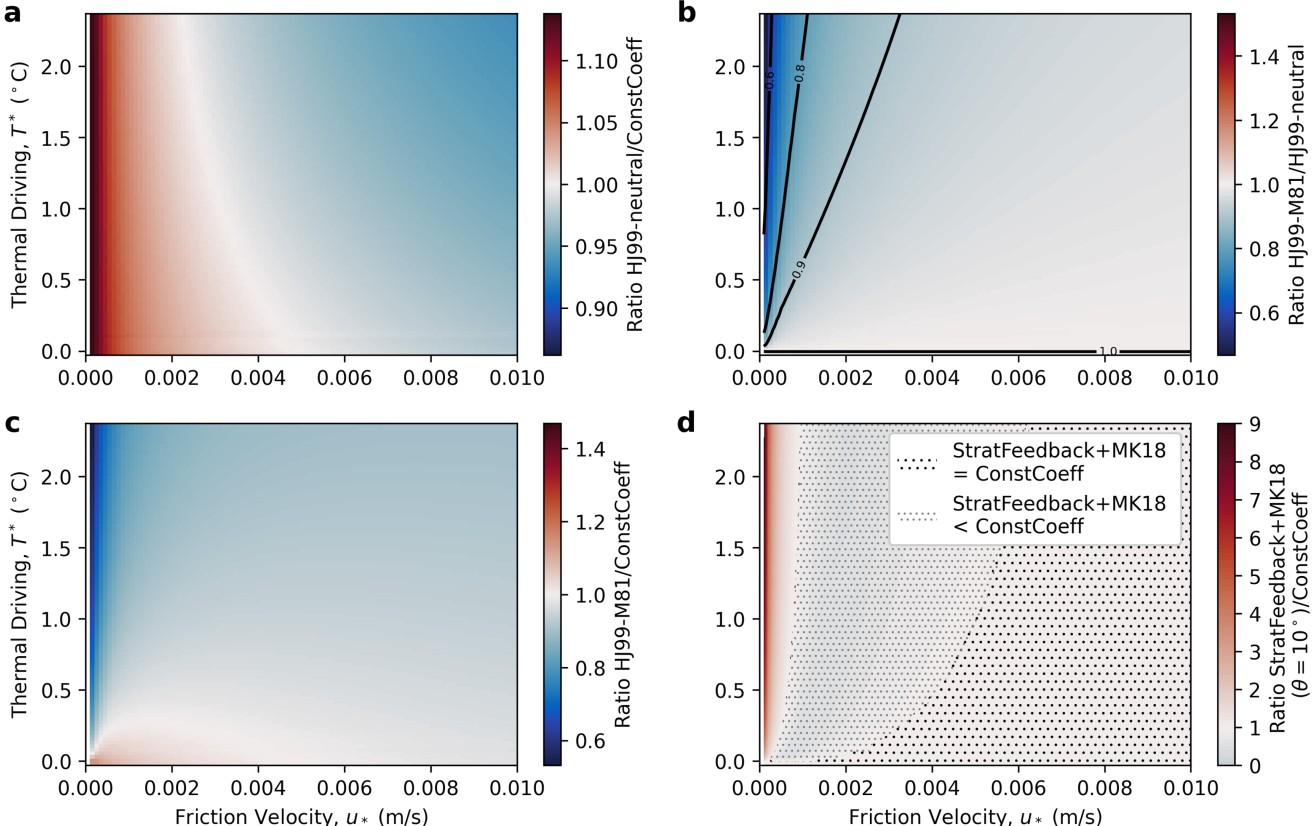

**Figure A1.** Ratio of melt rate calculated from varying-transfer coefficient methods to the constant coefficient parameterisation used in this study, presented in thermal driving – friction velocity regime space, assuming $S_M = 34.5$ g/kg and a pressure of 500 dbar ($\sim 500$ m depth). Panel a shows the Holland and Jenkins (1999) parameterisation with the McPhee (1981) stability parameter (Appendix A1) set to 1 (neutral conditions), and panel c shows it with the McPhee (1981) stability parameter varying. The difference between HJ99-M81 and HJ99-neutral is shown in panel c. Panel d shows the combined stratification feedback and McConnochie and Kerr (2018) low-velocity limit with $\theta = 10°$ (Section 2.5, Appendix A3) compared to ConstCoeff.

## A2 StratFeedback Parameterisation Sensitivity

There is considerable uncertainty in fitting to the Large Eddy Simulation results of Fig. 1. To test the sensitivity of the choice of parameterisation fit to the modelled melt rates, we ran sensitivity experiments with a steeper (max) and shallower (min) line of best fit, where the max and min versions were obtained by changing the critical $L^+$ from $1 \times 10^4$ to $5 \times 10^3$ and $5 \times 10^4$ respectively. The corresponding fit parameters for the StratFeedback transfer coefficients are presented in Table A1. The results of this uncertainty in melt rate for the ISOMIP+ style experiments with a prescribed tidal velocity in MOM6 are presented as uncertainty bars in Fig. 5, indicating that the qualitative results are not sensitive to the exact values of the empirical fit of LES results.





### A3 Transition to McConnochie and Kerr (2018) Parameterisation

To transition smoothly between the shear-driven three-equation parameterisation and a velocity-independent melt parameteri-
sation of McConnochie and Kerr (2018) (MK18) and Mondal et al. (2019) we reformulate the MK18 melt rate in terms of an
effective transfer velocity that depends on temperature, salinity, ice base slope, and constants.

Using laboratory experiments and theoretical models, MK18 and Kerr and McConnochie (2015) estimate the difference in
temperature of the far-field $T_M$ and interface $T_b$ to be

$$T_M - T_b = \frac{\rho_I L + \rho_I c_{p,I}(T_b - T_I)}{\rho_M c_{p,M}} \left(\frac{\kappa_s}{\kappa_T}\right)^{1/2} \left(\frac{S_M - S_b}{S_M - S_I}\right), \tag{A6}$$

where we modify the notation to match Section 2.1, so subscripts $b$, $M$ and $I$ are the boundary, mixed layer (far-field) and ice.
$\kappa_s$ is the compositional diffusivity (salt) whilst $\kappa_T$ is the heat diffusivity.

The ice ablation velocity, or melt rate, is

$$m = \chi \sin^{2/3}\theta \left(\frac{g(\rho_M - \rho_b)\kappa_s^2}{\mu}\right)^{1/3} \left(\frac{S_M - S_b}{S_M - S_I}\right), \tag{A7}$$

where $\chi$ is a non-dimensional constant from McConnochie and Kerr (2018)'s laboratory experiments, $\theta$ is measured from the
horizontal and $\mu = \nu\rho_0$ the dynamic viscosity (Table A1). Note that McConnochie and Kerr (2018) do not test shallow slope
angles, and that with very low slope angles the ice ablation is more likely to be dominated by current shear than buoyant
convection since buoyancy forces decrease as slope angles become lower (Rosevear et al., 2024). However, Mondal et al.
(2019) also suggest a $\sin^{2/3}\theta$ scaling for turbulent boundary layer flows (and a $\sin^{1/4}\theta$ scaling for laminar flows) for a wider
range of slope angles (2°-90° from the horizontal).

We can approximate the density change in Eqn. A7 with a linear equation of state:

$$\rho_M - \rho_b \approx \beta\rho_0(S_M - S_b) - \alpha\rho_0(T_M - T_b), \tag{A8}$$

Eqn. A7 can then be rearranged as

$$\rho_I S_b m = \chi \sin^{2/3}\theta \left(\frac{g\rho_0(\beta(S_M - S_b) - \alpha(T_M - T_b))\kappa_s^2}{\nu\rho_0}\right)^{1/3} \left(\frac{S_M - S_b}{S_M - S_I}\right)\rho_I S_b$$

$$= \underbrace{\chi \sin^{2/3}\theta \left(\frac{g(\beta(S_M - S_b) - \alpha(T_M - T_b))\kappa_s^2}{\nu}\right)^{1/3} \left(\frac{\rho_I S_b}{\rho_0(S_M - S_I)}\right)}_{\gamma_{S,\text{eff}}}\rho_0(S_M - S_b). \tag{A9}$$

We also neglect the heat capacity term with the ice conduction, as we have done with the shear-driven parameterisation
(Section 2.3) and ignore it. Combining Eqns. A6 and A7, we can rewrite the melt rate as a function of the temperature difference





term:

$$m = \chi \sin^{2/3}\theta \left(\frac{g(\rho_M - \rho_b)\kappa_s^2}{\nu \rho_0}\right)^{1/3} (T_M - T_b) \frac{1}{\frac{\rho_I L + \rho_I c_{p,I}(T_b - T_I)}{\rho_M c_{p,M}} \left(\frac{\kappa_s}{\kappa_T}\right)^{1/2}}$$

$$= \frac{c_{p,M}\rho_M}{L\rho_I}(T_M - T_b) \underbrace{\chi \sin^{2/3}\theta \left(\frac{g(\beta(S_M - S_b) - \alpha(T_M - T_b))}{\nu}\right)^{1/3} \kappa_s^{1/6}\kappa_T^{1/2}}_{\gamma_{T,\text{eff}}} . \tag{A10}$$

In this way, we define effective transfer velocities corresponding to the convective flow limit that may be more appropriate than the shear-driven melt transfer velocities $\gamma_{T,S} = u_* \Gamma_{T,S}$ at low velocities.

The ratio of $\gamma_T/\gamma_S$ is then

$$\frac{\gamma_T}{\gamma_S} = \frac{\rho_0 \kappa_s^{1/6}\kappa_T^{1/2}(S_M - S_I)}{\kappa_s^{2/3}\rho_I S_b} = \frac{\rho_0 S_M}{\rho_I S_b}\sqrt{\frac{\kappa_T}{\kappa_S}} . \tag{A11}$$

which for values given in Tables 1, A1 gives $\sim$20, smaller than the ratio in the well-mixed regime of $\sim$30 but in agreement with the StratFeedback parameterisation in the diffusive-convective regime (Fig. 1c).

To connect this velocity-independent parameterisation with the shear-driven formulation, we choose to take the greater of the MK18 effective transfer velocities and the shear-driven $u_*\Gamma_{T,S}$, where the MK18 formulation will be the larger of the two at small friction velocities. Since the MK18 effective transfer velocity depends on the boundary salinity and temperature, iteration will be required to converge to a solution for melt rate. This regime-aware parameterisation is shown in Fig. 2d, where the melt rate contours follow the shear-driven formulations (either StratFeedback or ConstCoeff) at higher friction velocities before transitioning to the velocity-independent melt rate in low-velocity conditions. This parameterisation depends on slope angle, and the local slope angle can be calculated from the ice base taking the maximal local angle.

The effect of the MK18 limit combined with the StratFeedback parameterisation on melt rate is compared to the ConstCoeff parameterisation in Fig. A1d. As in Fig. 2c, melt rates are identical to ConstCoeff in cold and fast conditions (large stippling). Moving to slower conditions in the stratified regime, melt rates are suppressed according to the StratFeedback parameterisation (small stippling). However, in very low-velocity conditions, melt rates are enhanced compared to the ConstCoeff parameterisation due to the MK18 limit (red colours). Note this does not mean melt rates increase with decreasing velocity, rather they become independent of velocity (Fig. 2b) whereas the ConstCoeff melt rate continues to decrease.

## Appendix B: Observational Data

Table B1 presents the observational data and computed melt rates used in Fig. 3. Note that parameterised melt rates (ConstCoeff and StratFeedback) and thermal driving may differ from calculated melt rates in the original references due to different choices of drag and transfer coefficients, as well as uncertainty both in hydrographic properties, instruments and collection of the data from the references (labelled by [a]). We do not perform an uncertainty analysis, but there is considerable uncertainty in the computed melt rates and observed melt rates, and observations are localised and may not represent conditions throughout each ice shelf.



**Table B1.** Table of input and computed values for borehole observation – melt rate parameterisation comparison in Fig. 3. $T_M$, $p$, $S_M$ and $u$ were obtained from the references as time-mean ice shelf far-field input variables for the three-equation parameterisation, and the computed thermal driving $T^*$, and melting under the ConstCoeff (CC) and StratFeedback (SF) parameterisations are presented, as are the directly observed melt rates.

| Location | $T_M$ (°C) | $S_M$ (g/kg) | $p$ (dbar) | $u$ (m/s) | $T^*$(°C) | Obs. melt (m/yr) | CC (m/yr) | SF (m/yr) |
|---|---|---|---|---|---|---|---|---|
| Amery | -2.1[a] | 34.59[a] | 523 | 0.04 | 0.19 | 0.46 | 1.17 | 0.87 |
| Rosevear et al. (2022a) | | | | | | | | |
| Filchner-Ronne (FRIS) | -2.39[a] | 34.51 | 700[a] | 0.06[a] | 0.03 | 0.55 | 0.33 | 0.33 |
| Jenkins et al. (2010) | | | | | | | | |
| Larsen C | -2.08 | 34.54 | 304 | 0.07[a] | 0.05 | 0.70 | 0.49 | 0.49 |
| Davis and Nicholls (2019) | | | | | | | | |
| Ross - Summer (RIS S) | -1.68 | 34.3 | 229 | 0.1 | 0.37 | 2.7 | 5.71 | 5.71 |
| Stewart (2018) | | | | | | | | |
| Ross - Winter (RIS W) | -1.94 | 34.5 | 229 | 0.125 | 0.13 | 1.4 | 2.42 | 2.42 |
| Stewart (2018) | | | | | | | | |
| Ross Grounding Zone (WGZ) | -2.3[a] | 34.74[a] | 665 | 0.015 | 0.1 | 0.05 | 0.25 | 0.11 |
| Begeman et al. (2018) | | | | | | | | |
| George VI | 0.3 | 34.62 | 317* | 0.04*,[a] | 2.44 | 1.4 | 17 | 7.4 |
| Kimura et al. (2015)* | | | | | | | | |
| Thwaites Glacier | -0.3[a] | 34.38[a] | 515[a] | 0.03 | 1.97 | 3.8[a] | 10.1 | 3.7 |
| Davis et al. (2023) | | | | | | | | |
| Pine Island Glacier | -0.82 | 33.85 | 460 | 0.13 | 1.39 | 14.6 | 29.6 | 29.6 |
| Stanton et al. (2013) | | | | | | | | |

[a] Values should be considered approximate since they are calculated from visual inspection or unclear data averaging. *Velocity from Middleton et al. (2022) rather than Kimura et al. (2013). Note friction velocities are calculated with drag coefficient $C_d = 0.0025$ and that parameterised melt rates may differ from that stated by individual studies and Rosevear et al. (2022a) due to the use of the ConstCoeff melt parameterisation and different parameter choices including $C_d$. We do not perform an uncertainty analysis, but quoted uncertainties for some ice shelves are large.

*Author contributions.* CKY, MGR, AKM and AMH designed the study. CKY coded the parameterisations, performed the analyses and wrote the first draft of the manuscript. MGR provided the LES data. YN contributed to the MITgcm control setups. All authors contributed to the interpretation of results and editing of the final manuscript.

*Competing interests.* The authors do not declare any competing interests.



*Acknowledgements.* We acknowledge Pedro Colombo, Angus Gibson, Ashley Barnes, Robert Hallberg, Gustavo Marques, Alistair Adcroft, Matthew Harrison, Olga Sergienko and Stephen Griffies for technical assistance with MOM6. In particular, we acknowledge Gustavo Marques for developing and sharing the MOM6 ISOMIP+ experimental setup.

735 This research was undertaken with the assistance of resources and services from the National Computational Infrastructure (NCI), which is supported by the Australian Government. This research was also supported by the Australian Research Council Special Research Initiative, the Australian Centre for Excellence in Antarctic Science (Project Number SR200100008) and the Australian Research Council Discovery Project DP190100494. CKY was supported by an Australian Government Research Training Program (RTP) Scholarship and the Consortium for Ocean Sea Ice Modelling in Australia (COSIMA).





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
