# Peer review of "Stratified suppression of turbulence in an ice shelf basal melt parameterisation"

_EGUsphere, 2024_

## Referee Comment (RC1)

**Review:**

**Egusphere-2024-3513: Stratified suppression of turbulence in an ice shelf basal melt parameterisation**

**Overall Statement:**

The submitted manuscript targets a very important goal, to improve parameterized melt rates over a range of oceanographic forcings in regional-scale ice shelf-ocean models. This is certainly a worthwhile study that will eventually produce a meaningful contribution to the cryosphere science community. The present manuscript draft starts out fairly strong, but becomes cluttered and hard to follow as it progresses, making the Results Section actually quite hard to interpret based on which methodological approach was taken. There are also some holes in the approach that I have outlined as areas of improvement. While I do not believe that all of these suggested comments need to be implemented, the authors should really consider them and at least add caveats to the text for transparency. The suggested changes will constitute a major revision, but I do believe that this study worthy of eventual publication and will make a nice contribution to the cryosphere science community.

**Larger Comments:**

The Introduction is quite good, but lacks some qualification on the effects of realistic small-scale slopes on ice shelf basal melting around Antarctica. Please add several sentences that play this out some. This could include some discussion of the very interesting variation in melt in a terrace, for instance.

Need to define the regions of the boundary layer in the Introduction, as there are parts of the Methodology that are unclear to the reader as to which part of the boundary layer the authors are referring to.

Methodology says very little about salinity differences, which are the key driver of stratification in the boundary layer in a warm cavity ice shelf.

Introduction should state the velocity ranges considered here before discussing friction velocity.

Consider boiling down the parameterization results to Stanton Numbers and constant Gammas, so that larger-scale models and observations that do not resolve all the way up to the ice base can implement these results into something easily useable.

Authors generally do not seek to ground their modeling in observations of the highly varied and sloping bases of Antarctica's ice shelves.

I think the authors have a real opportunity here to implement the StratFeedback+MK18 parameterization in the MITGCM Pine Island Glacier model run to take into account the external shear-driven turbulence, near-ice stratification, and the destabilizing effect rising meltwater on

sloping ice bases. This will really round out the study and should test the influence of external turbulence plus localized rising plumes in a significant manner.

Generally, the Results Sections are hard to interpret, because it is hard for the reader to disentangle the details of which Methodology was used for each section.

**Specific Revisions:**

**Abstract:**

This is a long abstract that Microsoft Word registers as 257 words. Please double check that this fits within the journal's word count limit.

Li 7: "and diffusive convection plays a role…" This is an unfinished statement. Please rewrite to finish this thought.

Li 11 – 15: This section focuses on the suppression of melt by stratification, but does not mention the effects of diffusive convection, which is mentioned previously. Rewrite this section to add some discussion of diffusive convection.

**Introduction:**

Li 23: It would be helpful to add a more recent citation here on recent acceleration in melting; the latest one was over a decade ago in 2014.

Li 32: While not from Antarctica, this study is very applicable to stratified ocean-driven melt of ice shelves and could be added to this list: (Washam et al., 2020). It also has to do with buoyancy-driven circulation and melt, so probably fits into the introduction well.

Washam, P., Nicholls, K. W., Münchow, A., & Padman, L. (2020). Tidal modulation of buoyant flow and basal melt beneath Petermann Gletscher Ice Shelf, Greenland. *Journal of Geophysical Research: Oceans*, *125*(10), e2020JC016427.

Li 39: I suggest to change to "the variable molecular diffusion of heat and salt"

Li 46: Should this be "turbulent mixing and heat **and salt** transport"

Li 50 – 54: I think somewhere in this part of the introduction or before it should be mentioned that "buoyancy-driven convection" only enhances melt along sloping ice bases, and that the growing number of observations from beneath ice shelves show that their bases' are quite rough with many slopes.

Li 70: Please add one or both of the following citation to the statement that variation in melt rate within each ice shelf is significant: (Vaňková and Nicholls, 2022; Vaňková et al., 2023).

Vaňková, I., & Nicholls, K. W. (2022). Ocean variability beneath the Filchner-Ronne ice shelf inferred from basal melt rate time series. *Journal of Geophysical Research: Oceans*, *127*(10), e2022JC018879.

Vaňková, I., Winberry, J. P., Cook, S., Nicholls, K. W., Greene, C. A., & Galton-Fenzi, B. K. (2023). High spatial melt rate variability near the Totten Glacier grounding zone explained by new bathymetry inversion. *Geophysical Research Letters*, *50*(10), e2023GL102960.

Li 87: This statement is way oversimplified and should be removed: "whereas Antarctic ice shelves are generally weakly sloped (<1°)". There are a growing number of observations that show this is not the case over many scales, from scalloped morphology (<1 m) to a several km grounding zone.

**Melt Parameterisation Design and Validation:**
**The Three-Equation Melt Paramterisation and Transfer Coefficients**

Li 127: I would say the ice-ocean boundary temperature is also a key unknown, but if you solve the three equations $T_b$ and $S_b$ will drop out through the quadratic expression. Perhaps you can add some discussion on this to this section, which is typically glossed over in papers. This could also be added to the Li 132.

Li 140: Also an opportunity to cite the Washam et al. (2020) paper.

Li 146 – 151: Again, this is focused on flat portions of the ice shelf and requires a qualifying statement that sloped ice can melt faster that the J10 and HJ99-M81 parameterizations, e.g., Schmidt et al., (2023).

Li 148: I think it would be helpful to add the M81 stability parameter as an equation, so that the reader can be compare it with (5) in the following discussions. I do see it later in the Appendix, but it may help to have it in the main body or at least reference that it is in the Appendix.

Li 151 – 160: Please include a discussion of Washam et al., (2023) and Lawrence et al., (2023) in this section on drag coefficients, as both studies quantified ice shelf morphology and related them to a $u_*$ and $C_D$ from observations.

Washam, P., Lawrence, J. D., Stevens, C. L., Hulbe, C. L., Horgan, H. J., Robinson, N. J., ... & Schmidt, B. E. (2023). Direct observations of melting, freezing, and ocean circulation in an ice shelf basal crevasse. *Science Advances*, *9*(43), eadi7638.

Lawrence, J. D., Washam, P. M., Stevens, C., Hulbe, C., Horgan, H. J., Dunbar, G., ... & Schmidt, B. E. (2023). Crevasse refreezing and signatures of retreat observed at Kamb Ice Stream grounding zone. *Nature Geoscience*, *16*(3), 238-243.

**Stratification Feedback on Turbulence – Insights from Large Eddy Simulations:**
Li 161 – 167: This paragraph never mentions salinity, which is the principle driver of density at the ocean temperatures responsible for melting ice shelves. Please properly attribute stratification to difference in density, driven by salinity changes. Additionally, and potentially more important,

this discussion only applies to flat or gently-sloping ice where meltwater pools instead of rises vigorously to act as a source of turbulence that destratifies the boundary layer. This must be said in this paragraph also.

Li 173: Does a small $L^+$ here refer to an absolute sense or a highly negative value? This is slightly non-intuitive, since there is a negative in front of $u_*$ in (5), g is a positive 9.80 m/s2, and one might expect $T_b - T_M$ to be larger (more negative) than $S_b - S_M$, which would result in a positive buoyancy flux. Please spell this out for the reader, or preferably, move the expressions around in the 3 equation parameterization to place a positive sign in front of the heat/salt flux and make it $T_M - T_b$ ($S_M - S_b$), then place a negative sign in front of the heat conduction into the ice shelf.

Li 173: Is this kinematic viscosity or eddy viscosity?

Li 189 – 190: I do not understand how the cooling effect of melting is not accounted for in $L^+$, since the buoyancy term ($B_b$) should exhibit some change in $T_M$ and $S_M$ as the boundary layer cools and freshens from melting. Unless $T_M$ and $S_M$ are chosen at a sufficient distance to be truly "far-field." To be honest, I have a hard time understanding where $T_M$ and $S_M$ are chosen in most papers. Please elucidate this more clearly in this section.

**Stratification Feedback Parameterisation Design:**

Li 194: Add "along flat or gently-sloping ice to the end of this sentence."

Li 205 – 207: Briefly comment on how neglecting the heat conductive flux will influence results, citing literature that has considered this variable, e.g., Arzeno et al. (2014), Washam et al. (2020), and others. Note that importantly at low ocean heat fluxes, the conductive heat flux through the ice shelf can be nearly as high as the ocean heat flux, which plays an important role in transitioning from melting to freezing.

Arzeno, I. B., Beardsley, R. C., Limeburner, R., Owens, B., Padman, L., Springer, S. R., ... & Williams, M. J. (2014). Ocean variability contributing to basal melt rate near the ice front of Ross Ice Shelf, Antarctica. *Journal of Geophysical Research: Oceans*, *119*(7), 4214-4233.

Li 211 – 212: The velocity in the boundary layer should be less than far-field, because the oceanic flow is starting to feel the friction of the ice base, which generates turbulent eddies that mix heat and salt towards the ice. There are also multiple regions of the boundary layer, such as the outer, surface, and viscous sublayer, that have not been properly defined at this point (see general comment). All of this needs to be spelled out clearly to the reader, and sentences like this are presently confusing, because the boundary layer has not been properly introduced.

Li 216 – 217: If the whole domain is laminar, then there is no turbulence, right? Then if the drag coefficient is related to the friction velocity and therefore turbulence, how can there be a drag coefficient if there is no turbulence? I realize that this is a literature review, but I think this should be presented more clearly to the reader.

Li 221: Please change to vary the thermal and haline Stanton numbers ($\Gamma_T((C_D)^{1/2})$ and $\Gamma_T((C_D)^{1/2})$))

Li 223 – 228: This sentence suggests that the authors are solving the two-equation parametrization, where there is a salinity difference across the boundary layer of 0. I don't think this is actually the case, but it would be helpful for the authors to clarify this in the text.

Li 223 – 239: I believe that it is roughly an order of magnitude greater, but it would be helpful to just state the actual far-field velocity range that could produce $u_*$ values of 0 – 0.010 m/s.

**Comparison to Observations:**

Li 241 – 260: See comment on Figure 3 below also. Please add Ross Ice Shelf data from Washam et al. (2023) to these plots. While not Antarctica, it may also be helpful to place the detailed data from Petermann Glacier into this discussion.

Washam, P., Lawrence, J. D., Stevens, C. L., Hulbe, C. L., Horgan, H. J., Robinson, N. J., ... & Schmidt, B. E. (2023). Direct observations of melting, freezing, and ocean circulation in an ice shelf basal crevasse. *Science Advances*, *9*(43), eadi7638.

Washam, P., Nicholls, K. W., Münchow, A., & Padman, L. (2020). Tidal modulation of buoyant flow and basal melt beneath Petermann Gletscher Ice Shelf, Greenland. *Journal of Geophysical Research: Oceans*, *125*(10), e2020JC016427.

Li 248: I would be careful to say that ignoring heat conduction makes no difference on melt rate parameterization at low ocean heat and salt fluxes, i.e., cold cavity conditions.

**Limiting to a Velocity-Independent Parameterisation:**

Li 266: Following the comment above, it would be helpful to define what velocity range constitutes "low-velocity."

Li 274 – 275: Thank you for identifying the varying ocean-forced mechanism that relate to differing ice slopes.

Li 299: turbulent or molecular diffusion? If molecular, then the sublayer thickness will need to be accounted for and the temperature and salinity gradient across it. If turbulent, please explain this further. Is it just a really low U multiplied by sqrt($C_D$)?

Li 266 – 306: I see no discussion of a combined StratFeedback+MK18 in this section, but (I think) this is most likely what happens in the real world along sloping ice shelf bases and I see it in Figure 2. I suggest to add a few sentences that discuss this parameterization to this section.

**Model Configurations**
**ISOMIP+ Setup and Modifications**

Li 324 – 325: I do not think that the cold cavity setup is restored to a very realistic T/S profile. It is fine to take a full isothermal temperature profile to represent deep convection, but if that is the case, then the salinity profile should also be nearly uniform. In any case, the choice of a surface salinity of 33.8 g/kg seems too fresh if all the freshening is to be accounted from ice melt. Take a look at hydrographic sections in front of the Ross and Filchner-Ronne and adjust accordingly. It doesn't have to be perfect, but I suggest a lower salinity range.

Li 329: Please consider adding in the heat conduction term to these models, as it will become important in the cold cavity, low heat/salt flux scenarios, especially when the ice draft is thin. If this is not possible, remark on this as a weakness of this experiment and discuss the caveats.

**Idealised MOM6 Configuration**

LI 335 – 349: Please explicitly mention the uppermost layer vertical grid cell size range here and remark on how well it represents observations of the boundary layer beneath ice shelves.

**Idealised MITgcm Configuration**
Li 351 – 3558: Similarly, remark on whether the 5 m partial grid cell can adequately resolve the boundary layer.

**Idealised Explicit Tidal Forcing:**

Li 360 – 372: Do these tides pass the critical M2 latitude in your simulations? Is this included?

Li 370 – 372: Do these decreased tidal velocities represent observations, e.g., Jenkins et al. (2010) or Davis & Nicholls (2019)? I don't think so. Please remark on why this is the case.

**Pine Island Glacier Configuration:**

Li 388 – Li 390: I highly recommend adding in the StratFeedback+MK18 parameterization to this study, as you are now dealing with a (somewhat) realistic model configuration that will experience external shear-driven turbulence, stratification, and rising meltwater plumes. This would really take this study over the top!

**Results:**
**Idealised ISOMIP+ Results:**

Li 396 – 398: Add a similar discussion on salinity differences, which are the primary driver of stratification (last time I say this), and see comment below on Figure 4.

Li 408 – 414: Perhaps this can wait until the Discussion Section, but I think $L^+$ could be artificially low in these simulations, because of the course grid size. Consider adding a section that explicitly discusses how vertical resolution affects $L^+$ and compare it to either high resolution model runs or observations to properly ground these results.

Li 429: Suggest to not cite manuscripts in preparation.

**Sensitivity to the Low-Velocity Limit:**

431: Wait, were tides included in all of the results from the previous section? If so, it was not clear to the reader, so please restate it in that section. Later in Li 439 I see that it is fully thermohaline. Please make this clearer.

**Energetic Ice Shelf Cavity Regimes:**

Li 460 – 491: This section should evolve after the appropriateness of $L^+$ in these simulations has been assessed following the prior comment (Li 408 - 414). Or, this can wait to the discussion section.

**Realistic Pine Island Glacier Simulation:**

Li 493 – 532: I highly suggest to also implement the StratFeedback+MK18 parameterization into this analysis, i.e., a parameterization that includes the influence of ice base slope, stratification, and external turbulence.

Li 521 – 524: If I understand this right, this was hardly a 'tuning' of the drag coefficient and more of picking a single observed drag coefficient and applying it to the whole model. At this point it is quite difficult to follow what method has been used where. Regardless, I would not refer to this as a 'tuning,' but instead an 'altering' of the drag coefficient, then please state explicitly in this section what $C_D$ was changed from and to. I also took a look back at the Stanton et al. (2013) paper and don't see a value for $C_D$, but instead only a timeseries of $u_*$ without any mention of U. How as $C_D$ computed then? Another approach to this problem would be to use the range of $C_D$ values observed beneath ice shelves and force the model with each of them to see how it influences the melt rates.

**Discussion:**

Li 534 – 572: All of this reads more like a Summary than a Discussion and should be rewritten to provide more helpful analysis of the results.

Li 573 – 585: Ok, this somewhat satisfies my prior comments on the validity of $L^+$ in these simulations, but given (what I think is) the goal of this study to more accurately parameterize melt rates, I think this section should be expanded to include model-obs comparisons or coarse-fine model comparisons.

**Figures:**

**Figure 1d:** Consider adding lines for $C_D$ from more observations beneath ice shelves, such as Jenkins et al. (2010), Davis and Nicholls (2019), Washam et al. (2023), and Lawrence et al. (2023).

**Figure 2b:** Consider adding a range of ice base angles (4 or 5 angles between 10° and 90°) to this figure or to the Appendix, since 10° seems to be somewhat arbitrary and low, without any real

acknowledgement of the many small-scale slopes observed beneath ice shelves, e.g., Dutrieux et al. (2014), Schmidt et al. (2023), Lawrence et al. (2023), Washam et al. (2023), etc…

Dutrieux, P., Stewart, C., Jenkins, A., Nicholls, K. W., Corr, H. F., Rignot, E., & Steffen, K. (2014). Basal terraces on melting ice shelves. *Geophysical Research Letters*, *41*(15), 5506-5513.

**Figure 3:** Please add Ross Ice Shelf data from Washam et al. (2023) to these plots. While not Antarctica, it may also be helpful to place the detailed data from Petermann Glacier into this figure.

Washam, P., Lawrence, J. D., Stevens, C. L., Hulbe, C. L., Horgan, H. J., Robinson, N. J., ... & Schmidt, B. E. (2023). Direct observations of melting, freezing, and ocean circulation in an ice shelf basal crevasse. *Science Advances*, *9*(43), eadi7638.

Washam, P., Nicholls, K. W., Münchow, A., & Padman, L. (2020). Tidal modulation of buoyant flow and basal melt beneath Petermann Gletscher Ice Shelf, Greenland. *Journal of Geophysical Research: Oceans*, *125*(10), e2020JC016427.

**Figure 4:** I suggest to add two rows that are similar to a-h, but present Salinity.

**Figure 7**: Given that there have been scalebar changes in the comparison of cold and warm cavities in Fig 4 and 6, I suggest to change the vertical axis in panel **b** and use lower melt rate contour lines to make it more useful. Simply state once again to note the scale change.

**References:**

Arzeno, I. B., Beardsley, R. C., Limeburner, R., Owens, B., Padman, L., Springer, S. R., ... & Williams, M. J. (2014). Ocean variability contributing to basal melt rate near the ice front of Ross Ice Shelf, Antarctica. *Journal of Geophysical Research: Oceans*, *119*(7), 4214-4233.

Dutrieux, P., Stewart, C., Jenkins, A., Nicholls, K. W., Corr, H. F., Rignot, E., & Steffen, K. (2014). Basal terraces on melting ice shelves. *Geophysical Research Letters*, *41*(15), 5506-5513.

Lawrence, J. D., Washam, P. M., Stevens, C., Hulbe, C., Horgan, H. J., Dunbar, G., ... & Schmidt, B. E. (2023). Crevasse refreezing and signatures of retreat observed at Kamb Ice Stream grounding zone. *Nature Geoscience*, *16*(3), 238-243.

Vaňková, I., & Nicholls, K. W. (2022). Ocean variability beneath the Filchner-Ronne ice shelf inferred from basal melt rate time series. *Journal of Geophysical Research: Oceans*, *127*(10), e2022JC018879.

Vaňková, I., Winberry, J. P., Cook, S., Nicholls, K. W., Greene, C. A., & Galton-Fenzi, B. K. (2023). High spatial melt rate variability near the Totten Glacier grounding zone explained by new bathymetry inversion. *Geophysical Research Letters*, *50*(10), e2023GL102960.

Washam, P., Nicholls, K. W., Münchow, A., & Padman, L. (2020). Tidal modulation of buoyant flow and basal melt beneath Petermann Gletscher Ice Shelf, Greenland. *Journal of Geophysical Research: Oceans*, *125*(10), e2020JC016427.

Washam, P., Lawrence, J. D., Stevens, C. L., Hulbe, C. L., Horgan, H. J., Robinson, N. J., ... & Schmidt, B. E. (2023). Direct observations of melting, freezing, and ocean circulation in an ice shelf basal crevasse. *Science Advances*, *9*(43), eadi7638.

---

## Author Response (AR1)

**Response to reviewer comments for the manuscript Stratified suppression of turbulence in an ice shelf basal melt parameterisation**

EGUsphere-2024-3513 C.K. Yung, M.G. Rosevear, A.K. Morrison, A.McC. Hogg & Y. Nakayama April 20, 2025

We thank the three reviewers for their thorough reading of our paper and insightful comments and feedback. We have spent time thinking about their suggestions. In particular, we have rerun the Pine Island Glacier model simulations by tuning the drag coefficient in all simulations to better compare with satellite-derived observations, and then compared the spatial distribution of melt rates between them. We hope that we have implemented the suggested changes to the satisfaction of the reviewers, but are very happy to continue discussing any aspects of the review and manuscript.

Below we respond to each comment in turn (with our responses indicated in blue) and changes that we have made to the manuscript. Edits to the text are written in purple, and line numbers refer to the revised manuscript.

**Reviewer 1**

The submitted manuscript targets a very important goal, to improve parameterized melt rates over a range of oceanographic forcings in regional-scale ice shelf-ocean models. This is certainly a worthwhile study that will eventually produce a meaningful contribution to the cryosphere science community. The present manuscript draft starts out fairly strong, but becomes cluttered and hard to follow as it progresses, making the Results Section actually quite hard to interpret based on which methodological approach was taken. There are also some holes in the approach that I have outlined as areas of improvement. While I do not believe that all of these suggested comments need to be implemented, the authors should really consider them and at least add caveats to the text for transparency. The suggested changes will constitute a major revision, but I do believe that this study worthy of eventual publication and will make a nice contribution to the cryosphere science community.

We thank the reviewer for their valuable comments that have helped to improve the manuscript, and also thank them for pointing out highly relevant literature.

**Larger Comments:**

1. The Introduction is quite good, but lacks some qualification on the effects of realistic small-scale slopes on ice shelf basal melting around Antarctica. Please add several sentences that play this out some. This could include some discussion of the very interesting variation in melt in a terrace, for instance.

Thank you for the suggestion. Your comment highlighted the many different scales involved in ice shelf melting. We have added a paragraph in the Introduction discussing the various scales of ice shelf melt and ice base variation and comparing them to the much larger scales of ocean model grid sizes:

L102-118: It is important to highlight the many spatial scales involved in ice shelf basal melting. Considering vertical resolution, the processes within the ice shelf ocean boundary layer can be less than  $\mathcal{O}(10^{-3})$  m in size, hence the need for basal melt parameterisations in ocean models. Horizontally, the ice shelf base and bottom topography have significant spatial variability on scales between  $\mathcal{O}(10^{-1}-10^3)$  m, with melt rate varying correspondingly (Nicholls

et al., 2006; Dutrieux et al., 2014; Alley et al., 2016; Watkins et al., 2021; Schmidt et al., 2023; Washam et al., 2023; Wåhlin et al., 2024). For example, in an ice base crevasse, melt rates can be enhanced at the terrace side-walls, with freezing by buoyant, supercooled water at the top of the crevasse (Washam et al., 2023), indicating multiple physical drivers of melt within a small distance. A variety of ice features such as scallops and terraces can form depending on the ice melt regime (Washam et al., 2023; Wåhlin et al., 2024). Though idealised and process models have simulated some of these small-scale features (Jordan et al., 2014; Zhou and Hattermann, 2020; Wilson et al., 2023), and some high-resolution regional models may capture part of the spatial variability (Nakayama et al., 2019, 2021; Shrestha et al., 2024), large-scale ocean models generally have horizontal grid sizes greater than  $\mathcal{O}(10^3)$  m and vertical resolutions  $\mathcal{O}(10^1)$  m and cannot resolve ice base variability at the required scales, nor do commonly-used bathymetry and ice base forcing products (Morlighem et al., 2020). Therefore, although there are known regions of significant ice shelf base variability and high slopes (Washam et al., 2023; Schmidt et al., 2023; Wåhlin et al., 2024), much of Antarctic ice shelves are represented in ocean models as weakly sloped ( $< 1^{\circ}$ ) from the horizontal. Quantifying the effect of small-scale ice shelf base variation on large-scale melt and optimising their representation in ocean model melt rate parameterisations requires ongoing observational and modelling work.

2. Need to define the regions of the boundary layer in the Introduction, as there are parts of the Methodology that are unclear to the reader as to which part of the boundary layer the authors are referring to.

We have added a paragraph in the Introduction defining the boundary layer sublayers:

L39-49: The ice shelf-ocean boundary layer is typically defined as the boundary layer formed by friction of a mean ocean flow against the ice shelf. Within this layer, there is a viscous sublayer closest to the ice, which is order mm thick and where flow is laminar (Pope, 2001). Further away from the ice, a "log" sublayer forms within which turbulence is affected by the wall boundary, and velocities scale logarithmically with distance from the ice (Pope, 2001; McPhee, 2008). Outside of this surface sublayer is the turbulent outer sublayer. The ice shelf-ocean boundary layer is affected by Earth's rotation, which sets the boundary layer depth (McPhee, 2008; Jenkins, 2016). Multiple physical processes contribute to melting beneath ice shelves in the ice shelf-ocean boundary layer. These include the molecular diffusion of heat and salt, turbulence generated by ocean currents interacting with the ice, and convective flows driven by buoyant meltwater (Malyarenko et al., 2020; Jenkins, 2021; Rosevear et al., 2024). Various parameterisations (e.g. McPhee et al., 1987; Hellmer and Olbers, 1989; Holland and Jenkins, 1999; Kerr and McConnochie, 2015; McConnochie and Kerr, 2017; Schulz et al., 2022; Zhao et al., 2024) exist to account for these processes where they cannot be resolved.

3. Methodology says very little about salinity differences, which are the key driver of stratification in the boundary layer in a warm cavity ice shelf.

We have added a line in the methods to explicitly indicate the importance of salinity in ocean stratification in Section 2.2:

L210-211: The stratification within Antarctic ice shelf cavities is dominated by salinity variation. Meltwater, which is relatively fresh and therefore buoyant, tends to stratify the water column.

4. Introduction should state the velocity ranges considered here before discussing friction velocity.

The friction velocity was not mentioned in the Introduction, but we presume the reviewer meant Section 2.1. We have added a line indicating these velocity ranges:

L166-167: Typical far-field velocities in ice shelves are 0.01 to  $0.1 \,\mathrm{m\,s^{-1}}$  (Table B1), corresponding to friction velocities of  $10^{-4}$  to  $10^{-3} \,\mathrm{m\,s^{-1}}$ .

5. Consider boiling down the parameterization results to Stanton Numbers and constant Gammas, so that larger-scale models and observations that do not resolve all the way up to the ice base can implement these results into something easily useable.

Most ocean models separate transfer coefficients and drag coefficients (Asay-Davis et al., 2016), since often the drag coefficient also acts in the momentum equation. Since the transfer and drag coefficients are multiplied together in the melt parameterisation, the choice to only vary the transfer coefficient is a choice built on the results of Rosevear et al. (2022b) as discussed in Section 2.3. The intention of the parameterisation was to have varying transfer coefficients, and therefore varying Stanton numbers. In case it was confusing, we have removed a sentence that mentioned the Stanton number as a possible alternative parameterisation (since choosing to modify the transfer or drag coefficients is equivalent; what we meant was that the drag coefficient could also be modified independently or at the same time) and have instead defined the Stanton number in a more helpful place when the three-equation parameterisation is introduced.

L178-179: Note that the thermal  $(\Gamma_T C_d^{1/2})$  and haline  $(\Gamma_S C_d^{1/2})$  Stanton numbers are often used to describe the combined effect of the transfer and drag coefficients.

- 6. Authors generally do not seek to ground their modeling in observations of the highly varied and sloping bases of Antarctica's ice shelves.
  - We agree with your observation and have included more mentions of this variation in our revised manuscript, including in places you have suggested (thank you!). In particular, we highlight that our models are limited by their relatively coarse horizontal and vertical resolution, so the different scales that the ocean model simulates versus the scale of the small-scale processes and features (e.g. turbulence driving melt, terraces and scalloping) are distinct, presenting a challenge for the accurate modelling of these processes in large-scale models. We have added a paragraph in the introduction to highlight this difference in spatial scales (see response to comment #1).
- 7. I think the authors have a real opportunity here to implement the StratFeedback+MK18 parameterization in the MITGCM Pine Island Glacier model run to take into account the external shear-driven turbulence, near-ice stratification, and the destabilizing effect rising meltwater on sloping ice bases. This will really round out the study and should test the influence of external turbulence plus localized rising plumes in a significant manner. We did implement the StratFeedback+MK18 parameterisation in the MITgcm Pine Island
  - We did implement the StratFeedback+MK18 parameterisation in the MITgcm Pine Island Glacier simulation, which was briefly mentioned in our original manuscript but not included in our Figures. Based on your suggestion, and that of Reviewers 2 and 3, we have rerun the models but tuned them to an observation-derived baseline melt rate and compared their spatial distributions. The revised Fig. 9 and new Fig. D1 describe this, and we expand our discussion of the StratFeedback+MK18 parameterisation in Sect. 4.4. This new section is attached to the end of this response.
- 8. Generally, the Results Sections are hard to interpret, because it is hard for the reader to disentangle the details of which Methodology was used for each section.

Thank you for bringing this to our attention. We have tried to streamline our results section

by only showing temperature, salinity transects, melt rates and overturning streamfunctions for one model (MOM6) with the other in the Appendix, therefore decreasing the number of main text figures. We have also rearranged the results so that the first section only describes the ISOMIP+ results with prescribed tidal velocity, section 2 explores the three low-velocity limits without tides, section 3 looks at the effect of explicitly simulating tides, and section 4 is for the Pine Island Glacier analysis. We have also added signposting phrases to better indicate this division. We hope these edits improve the readability and clarity of the results section.

**Abstract:**

This is a long abstract that Microsoft Word registers as 257 words. Please double check that this fits within the journal's word count limit. We have revised and shortened the abstract.

Li 7: "and diffusive convection plays a role..." This is an unfinished statement. Please rewrite to finish this thought.

As this study focuses on improving parameterisations of the stratified regime of ice shelf melt rather than diffusive convection, we have elected to remove the mention of diffusive convection in the abstract.

Li 11 - 15: This section focuses on the suppression of melt by stratification, but does not mention the effects of diffusive convection, which is mentioned previously. Rewrite this section to add some discussion of diffusive convection.

See above response: no longer relevant.

**Introduction:**

Li 23: It would be helpful to add a more recent citation here on recent acceleration in melting; the latest one was over a decade ago in 2014.

We have now added citations to Paolo et al. (2015) and Rignot et al. (2019).

Li 32: While not from Antarctica, this study is very applicable to stratified ocean-driven melt of ice shelves and could be added to this list: (Washam et al., 2020). It also has to do with buoyancy-driven circulation and melt, so probably fits into the introduction well.

Washam, P., Nicholls, K. W., Mu"nchow, A., & Padman, L. (2020). Tidal modulation of buoyant flow and basal melt beneath Petermann Gletscher Ice Shelf, Greenland. Journal of Geophysical Research: Oceans, 125(10), e2020JC016427.

Thank you for pointing out the relevance of this study. We have added it.

Li 39: I suggest to change to "the variable molecular diffusion of heat and salt"

Thanks, done.

Li 46: Should this be "turbulent mixing and heat and salt transport"

Thanks, done.

Li 50 - 54: I think somewhere in this part of the introduction or before it should be mentioned that "buoyancy-driven convection" only enhances melt along sloping ice bases, and that the growing number of observations from beneath ice shelves show that their bases' are quite rough with many slopes.

Thank you, we have added this.

L60-62: The parameterisation also does not account for buoyancy-driven convection that may enhance melt along sloped ice bases (e.g. McConnochie and Kerr, 2017), with significant ice base slopes recently observed beneath Antarctic ice shelves (e.g. Washam et al., 2023; Wåhlin et al., 2024), or the effect of diffusive convection....

Li 70: Please add one or both of the following citation to the statement that variation in melt rate within each ice shelf is significant: (Vaňková and Nicholls, 2022; Vaňková et al., 2023).

Vaňková, I., & Nicholls, K. W. (2022). Ocean variability beneath the Filchner-Ronne ice shelf inferred from basal melt rate time series. Journal of Geophysical Research: Oceans, 127(10), e2022JC018879.

Vaňková, I., Winberry, J. P., Cook, S., Nicholls, K. W., Greene, C. A., & Galton-Fenzi, B. K. (2023). High spatial melt rate variability near the Totten Glacier grounding zone explained by new bathymetry inversion. Geophysical Research Letters, 50(10), e2023GL102960.

Added, thank you for the suggestion.

Li 87: This statement is way oversimplified and should be removed: "whereas Antarctic ice shelves are generally weakly sloped  $(<1^{\circ})$ ". There are a growing number of observations that show this is not the case over many scales, from scalloped morphology (<1 m) to a several km grounding zone.

We have removed the statement. Our intention with this statement was in reference to the large-scale slopes resolved by an ocean model (e.g. from BedMachine v3 data, which has a resolution of 500 m). As mentioned in response to major point 6, we have expanded our introduction to discuss ice base slopes in Antarctica, including small-scale, high-slope features as seen in recent work and their distinction from large-scale, low-slope features resolved by ocean models. We have also rephrased the statement in question to refer specifically to the Ross Ice Shelf borehole observation used in the Malyarenko et al. 2020 analysis as

L94-98: Some of these parameterisations match well with *in situ* observations, such as the Kerr and McConnochie (2015) parameterisation which captures convective melt rates at vertical ice faces in Greenland (Schulz et al., 2022; Zhao et al., 2024) and beneath Ross Ice Shelf (Malyarenko et al., 2020). The latter is notable since the Kerr and McConnochie (2015) laboratory study uses vertical ice faces whereas the studied region of the Ross Ice Shelf is weakly sloped (< 1°) from the horizontal (Stewart, 2018; Malyarenko et al., 2020).

**Melt Parameterisation Design and Validation:**

**The Three-Equation Melt Paramterisation and Transfer Coefficients**

Li 127: I would say the ice-ocean boundary temperature is also a key unknown, but if you solve the three equations Tb and Sb will drop out through the quadratic expression. Perhaps you can add some discussion on this to this section, which is typically glossed over in papers. This could also be added to the Li 132.

Sorry for the confusion, we meant that the transfer coefficient is a key unconstrained parameter, not an unknown of the system of equations. We have replaced this phrase with

L155: "The key unconstrained parameter here that must be chosen according to empirical values or theory is the transfer velocity for heat..."

and added to our explanation for the unknowns in the system of equations:

L160-162: "These three equations (1-3) are solved to obtain the three unknowns; the salinity  $S_b$

and temperature  $T_b$  at the ice-ocean interface, and the melt rate m. If the transfer coefficient is independent of these three unknowns, this system of equations reduces to a quadratic equation."

Li 140: Also an opportunity to cite the Washam et al. (2020) paper.

We have added this citation as well as a few others.

L169-173: The values of these transfer coefficients are not well known: they can be tuned to observed estimates, as Jenkins et al. (2010) (hereafter J10) did at the Filchner-Ronne ice shelf using co-located borehole ocean measurements and radar-derived melt rates (and others, e.g. Davis and Nicholls, 2019; Washam et al., 2020; Rosevear et al., 2022a; Davis et al., 2023, have done elsewhere) or tuned in an ocean model to give a desired melt rate (Asay-Davis et al., 2016; Nakayama et al., 2018; Hyogo et al., 2024).

Li 146 – 151: Again, this is focused on flat portions of the ice shelf and requires a qualifying statement that sloped ice can melt faster that the J10 and HJ99-M81 parameterizations, e.g., Schmidt et al., (2023).

**We added**

L187-188: Significant basal slopes can also contribute to deviations from the shear-driven J10 and HJ99-M81 parameterisations (Schmidt et al., 2023).

Li 148: I think it would be helpful to add the M81 stability parameter as an equation, so that the reader can be compare it with (5) in the following discussions. I do see it later in the Appendix, but it may help to have it in the main body or at least reference that it is in the Appendix.

We decided to simply reference that it is in the Appendix to avoid duplicate definitions of variables.

L177: see Eqn. A5 for the stability parameter definition

Li 151 - 160: Please include a discussion of Washam et al., (2023) and Lawrence et al., (2023) in this section on drag coefficients, as both studies quantified ice shelf morphology and related them to a  $u^*$  and CD from observations.

Washam, P., Lawrence, J. D., Stevens, C. L., Hulbe, C. L., Horgan, H. J., Robinson, N. J., ... & Schmidt, B. E. (2023). Direct observations of melting, freezing, and ocean circulation in an ice shelf basal crevasse. Science Advances, 9(43), eadi7638.

Lawrence, J. D., Washam, P. M., Stevens, C., Hulbe, C., Horgan, H. J., Dunbar, G., ... & Schmidt, B. E. (2023). Crevasse refreezing and signatures of retreat observed at Kamb Ice Stream grounding zone. Nature Geoscience, 16(3), 238-243.

**We have added:**

L193–196: Most suggested values range from 0.0015 (Holland and Jenkins, 1999) to 0.0097 (Jenkins et al., 2010), with a value of 0.0022 estimated from turbulence measurements beneath the smooth underside of Larsen C Ice Shelf (Davis and Nicholls, 2019) and 0.0036 estimated from basal ice morphology beneath the crevassed Ross Ice Shelf grounding zone (Lawrence et al., 2023).

**and**

L200-201: Additionally, Washam et al. (2023) find an order of magnitude of spatial variation in drag coefficient within a single ice shelf basal crevasse.

**Stratification Feedback on Turbulence – Insights from Large Eddy Simulations:**

Li 161 – 167: This paragraph never mentions salinity, which is the principle driver of density at the ocean temperatures responsible for melting ice shelves. Please properly attribute stratification to difference in density, driven by salinity changes. Additionally, and potentially more important, this discussion only applies to flat or gently-sloping ice where meltwater pools instead of rises vigorously to act as a source of turbulence that destratifies the boundary layer. This must be said in this paragraph also.

As mentioned in major point #3, we have added a sentence highlighting the importance of salinity in stratification. We have also clarified that the stratification effect we discuss applies to flat or weakly sloped ice shelves:

L207–209: The other is the ability of stratification to suppress boundary layer turbulence beneath horizontal or gently sloping ice shelves (noting that the same turbulence feedback may not apply beneath steeply sloped ice bases where meltwater can drive buoyant flow up-slope, generating turbulence).

and emphasise the LES simulations looked at horizontal and weakly sloped ice bases:

L212–213: Vreugdenhil and Taylor (2019) and Rosevear et al. (2022b) use Large Eddy Simulations (LES) beneath horizontal and weakly sloped ice bases to diagnose regimes of Antarctic ice shelf melt.

Li 173: Does a small L+ here refer to an absolute sense or a highly negative value? This is slightly non-intuitive, since there is a negative in front of  $u^*$  in (5), g is a positive 9.80 m/s2, and one might expect Tb – TM to be larger (more negative) than Sb – SM, which would result in a positive buoyancy flux. Please spell this out for the reader, or preferably, move the expressions around in the 3 equation parameterization to place a positive sign in front of the heat/salt flux and make it TM - Tb (SM – Sb), then place a negative sign in front of the heat conduction into the ice shelf.

We focus only on positive values as the destabilising buoyancy flux parameter space has not been tested by the LES simulations we use. We have added a line to clarify this:

L220–221:We focus only on positive values of  $L^+$ , indicating a stabilising (negative sign) buoyancy flux; the LES simulations do not explore freezing and destabilising conditions.

We have also rearranged Eqns. 2 and 3 as suggested.

We don't think it is possible (if heat conduction is neglected) for melting conditions to result in a destabilising flux (i.e.  $\alpha(T_M - T_b) > \beta(S_M - S_b)$ ). Note  $\beta/\alpha \sim 20^{\circ}\text{C/(psu)}$  in ice shelf cavity conditions so a destabilising flux with  $T_M > T_b$  and  $S_M > S_b$  would require  $T_M - T_b > 20(S_M - S_b)^{\circ}\text{C/psu}$ . Rearranging Eqns. 2 and 3,

$$\frac{T_M - T_b}{S_M - S_b} = \frac{\gamma_S L_f}{\gamma_T C_p S_b} \tag{1}$$

and with  $\gamma_S/\gamma_T \sim 1/35$ ,  $L_f/C_p \sim 80$  °C and  $S_b \sim 30$ psu this leaves the RHS approximately  $\frac{1}{15}$  °C/psu, clearly not large enough to cause a destabilising flux. We have highlighted the importance of the salinity in the buoyancy flux here:

L222-223: Note  $\beta/\alpha \sim 20\,^{\circ}\text{C/psu}$  in ice shelf cavity conditions (Asay-Davis et al., 2016) so salinity changes control the buoyancy flux.

Li 173: Is this kinematic viscosity or eddy viscosity?

It is molecular viscosity. We have made this explicit.

Li 189 – 190: I do not understand how the cooling effect of melting is not accounted for in L+, since the buoyancy term (Bb) should exhibit some change in TM and SM as the boundary layer cools and freshens from melting. Unless TM and SM are chosen at a sufficient distance to be truly "far-field." To be honest, I have a hard time understanding where TM and SM are chosen in most papers. Please elucidate this more clearly in this section.

In ocean models, far-field temperatures and salinities,  $T_M$  and  $S_M$ , are typically averaged over the mixed layer or top-most grid cell, order 10 m or larger (Asay-Davis et al., 2016; Gwyther et al., 2020). In the borehole observations tested in Fig.3, generally far-field temperatures are also taken  $\sim 10$  m from the ice. Thus, far-field temperatures may not generally capture the boundary layer cooling. We have added some more words to clarify this:

L205–207: Stratification due to buoyant meltwater has two distinct effects on the melt rate. One is the effect of meltwater to cool and freshen the surface boundary layer, which decreases the relevance of the far-field temperature that parameterisations generally consider (Rosevear et al., 2022b) as a heat source for melting.

L237–239: In contrast, the Rosevear et al. (2022b) stratified regime definition includes the effect of stratified meltwater to cool the boundary layer relative to the far-field temperature as mentioned earlier, which is not captured by  $L^+$ .

**Stratification Feedback Parameterisation Design:**

Li 194: Add "along flat or gently-sloping ice to the end of this sentence."

**Done, thank you.**

Li 205-207: Briefly comment on how neglecting the heat conductive flux will influence results, citing literature that has considered this variable, e.g., Arzeno et al. (2014), Washam et al. (2020), and others. Note that importantly at low ocean heat fluxes, the conductive heat flux through the ice shelf can be nearly as high as the ocean heat flux, which plays an important role in transitioning from melting to freezing.

Arzeno, I. B., Beardsley, R. C., Limeburner, R., Owens, B., Padman, L., Springer, S. R., ... & Williams, M. J. (2014). Ocean variability contributing to basal melt rate near the ice front of Ross Ice Shelf, Antarctica. Journal of Geophysical Research: Oceans, 119(7), 4214-4233.

Thank you for pointing this out. We ran brief tests with the conductive flux (the recommended linearised advection-diffusion form of Holland and Jenkins (1999)) turned on in MOM6, and mean melt rates were less than 10% smaller (Fig. R1), so we do not expect the results to vary qualitatively with conductive heat flux included. Holland and Jenkins (1999) also suggest a similar 10% change.

**We have added the following comment:**

L257–261: Note that we also neglect the conductive heat flux term of Eqn. 2; although the conductive heat flux may be an important term in some ice shelf cavity conditions (Holland and Jenkins, 1999; Arzeno et al., 2014; Washam et al., 2020; Wiskandt and Jourdain, in review), melt rates are not expected to decrease by more than 10% (Holland and Jenkins, 1999). Thus, we do not expect qualitatively different conclusions when we omit the conductive heat flux term.

Li 211 – 212: The velocity in the boundary layer should be less than far-field, because the

Figure R1: Effect of heat conductive flux in MOM6 ISOMIP+ with insulating ice (no heat flow into ice,  $Q_I^T = 0$ ) in blue and conducting ice (with the linearised advection-diffusion form of Holland and Jenkins (1999) (option C of Wiskandt and Jourdain, in review) in orange. Left panel shows the warm experiment and right the cold. The StratFeedback parameterisation is used with ISOMIP+ protocol of prescribed tidal velocity (i.e.  $U_{tide} = 0.01 \,\mathrm{m\,s^{-1}}$ ).

oceanic flow is starting to feel the friction of the ice base, which generates turbulent eddies that mix heat and salt towards the ice. There are also multiple regions of the boundary layer, such as the outer, surface, and viscous sublayer, that have not been properly defined at this point (see general comment). All of this needs to be spelled out clearly to the reader, and sentences like this are presently confusing, because the boundary layer has not been properly introduced.

As discussed in response to major comment #2, we have added the boundary layer definitions to the Introduction. We also modified this sentence, also in response to reviewer 3:

L267: Monin-Obukhov theory expects that under a stabilising buoyancy flux, the drag coefficient is also reduced as the friction velocity is suppressed relative to a fixed far-field velocity (the drag coefficient is defined as the ratio of these speeds).

Li 216 – 217: If the whole domain is laminar, then there is no turbulence, right? Then if the drag coefficient is related to the friction velocity and therefore turbulence, how can there be a drag coefficient if there is no turbulence? I realize that this is a literature review, but I think this should be presented more clearly to the reader.

Your comment highlighted that the drag coefficient is not particularly meaningful for the intermittently laminar case in Vreugdenhil and Taylor (2019). We have simplified and clarified the text which justifies our use of a constant Cd on the basis of the results of Rosevear et al. (2022b).

L266–274: We could also consider an alternative parameterisation where the drag coefficient, as well as the transfer coefficients, is varied. Monin-Obukhov theory expects that under a stabilising buoyancy flux, the drag coefficient is also reduced as the friction velocity is suppressed relative to a fixed far-field velocity (the drag coefficient is defined as the ratio of these speeds). Indeed, Vreugdenhil and Taylor (2019) see a reduction in the drag coefficient in LES experiments with smaller  $L^+$ . However, Rosevear et al. (2022b) do not see a systematic variation in drag coefficient with  $L^+$  (Fig. 1d). The difference in the behaviour of the drag coefficients between the LES studies, which otherwise agree strongly, is likely due to the different methods of forcing the current beneath the ice. We assume the approach of Rosevear et al. (2022b), which involves forcing the model domain with a steady, far-field flow in geostrophic balance and allowing an

Ekman boundary layer to form, to be somewhat more realistic. We therefore choose to follow the data of Rosevear et al. (2022b) in Fig. 1d, and keep the drag coefficient constant in our study. Note that changing  $C_d$  would also change the surface boundary drag law parameterisation in some models.

Li 221: Please change to vary the thermal and haline Stanton numbers  $(\Gamma_T((C_D)^{1/2}))$  and  $\Gamma_T((C_D)^{1/2})$

**Done.**

Li 223 – 228: This sentence suggests that the authors are solving the two-equation parametrization, where there is a salinity difference across the boundary layer of 0. I don't think this is actually the case, but it would be helpful for the authors to clarify this in the text.

Thank you for picking up this confusion. The thermal driving metric,  $T_M - T_{fr}(S_M)$  used in the plots differs from the thermal driving term in the three-equation parameterisation  $T_M - T_{fr}(S_b)$ . Though we use the latter in the calculation of melt rates, we chose to plot the former since it is easier to calculate, particularly for observations, and it is independent of transfer coefficient so allows for easier comparison of parameterisations.  $T_M - T_{fr}(S_M)$  quantifies the maximum amount of heat available rather than the actual heat delivered due to changes in the freezing point with salinity. We have made this clear in the text.

L281–284: Note this thermal driving may be smaller than the actual temperature difference delivering heat from the ocean for melting that we computed using the three-equation parameterisation  $(T_M - T_b \text{ in Eqn. 2})$ , but is independent of transfer coefficient choices and therefore more appropriate when comparing parameterisations.

Li 223 - 239: I believe that it is roughly an order of magnitude greater, but it would be helpful to just state the actual far-field velocity range that could produce u\* values of 0 - 0.010 m/s.

**We added**

L293: At a thermal driving of  $T^* = 2^{\circ}$ C and  $u_* = 0.001 \text{ m s}^{-1}$  (far-field velocities of  $\sim 2 \text{ cm/s}$ , on the lower end of observed speeds; Table B1)

**Comparison to Observations:**

Li 241 – 260: See comment on Figure 3 below also. Please add Ross Ice Shelf data from Washam et al. (2023) to these plots. While not Antarctica, it may also be helpful to place the detailed data from Petermann Glacier into this discussion.

Washam, P., Lawrence, J. D., Stevens, C. L., Hulbe, C. L., Horgan, H. J., Robinson, N. J., ... & Schmidt, B. E. (2023). Direct observations of melting, freezing, and ocean circulation in an ice shelf basal crevasse. Science Advances, 9(43), eadi7638.

Washam, P., Nicholls, K. W., Mu"nchow, A., & Padman, L. (2020). Tidal modulation of buoyant flow and basal melt beneath Petermann Gletscher Ice Shelf, Greenland. Journal of Geophysical Research: Oceans, 125(10), e2020JC016427.

Thank you for the suggestions. We have looked through these papers but feel it would be inconsistent to add them to Figure 3. Our criteria for including studies in this figure was that studies should have a co-located radar-derived melt rate and profile of ocean conditions, including measured far-field temperature, salinity and velocity so that they can be compared in Fig.3a. To the best of our knowledge, Washam et al. (2020) does not directly measure velocities, which are instead inferred from a tuning between meltwater pulse time-lags between different

sites on the same glacier and the use of a melt parameterisation. Whilst Washam et al. (2023) does directly measure velocities, we could not find co-located radar measurements of melt rate. Washam et al. (2023) could technically be added to panel b of the figure. However, our intent in this figure is to show the regimes associated with the comparison of co-located radar-derived and parameterisation-derived melt rates, rather than provide a complete overview of the observed ice shelf-ocean boundary layer regimes around Antarctica and Greenland (which would add many other studies and is a great suggestion for future work!) so we would prefer to retain the figure as is. We have ensured these studies are mentioned elsewhere in the manuscript. To clarify our criteria, we added:

L298-299: Following Rosevear et al. (2022a), we compare the melt rate produced by the Strat-Feedback and ConstCoeff melt parameterisations using limited direct observations of borehole ocean conditions and co-located, direct melt rate measurements in Antarctic ice shelves.

Li 248: I would be careful to say that ignoring heat conduction makes no difference on melt rate parameterization at low ocean heat and salt fluxes, i.e., cold cavity conditions.

We have modified our statement to weaken it (and to emphasise that we don't expect the fact that the overestimation will change, even if the melt rates themselves do change), see also Figure R1 and response to L205-207 demonstrating a minor change in melt rate.

L303–305: recall we ignore heat conduction into the ice, but these choices are not expected to qualitatively change the overestimation of melt rates.

**Limiting to a Velocity-Independent Parameterisation:**

Li 266: Following the comment above, it would be helpful to define what velocity range constitutes "low-velocity."

We have defined it to be far-field flows of 1 cm/s or smaller, since this is the scale of the ISOMIP+ prescribed tidal velocity:

L323–325: There is both a numerical and physical reason for the low-velocity ice shelf cavity regime to be specially treated with the three-equation parameterisation (where this regime is characterised by low velocities, here taken as far-field flows of  $1\,\mathrm{cm\,s^{-1}}$  or smaller, but has considerable overlap with the  $L^+ < 2500$  diffusive-convective regime).

Li 274 - 275: Thank you for identifying the varying ocean-forced mechanism that relate to differing ice slopes.

**Thanks.**

Li 299: turbulent or molecular diffusion? If molecular, then the sublayer thickness will need to be accounted for and the temperature and salinity gradient across it. If turbulent, please explain this further. Is it just a really low U multiplied by  $\operatorname{sqrt}(C_D)$ ?

Molecular diffusion. However, we cannot resolve these processes in an ocean model, so this choice is a simplification used as a somewhat arbitrary parameter in some ocean models (e.g. Gwyther et al., 2016). We have added a clarification:

L363–365: where the minimum velocity is intended to represent heat transport occurring through molecular diffusion even at very low current speeds (Gwyther et al., 2016). Using a minimum velocity is a simplification for models that do not resolve the boundary layer, and this velocity does not account for the true viscous sublayer thermodynamics.

Li 266 – 306: I see no discussion of a combined StratFeedback+MK18 in this section, but (I think) this is most likely what happens in the real world along sloping ice shelf bases and I

see it in Figure 2. I suggest to add a few sentences that discuss this parameterization to this section.

Thanks for the suggestion. We added some sentences.

L372–377: Each choice of transfer and drag coefficient can be combined with each choice of low-velocity limit. In particular, the StratFeedback+MK18 parameterisation is intended to best represent real ice shelf–ocean regimes, since it encompasses the commonly used shear-driven melt parameterisation in well-mixed, shear-driven conditions, the stratified suppression of turbulence observed and suggested by LES simulations, as well as a lab-based velocity-independent convective parameterisation when far-field flows are weak. We also assess the sensitivity to the choice of low-velocity regimes with a fixed transfer coefficient parameterisation choice.

**Model Configurations**

**ISOMIP+** Setup and Modifications**

Li 324 – 325: I do not think that the cold cavity setup is restored to a very realistic T/S profile. It is fine to take a full isothermal temperature profile to represent deep convection, but if that is the case, then the salinity profile should also be nearly uniform. In any case, the choice of a surface salinity of 33.8 g/kg seems too fresh if all the freshening is to be accounted from ice melt. Take a look at hydrographic sections in front of the Ross and Filchner-Ronne and adjust accordingly. It doesn't have to be perfect, but I suggest a lower salinity range.

This is a good point and we agree that the cavity conditions in the cold ice shelf is not very realistic. We chose to use the ISOMIP+ protocol conditions mainly because it is a standard test case and therefore comparable to other models and studies, e.g. Gwyther et al. (2020). We also expect to make the same conclusions even with a more realistic cold cavity stratification profile, namely that the parameterisation is regime dependent and that cold ice shelves are in a higher  $L^+$  regime than warm ones. For simplicity, we would like to keep the hydrography as is, but have added a caveat mentioning the unrealistic profile.

L397-399: Note these temperature and salinity profiles are highly idealised, and the cold configuration is unrealistically fresh compared to conditions within and outside the Ross and Weddell Sea ice shelves (Orlanski, 1976; Nicholls et al., 2004; Darelius et al., 2014).

Li 329: Please consider adding in the heat conduction term to these models, as it will become important in the cold cavity, low heat/salt flux scenarios, especially when the ice draft is thin. If this is not possible, remark on this as a weakness of this experiment and discuss the caveats.

It would be very time-consuming for us to run all of the configurations (44 of them) again with the heat conduction term, therefore we wish to keep the simulations as is assuming an insulating ice shelf. Additionally, the lack of heat conduction term matches the ISOMIP+ protocol, providing better comparison to other models and studies (e.g. Gwyther et al., 2020). As mentioned in response to the Li 205-207 comment, we have run the models briefly with a conduction term and saw only small (< 10%) differences. We have added a caveat in the text.

L403–404: To simulate basal melt, we use the three-equation parameterisation (Eqns. 2-4) without the ice heat conduction term (noting melt rates would decrease by  $\sim 10\%$  if this term were to be included).

**Idealised MOM6 Configuration**

LI 335 – 349: Please explicitly mention the uppermost layer vertical grid cell size range here and remark on how well it represents observations of the boundary layer beneath ice shelves.

These models do not resolve the ice—ocean boundary layer structure, generally just having one or two cold layers near the surface.

L425-426: This vertical resolution is insufficient to resolve the structure of the ice shelf—ocean boundary layer, though the uppermost layers exhibit cooling and freshening in response to melting.

**Idealised MITgcm Configuration**

Li 351 - 3558: Similarly, remark on whether the 5 m partial grid cell can adequately resolve the boundary layer.

Similarly to above.

L436–437: As in MOM6, this vertical resolution is insufficient to resolve the structure of the ice shelf–ocean boundary layer.

**Idealised Explicit Tidal Forcing:**

Li 360 – 372: Do these tides pass the critical M2 latitude in your simulations? Is this included?

Thank you for picking up our omission. The idealised ISOMIP+ simulations are on an f-plane at latitude 75°S, therefore south of the M2 critical latitude. We have added a clarification of the f-plane to the methods:

L400–402: Unless specified, we follow the mixing, viscosity and equation of state protocols of ISOMIP+ (Asay-Davis et al., 2016). This protocol includes the f-plane approximation with a latitude of 75°S.

We have also commented on the critical latitude when the tidal forcing is introduced:

L445-447: Since the ISOMIP+ simulations are on an f-plane with latitude 75°S (Asay-Davis et al., 2016), the whole domain is effectively south of the M2 critical latitude (Makinson et al., 2006).

Li 370 – 372: Do these decreased tidal velocities represent observations, e.g., Jenkins et al. (2010) or Davis & Nicholls (2019)? I don't think so. Please remark on why this is the case.

Our result of weaker tidal velocities at depth beneath the ice is supported by several modelling studies: Mueller et al. (2012), Gwyther et al. (2016) and Jourdain et al. (2019) all see weaker tidal velocities near the deep grounding lines of ice shelves. To the best of our knowledge, the observational papers by Jenkins et al. (2010) and Davis & Nicholls (2019) don't explore spatial distributions of tidal velocities along the ice draft. We have added references to the modelling papers:

L450-452: However, the resulting tidal velocity at the ice-ocean interface is lower near the grounding line compared to the ice front (Fig. S1), as seen in other modelling studies (Mueller et al., 2012; Gwyther et al., 2016; Jourdain et al., 2019)

**Pine Island Glacier Configuration:**

Li 388 – Li 390: I highly recommend adding in the StratFeedback+MK18 parameterization to this study, as you are now dealing with a (somewhat) realistic model configuration that will experience external shear-driven turbulence, stratification, and rising meltwater plumes. This would really take this study over the top!

Thank you for the suggestion. The StratFeedback+MK18 parameterisation was included in the original manuscript but not highlighted. We have now devoted more time to it and included it in the revised Fig. 9 and Fig. D1, and analysed its spatial distribution compared to other parameterisations when tuned to satellite melt rates. The revised section is copied at the end of the document.

**Results:**

**Idealised ISOMIP+ Results:**

Li 396 – 398: Add a similar discussion on salinity differences, which are the primary driver of stratification (last time I say this), and see comment below on Figure 4.

Thank you, we have included salinity profiles in the figure as suggested and written:

L486–488: The salinity stratification, which dominates the density, is similar between warm and cold experiments with a fresh meltwater layer most prominent in Fig. 4e, though the warm experiment has saltier deep water following the ISOMIP+ protocol.

Li 408 – 414: Perhaps this can wait until the Discussion Section, but I think L+ could be artificially low in these simulations, because of the course grid size. Consider adding a section that explicitly discusses how vertical resolution affects L+ and compare it to either high resolution model runs or observations to properly ground these results.

We agree and have expanded our existing discussion in the Discussion section:

L486–493: Our results demonstrate the importance of testing basal melt parameterisations across various ice shelf cavity regimes. The basal melt-ocean circulation positive feedback makes idealised models extremely sensitive to specific choices in the parameterisation, possibly more so than realistic models. Still, achieving the expected  $L^+$  suggested by in situ borehole observations, where many locations had conditions with  $L^+ > 10^4$  (Fig. 3b), required large drag coefficients in the Pine Island Glacier experiment. Ocean models may not simulate true ice shelf melt regimes since they lack the small-scale flow variability observed at high frequencies beneath ice shelves, either through not resolving these scales of motion (through both horizontal and vertical resolution), or may have anomalously smooth bathymetry and ice base shape.

L696–698: This compensation, however, would be expected to depend on resolution; future research should investigate the model grid resolution dependence of simulated ice shelf regimes.

Li 429: Suggest to not cite manuscripts in preparation.

Yes, this was just a placeholder. We have removed the citation but will add it back if submitted before the present manuscript is accepted.

**Sensitivity to the Low-Velocity Limit:**

431: Wait, were tides included in all of the results from the previous section? If so, it was not clear to the reader, so please restate it in that section. Later in Li 439 I see that it is fully thermohaline. Please make this clearer.

They were not included. We have added some lines to signpost this.

**Sec 4.1 first paragraph**

L380–482: In this first section, we follow the ISOMIP+ protocol and use the low-velocity limit with a prescribed tidal contribution of  $U_t = 0.01 \,\mathrm{m\,s^{-1}}$  to the friction velocity (Table 2). Here, the simulations do not include explicit tides.

**Sec 4.2 first paragraph**

L522: In this section, we explore this sensitivity, noting as in Sect. 4.1 the simulations analysed do not include explicit tides.

**Energetic Ice Shelf Cavity Regimes:**

Li 460 – 491: This section should evolve after the appropriateness of L+ in these simulations has been assessed following the prior comment (Li 408 - 414). Or, this can wait to the discussion section.

We have moved part of this comment to the discussion section, and modified the rest of it here to read:

L582–586: However, the idealised tidal simulations demonstrate the difficulty in achieving realistic ice shelf cavity regimes in idealised models. Even with a large tidal forcing of  $0.2\,\mathrm{m\,s^{-1}}$  amplitude velocity (corresponding to a 6.4 m sea level anomaly forcing in this idealised cavity), the warm cavity is not entirely in the well-mixed regime, possibly associated with the smoothness of the geometry and insufficient spatial resolutions.

**Realistic Pine Island Glacier Simulation:**

Li 493 – 532: I highly suggest to also implement the StratFeedback+MK18 parameterization into this analysis, i.e., a parameterization that includes the influence of ice base slope, stratification, and external turbulence.

Thanks for the suggestion, as mentioned in response to Li 388 comment, we have revised this analysis and section, copied at the end of the document.

Li 521 – 524: If I understand this right, this was hardly a 'tuning' of the drag coefficient and more of picking a single observed drag coefficient and applying it to the whole model. At this point it is quite difficult to follow what method has been used where. Regardless, I would not refer to this as a 'tuning,' but instead an 'altering' of the drag coefficient, then please state explicitly in this section what CD was changed from and to. I also took a look back at the Stanton et al. (2013) paper and don't see a value for CD, but instead only a timeseries of u\* without any mention of U. How as CD computed then? Another approach to this problem would be to use the range of CD values observed beneath ice shelves and force the model with each of them to see how it influences the melt rates.

In the original manuscript,  $C_d$  for the benchmark, HJ99-neutral parameterisation (Nakayama et al., 2021) was taken as 0.0015 as suggested by Holland and Jenkins (1999) and used everywhere in the domain. We did not find (nor use) a  $C_d$  value from Stanton et al. (2013). The constant  $C_d$  was increased for the tuned StratFeedback simulation to 0.0042 to achieve the same mean melt rate as HJ99-neutral with  $C_d$ =0.0015 (average melt rates to within 0.1 m/yr, determined by trial and error with different choices  $C_d$ ).

In our revision, also in response to Reviewers 2 & 3, we have tuned the drag coefficient for three parameterisation options (HJ99-neutral, StratFeedback+ustarmin and StratFeedback+MK18 to achieve the same area-averaged melt rates as Adusumilli et al. (2020) and compared their spatial distributions. The revised section is copied at the end of the document.

**Discussion:**

Li 534 – 572: All of this reads more like a Summary than a Discussion and should be rewritten to provide more helpful analysis of the results.

Figure R2: Large Eddy Simulation data, with Vreugdenhil and Taylor (2019) in the black crosses and Rosevear et al. (2022b) in blue dots, indicating the relationship between transfer coefficients (a)  $\Gamma_T$ , (b)  $\Gamma_S$ , their ratio (c)  $\Gamma_T/\Gamma_S$  and (d) drag coefficient  $C_d$  against viscous Obhukov scale  $L^+$ . The maximum Vreugdenhil and Taylor (2019) (ConstCoeff) values of the transfer coefficients are included (grey lines), which are similar to the Jenkins et al. (2010) values (pink lines). The blue dashed line indicates the choice of fit of transfer coefficients as a function of viscous Obukhov scale for our stratification feedback parameterisation. Drag coefficients from observations of Jenkins et al. (2010), Davis and Nicholls (2019), Lawrence et al. (2023) and Washam et al. (2023) are also included.

We have edited the Discussion to reduce the summary content, but still feel a short summary is helpful.

Li 573 - 585: Ok, this somewhat satisfies my prior comments on the validity of L+ in these simulations, but given (what I think is) the goal of this study to more accurately parameterize melt rates, I think this section should be expanded to include model-obs comparisons or coarse-fine model comparisons.

Thank you for the suggestion. We feel that it is beyond the scope of this already lengthy study to investigate the resolution-dependence of the models, so we have added a prompt for future research:

L697: Future research should investigate the model grid resolution dependence of simulated ice shelf regimes.

**Figures:**

Figure 1d: Consider adding lines for CD from more observations beneath ice shelves, such as Jenkins et al. (2010), Davis and Nicholls (2019), Washam et al. (2023), and Lawrence et al. (2023).

Thanks for the suggestion, we have done this. See Fig. R2.

Figure 2b: Consider adding a range of ice base angles (4 or 5 angles between 10° and 90°) to this

figure or to the Appendix, since 10° seems to be somewhat arbitrary and low, without any real acknowledgement of the many small-scale slopes observed beneath ice shelves, e.g., Dutrieux et al. (2014), Schmidt et al. (2023), Lawence et al. (2023), Washam et al. (2023), etc...

Dutrieux, P., Stewart, C., Jenkins, A., Nicholls, K. W., Corr, H. F., Rignot, E., & Steffen, K. (2014). Basal terraces on melting ice shelves. Geophysical Research Letters, 41(15), 5506-5513.

We have added figures with three angles to the Appendix A as suggested, reproduced here in Fig. R3:

Fig. A2 presents alternative angle options to the 10° slope used in Fig. 2b and Fig. A1d.

Figure R3: Thermal driving – friction velocity parameter space diagram indicating melt rates calculated as a function of far-field temperature, salinity and pressure (which are set to S = 34.5 psu and p = 500 dbar) and friction velocity. The melt rates are solved for a variety of parameterisation options: ConstCoeff is in the blue solid lines, and StratFeedback is shown in white dashed lines. Constant melt rates obtained from slope-dependent McConnochie and Kerr (2018) convective parameterisation are in the pink dotted lines, and the combination of the StratFeedback+MK18 limit is in the red dash-dot line. The three panels show different slope angle choices to Fig. 2b, with angles from the horizontal of (a)  $2^{\circ}$ , (b)  $45^{\circ}$  and (c)  $80^{\circ}$ .

Figure 3: Please add Ross Ice Shelf data from Washam et al. (2023) to these plots. While not Antarctica, it may also be helpful to place the detailed data from Petermann Glacier into this figure.

Washam, P., Lawrence, J. D., Stevens, C. L., Hulbe, C. L., Horgan, H. J., Robinson, N. J., ... & Schmidt, B. E. (2023). Direct observations of melting, freezing, and ocean circulation in an ice shelf basal crevasse. Science Advances, 9(43), eadi7638.

Washam, P., Nicholls, K. W., Münchow, A., & Padman, L. (2020). Tidal modulation of buoyant flow and basal melt beneath Petermann Gletscher Ice Shelf, Greenland. Journal of Geophysical Research: Oceans, 125(10), e2020JC016427.

Please see our response to Li 241-260.

Figure 4: I suggest to add two rows that are similar to a-h, but present Salinity.

We have done this, and removed the kinetic energy plots, combining the overturning streamfunction with the other diagnostics. In response to Reviewer 2, we now only show MOM6 results in the main text and have moved the MITgcm plots to the Appendix. The updated MOM6 figures is shown in Fig. R4.

Figure 7: Given that there have been scalebar changes in the comparison of cold and warm cavities in Fig 4 and 6, I suggest to change the vertical axis in panel b and use lower melt rate

Figure R4: Temperature (a-d) and salinity (e-h) transects, melt rate distribution (i-l) and zonally averaged overturning streamfunction in density coordinates (m-p) for MOM6 simulations. All experiments use the ISOMIP+ protocol-specified tidal velocity  $U_t = 0.01 \text{m s}^{-1}$  as the low-velocity limit in the melt rate parameterisation. Variables are averaged over the last 180 days of the simulation, with the temperature and salinity profiles taken at the y=40 km transect. Warm experiments are in columns 1 and 2, cold in 3 and 4. Columns 1 and 3 show the constant coefficient melt parameterisation results, and columns 2 and 4 contain the stratification feedback parameterisation. Melt rates averaged over the ice shelf are listed in panels i-l. Black contours in m-p are spaced by 10 mSv in panels (m-n) and 0.5 mSv in panels (o-p), and the text lists the maximum value of the overturning streamfunction in the domain. Note the different colourbar ranges between the warm and cold simulations. Equivalent results for MITgcm are in the Appendix C.

contour lines to make it more useful. Simply state once again to note the scale change. We have done this (Fig. R5).

Figure R5: Thermal driving – friction velocity regime diagrams for selected MOM6 StratFeedback experiments, indicating the number of grid cells in each regime time-averaged over the final 180 days of the simulation. Panel (a) shows warm experiments and (b) cold. The minimum friction velocity  $1 \times 10^{-4} \,\mathrm{m\,s^{-1}}$  experiments are shown in blue (leftmost vertical line), prescribed tidal velocity  $U_t = 0.01 \,\mathrm{m\,s^{-1}}$  in purple (middle vertical line) and explicit tidal forcing with amplitude  $0.1 \,\mathrm{m\,s^{-1}}$  in orange colours to the right. StratFeedback melt rates are shown in the solid contours and stippling shows where transfer coefficients are reduced from the ConstCoeff values, both calculated assuming a salinity  $S_M = 34.05 \,\mathrm{psu}$  and pressure 300 dbar, which are representative values for the ISOMIP+ cavity. Note the difference in y-axis extent between panels.

**Reviewer 2**

**General comments**

This manuscript is a valuable contribution to the literature. The authors bridge small-scale ocean modeling studies with larger-scale ocean modeling by proposing a new variant on a parameterization of ice-shelf melting. They then assess this parameterization against local observations of ice-shelf melting and remote-sensing-based estimates of ice shelf melting at one ice shelf. They also demonstrate the parameterization's impact on ice-shelf cavity dynamics through cavity-scale ocean modeling in an idealized domain previously used in the literature. I believe that this parameterization has the potential for becoming the state-of-the-art. However, I do think at times the key points get lost in the text, and I offered some suggestions for improving the communication of the author's results. The main scientific deficiencies I see are an insufficient comparison of Pine Island Glacier simulated melt rates compared with remote-sensing products and a lack of clarity on whether the state of scientific knowledge supports the adoption of your MK18 low-velocity limit variant of your parameterization. As a result, I think these qualify as major revisions but I would emphasize that the quality of the science in this manuscript seems to be high.

We thank Dr. Carolyn Begeman for your valuable comments that have helped to improve the presentation and clarity of the manuscript. Thank you also for your support of our science.

**Specific comments**

1. Consider moving Section 2.4 to early on in the Results.

Thank you for the suggestion. We feel that the observation comparison fits better in Section 2 rather than getting lost after the model description section, so propose to keep the order as is.

2. Consider moving either of the MOM6 or MITgcm idealized simulations to a supplement/appendix. Given that they are qualitatively similar, it may be best to try to de-clutter the main text a bit.

Thank you for the suggestion to simplify the main text. We have followed your suggestion and now only show the MOM6 transects and circulation/melt pattern results in the main text (thereby combining Fig 4 and 6 into one, printed in Fig. R6) and MITgcm results are included the appendix. We have kept both model results in the melt rate bar plots of Fig. 5 because there are known, significant differences in ice shelf cavity models (especially ice—ocean boundary layers) between models with different vertical coordinates (e.g. Gwyther et al., 2020) so we feel that demonstrating the similarity in behaviour of melt rates between the models is helpful.

3. L150: It's worth highlighting sooner that the evidence suggests a greater response to stratification than any of the current parameterizations feature.

Thank you for the suggestion, we have added:

L184–186: Even though HJ99-M81 is designed to account for stabilisation due to stratification, its effect on melting in the parameterisation is modest (Appendix A, Fig. A1) and does not capture the observed response to stratification (Begeman et al., 2018; Rosevear et al., 2022a; Davis et al., 2023).

4. L152: It's not clear why you are discussing the uncertain drag coefficient. I think it's helpful to give readers a sense of where you are going with all of this. E.g., that you will be proposing a different parameterization for  $\Gamma_T$  and  $\Gamma_S$  and leaving  $c_d$  as a tunable parameter.

Figure R6: Temperature (a-d) and salinity (e-h) transects, melt rate distribution (i-l) and zonally averaged overturning streamfunction in density coordinates (m-p) for MOM6 simulations. All experiments use the ISOMIP+ protocol-specified tidal velocity  $U_t = 0.01 \text{m s}^{-1}$  as the low-velocity limit in the melt rate parameterisation. Variables are averaged over the last 180 days of the simulation, with the temperature and salinity profiles taken at the y=40 km transect. Warm experiments are in columns 1 and 2, cold in 3 and 4. Columns 1 and 3 show the constant coefficient melt parameterisation results, and columns 2 and 4 contain the stratification feedback parameterisation. Melt rates averaged over the ice shelf are listed in panels i-l. Black contours in m-p are spaced by 10 mSv in panels (m-n) and 0.5 mSv in panels (o-p), and the text lists the maximum value of the overturning streamfunction in the domain. Note the different colourbar ranges between the warm and cold simulations. Equivalent results for MITgcm are in the Appendix C.

Thank you for this suggestion that helps to clarify our intention. We have added this in front of the drag coefficient paragraph;

L187–191: Our approach in this study is to develop an alternative parameterisation for the transfer coefficients  $\Gamma_T$  and  $\Gamma_S$  which better represents melting across Antarctic ice shelf regimes, whilst treating the drag coefficient as a tunable constant (see Sect. 2.3 for further details). However, the drag coefficient is also a large factor in the uncertainty of...

5. Since you choose L=2500 as the cutoff for the shear regime, it's not clear why you are computing the best fit line without a floor at this value

If we used only data points in the range  $2500

Figure R7: Parameterised melt against observed melt rate (a), for borehole observational data updated from Rosevear et al. (2022a), with the ConstCoeff parameterisation in circles and Strat-Feedback in triangles. Thermal driving – friction velocity regime (b) updated from Rosevear et al. (2022b), where the thick  $L^+$  line of  $1 \times 10^4$  divides where the StratFeedback parameterisation diverges (to the left) and where the transfer coefficients are constant and equal to ConstCoeff (to the right). The diffusive-convective ( $L^+ < 2500$ ), stratified (2500  $< L^+ < 10^4$ ) and well-mixed, shear-driven ( $L^+ > 10^4$ ) regimes are shaded. Data is obtained from the Filchner-Ronne Ice Shelf (FRIS, Jenkins et al., 2010), Larsen C Ice Shelf (Davis and Nicholls, 2019), Amery Ice Shelf (Rosevear et al., 2022a), Ross Ice Shelf (Ross S for summer and Ross W for winter data, Stewart, 2018), WISSARD Grounding Zone of the Ross Ice Shelf (WGZ (Ross), Begeman et al., 2018), George VI ice shelf (Kimura et al., 2015; Middleton et al., 2022), Pine Island Glacier (Stanton et al., 2013) and Thwaites Glacier (Davis et al., 2023). Further computation details are supplied in Table B1.

9. I didn't love how the axes of Figure 3b didn't match those in Figure 2. It made it difficult to tell where in the T-u\* space we have obs constraints. Consider changing axes or plotting the obs points on one of the figure 2 panels.

Thanks for the suggestion. We changed the axes on Fig. 3b (see Fig. R7).

10. In section 4.4, you first present results without tuning  $C_d$ , which do not match observed melt rates, and then tune  $C_d$  to match observed melt rates. The untuned results are less relevant to how the community would go about regional ice shelf cavity modeling. Thus, I would argue that you should only present the results from the tuned simulations in the main text and move a comparison of the parameterizations with the same drag rate to a supplement or appendix.

Thank you for the suggestion. We have significantly revised Sect 4.4 as suggested, tuning melt rates for HJ99-neutral, StratFeedback and StratFeedback+MK18 parameterisations to Adusumilli et al. (2020) melt rates and then comparing spatial distributions and PDFs of melt. The revised Section 4.4 is provided at the end of the document.

11. One of the main arguments in favor of adopting your proposed parameterization is that it produced a more realistic distribution of melt rates at PIG. Thus, I think it's really important to plot the results from one of the observational products you cite in the paper compared

with your simulated melt rates with and without StratFeedback. (The motivation for this parameterization is also a bit weakened by your statement that Nakayama does a good job at representing melt rates.) RMS would be a relevant metric to compare these simulations, and a figure with a PDF/histogram of melt rates could be instructive if there are significant differences for the same mean melt rate.

Thank you for the suggestion. We have re-run the Pine Island Glacier simulations tuned to Adusumilli et al. (2020) melt rates so that they can be better compared with the satellite patterns. The new section 4.4 is attached at the end of the document, and we made our histogram of melt rates in the Supplementary more prominent by moving it to the appendix. We tuned to Adusumilli et al. (2020) given it is a published and publicly accessible product, but its coarse resolution and missing data in some locations means an RMS comparison with the MITgcm models is not appropriate. Whilst all three HJ99-neutral, StratFeedback and StratFeedback+MK18 parameterisations have similar melt rate patterns, the intensification of melt rates near the grounding line in the StratFeedback and StratFeedback+MK18 cases compared to HJ99-neutral align well with the high-resolution satellite melt rate products of Shean et al. (2019) and Zinck et al. (in review).

12. I think the reader could benefit from a little more discussion of the IOBL in the MITgcm PIG simulation, such as what properties look like over the layers that are sampled for the parameterization for the base case vs. the StratFeedback case. The friction velocity shown in Fig. 10e provides a partial understanding.

Given the 5 m vertical resolution in MITgcm, it is difficult to provide helpful information about the ice—ocean boundary layer. Anomalies of thermal driving relative to HJ99-neutral are provided in Supplementary, but generally correlate with melt rate changes, with reductions in melt rate resulting in warmer, more saline water. Temperature and salinity in two transects are also shown in the Supplement, with changes also dominated by melt rate patterns.

13. It's quite unclear what readers are supposed to take away from your low-velocity limit experiments. You say "We have proposed one option for a transition to a velocity-independent convective parameterisation at low velocities" but this statement is so neutral. It would be more helpful to readers if you explained the pros and cons of this option in the discussion so readers can make an informed choice for themselves. But on the other hand, there appears to be so little that can be used reliably to evaluate the MK18 parameterization that I was left wondering whether it was worth devoting so much space in the paper to it. You seem to be arguing that the ISOMIP+ experiment is not a good one for evaluating the MK18 parameterization. You mention that the PIG MK18 simulation increases melt rates, but it's unclear whether adding MK18 yields an improved melt rate distribution relative to observations. Furthermore, it wasn't clear whether any of the observational estimates shown in Figure 3 could be used to assess MK18. (If so, consider discussing and plotting these symbols as well for those affected.)

Thank you for the comment. We agree we had left out MK18 for much of the conclusions since we didn't feel we could confidently say it had improved the parameterisations relative to satellite observations. In the revised manuscript, we tune the parameterisations to Adusumilli et al. (2020) melt rates and compare them on equal footing, including histograms of melt rates, and analyse the difference. We find there is not a substantial difference between StratFeedback and StratFeedback+MK18 in our Pine Island Glacier simulation, though the MK18 option allows less drastic (perhaps more realistic) changes to the drag coefficient and better agreement with satellite observations of melt near the grounding line. Regarding the recommendations for the use of MK18 in other models, we have added the following to the discussion:

L709–716: We have proposed one option for a transition to a velocity-independent convective

parameterisation at low velocities, where the McConnochie and Kerr (2018) parameterisation increases melt rates near sloped ice bases (and therefore better matches satellite-derived melt rates at the Pine Island Glacier grounding line). However, this parameterisation's applicability to weakly sloped ice bases remains unknown, and the representation of sloping or featured topography depends on the resolution of ice draft products and the resolution at which the model is run. Prescribed tidal velocities and minimum friction velocities are also easily-implemented low-velocity limit options, which could be tuned in realistic experiments. However, both melting and circulation are extremely sensitive to these (relatively unconstrained) parameters. Furthermore, other physical processes such as diffusive convection are currently not included.

14. A point that you make but gets a little lost in the text is that the parameterization is not optimal for DDC regime but yet it likely improves the accuracy of predicted melt rates in that regime by reducing melt rates. If I am understanding this correctly, I would encourage you to emphasize this.

Thank you for the comment, this is indeed what we hoped to communicate. We have added text to emphasise this when explaining the fit choice of the parameterisation:

L251–254: We also include data points in the diffusive-convective regime ( $L^+

Figure R8: Thermal driving – friction velocity parameter space diagram indicating melt rates calculated as a function of far-field temperature, salinity and pressure (which are set to S =34.5 psu and p = 500 dbar) and friction velocity. The melt rates are solved for a variety of parameterisation options in (a) and (b). The orange, dotted lines in panel (a) are the Holland and Jenkins (1999) formulation with the  $\eta_*$  stratification parameter (McPhee, 1981). A constant transfer coefficient formulation (ConstCoeff) is in the blue solid lines (using the maximal values of Vreugdenhil and Taylor (2019), which is similar to Jenkins et al. (2010) in the light yellow solid lines), and the stratification feedback (StratFeedback) parameterisation we develop here is shown in white dashed lines. In panel (b), we add in the constant melt rates obtained from McConnochie and Kerr (2018) with a slope angle of  $\theta = 10^{\circ}$  from the horizontal in the pink dotted line, and the combination of the StratFeedback+MK18 limit in the red dash-dot line. Panel (c) shows the viscous Obukhov scale  $L^+$  derived from the stratification feedback parameterisation, and where the white  $L^+ = 1 \times 10^4$  indicates where the white dashed lines (StratFeedback) and blue line (ConstCoeff) transition from having the same melt rate to the right and different to the left. Panel (d) also shows this in the ratio of the StratFeedback to ConstCoeff melt rates, with stippling indicating where they are equal.

**No longer relevant due to revision of section 4.4.**

Fig A1b Is it difficult to get useful information from HJ99-M81/CC? It is a little weird that all the other panels but this one are ratios with respect to CC.

Note that HJ99-M81/CC is already in panel c. Panel b (HJ99-M81/HJ99-neutral) is provided to show the effect of the  $\eta_*$  parameterisation only, isolating it from the  $u_*$  dependence of HJ99-neutral transfer coefficients.

Fig A1d: I think it would be helpful to have a contour somewhere near the minimum value in panel (d). The light blue values can be hard to make out.

Thanks, we have added this: Fig. R9.

Figure R9: Ratio of melt rate calculated from varying-transfer coefficient methods to the constant coefficient parameterisation used in this study, presented in thermal driving – friction velocity regime space, assuming  $S_M = 34.5$  psu and a pressure of 500 dbar ( $\sim 500 \, \mathrm{m}$  depth). Panel a shows the Holland and Jenkins (1999) parameterisation with the McPhee (1981) stability parameter (Appendix A1) set to 1 (neutral conditions), and panel c shows it with the McPhee (1981) stability parameter varying. The difference between HJ99-M81 and HJ99-neutral is shown in panel c. Panel d shows the combined stratification feedback and McConnochie and Kerr (2018) low-velocity limit with  $\theta = 10^{\circ}$  (Section 2.5, Appendix A3) compared to ConstCoeff. Contours at ratios 0.5 and 0.75 are provided.

**Reviewer 3**

The authors use data from published LES experiments to develop a parameterization to diagnose vertical fluxes of heat and salt through the oceanic boundary layer at an ice shelf base. The improvement they make is to account for the way stratification can cause suppression of the vertical transports, generally in warm and relatively quiescent environments; however, they make the case that some ice shelf cavities with only weak thermal driving are also likely to be affected by stratification in the same way, and simulations of such cavities would also benefit from using the improved parameterization.

They apply the parameterization to two ocean models, run within the MISOMIP+ framework, comparing with other routinely used parameterizations. They also apply one of the models to the cavity beneath Pine Island Glacier, again with their own and one other parameterization.

I am not a modeller, but I find the paper well-written, generally easy to follow, and the results are convincing. It's a relatively simple story, but has some important implications, one of which concerns the utility of the MISOMIP+ setup that they adopted. I would like to see it published, with some minor revisions.

I'm submitting a marked-up pdf. There are several minor suggestions that might help improve the clarity of the text, but the authors should feel free to ignore any or all of them.

There are a small number of more substantive comments in the pdf, which I will repeat here, along with some other general remarks. They mostly don't require revisions, perhaps a little more explanatory text.

We thank the reviewer for your valuable comments that have helped to improve the presentation, clarity and scientific content of the manuscript.

1. The authors extend their parameterization for very low velocities using a formulation based on laboratory and DNS experiments. The authors who developed the lab-based parameterization note that it is not recommended for basal slopes of less than 10 degrees, a caveat also pointed out by the present authors. As the basal slopes that can be resolved by the vast majority of models are at least an order of magnitude smaller, this is clearly an inappropriate extension. I see that the authors want to extend the parameterization somehow, but using one developed for a different regime might not be wise: the fact that it came from a study gives its application the cloak of respectability when it is simply inapplicable. I would prefer to see an arbitrary set of transfer velocities used at low velocities, possibly tuned in some way. I'm sure the authors wouldn't want to make such a change, but I would be interested to hear their response.

Thank you for bringing this issue up. We agree that the convective parameterisation may not be suitable for angles shallower than 2°, but note Mondal et al. (2019) find a similar scaling to McConnochie and Kerr (2018) with a DNS down to angles of 2°. Additionally, Malyarenko et al. (2020) find good agreement with a convective parameterisation (Kerr and McConnochie, 2015, , designed for vertical ice faces) despite the Ross Ice Shelf observations used coming from a region with slope less than 1°. We agree that melt parameterisations in shallow-sloped, non-shear driven melt regimes is a key unknown which we propose should be further explored:

L18–720: For example, conducting similar experiments to Rosevear et al. (2022b) that resolve the boundary layer with shallow ice slopes would fill a currently undersampled regime.

We note that choosing an arbitrary set of transfer velocities as a lower limit may be a solution to what the reviewer may describe as our over-fitting of the parameter space, and could

be a tuning task in future work. We have expanded on our evaluation of the MK18 limit to our parameterisation or other choices in the Discussion, assessing their strengths and weaknesses.

L07–720: There remain questions around formulating a regime-aware, physically accurate basal melt parameterisation. Future work should further explore the transition between the shearcontrolled stratified regime and the transient diffusive-convective regime (e.g. Rosevear et al., 2022b). We have proposed one option for a transition to a velocity-independent convective parameterisation at low velocities, where the McConnochie and Kerr (2018) parameterisation increases melt rates near sloped ice bases (and therefore better matches satellite-derived melt rates at the Pine Island Glacier grounding line). However, this parameterisation's applicability to weak sloped ice bases remains unknown, and the local basal slopes used in the simulations are resolution-dependent and may not capture steep slopes near unresolved ice features, therefore evaluation of MK18 requires further work. Prescribed tidal velocities and minimum friction velocities are also easily implemented low-velocity limit options, which could be tuned in realistic experiments, but there is great sensitivity to relatively unconstrained parameters. Furthermore, other physical processes such as diffusive convection are currently not included. Developing a truly regime-aware parameterisation likely requires further understanding of the physics governing each regime and the transitions, through more high-resolution numerical simulations, laboratory experiments and in situ ice shelf-ocean boundary layer observations. For example, conducting similar experiments to Rosevear et al. (2022b) that resolve the boundary layer with shallow ice slopes would fill a currently undersampled regime.

2. A more general remark about the way the different regimes are discussed, particularly the "velocity-independent" regime described by MK18. My understanding of the way this is discussed is that we can have some sort of regional external forcing (eg a coastal current flowing beneath a small ice shelf); tidal forcing (also largely externally-forced); and then a buoyancy-driven current, where the forcing comes from melting within the cavity. When that melting is occurring locally, that latter case is what seems to be called "velocity-independent". Which means that it's independent of the free stream velocity, the velocity outside the boundary layer. In old parlance, it's what was called a gravity plume. If my understanding is correct, I think the terminology is confusing and I encourage the authors to make the point that the parameterisation is not so much velocity-independent, but rather, it's free-stream velocity independent. If I'm wrong, then please educate me. Am I the only one confused by this? Possibly.

Thank you for picking this up, you are correct. We have explained this in the low-velocity limit section and again reiterated it in the discussion. However, for conciseness, we would like to still refer to it in the rest of the text by the name velocity-independent.

L334–339: To address this limit, we also implement a transition between the shear-driven parameterisation to a free-stream velocity-independent parameterisation based on laboratory studies of sloped ice (McConnochie and Kerr, 2018, hereafter MK18) and similar direct numerical simulations (Mondal et al., 2019). Velocity-independent refers to the velocity of the free stream flow as captured by the ocean model, which does not appear in the convective melt equations. We note, however, that melting of a sloping ice face produces a buoyant plume with its own velocity, which is implicitly included in the convective parameterisation.

3. Mainly in section 2, the authors use a variety of different terms to describe distance from the ice base, particularly with respect to velocity. We had "far field velocity", the velocity in the "uppermost part of the boundary layer", the "upper layer velocity" and the "mixed layer velocity". Although I got the general sense, it would be nice to attempt to standardise, or perhaps define these terms more precisely. Similarly, when describing the way the two models handle the upper part of the water column, a bit more clarity about how the levels for water

speed and temperature are selected would help.

Thank you, we received a similar comment from Reviewer 1 and have added definitions of the ice—ocean boundary sublayer in the Introduction.

L39–49: The ice shelf–ocean boundary layer is typically defined as the boundary layer formed by friction of a mean ocean flow against the ice shelf. Within this layer, there is a viscous sublayer closest to the ice, which is order mm thick and where flow is laminar (Pope, 2001). Further away from the ice, a "log" sublayer forms within which turbulence is affected by the wall boundary, and velocities scale logarithmically with distance from the ice (Pope, 2001; McPhee, 2008). Outside of this surface sublayer is the turbulent outer sublayer. The ice shelf–ocean boundary layer is affected by Earth's rotation, which sets the boundary layer depth (McPhee, 2008; Jenkins, 2016). Multiple physical processes contribute to melting beneath ice shelves in the ice shelf–ocean boundary layer. These include the variable molecular diffusion of heat and salt, turbulence generated by ocean currents interacting with the ice, and convective flows driven by buoyant meltwater (Malyarenko et al., 2020; Jenkins, 2021; Rosevear et al., 2024). Various parameterisations (e.g. McPhee et al., 1987; Hellmer and Olbers, 1989; Holland and Jenkins, 1999; Kerr and McConnochie, 2015; McConnochie and Kerr, 2017; Schulz et al., 2022; Zhao et al., 2024) exist to account for these processes where they cannot be resolved.

We have also removed mentions of mixed layer since the ice shelf ocean boundary layer is not necessarily the same as the mixed layer, except in the discussion of the MOM6 bulk mixed layer module. We also added some text to clarify the sampling of temperature and velocity in these layers:

L413–417: We use a Kraus and Turner (1967) and Niiler and Kraus (1977)–like bulk mixed layer parameterisation for the surface boundary layer (Hallberg, 2003) which energetically constrain the boundary layer depth, and the Jackson et al. (2008) vertical mixing parameterisation with critical Richardson number 0.25.

L420–427: The model samples temperature, salinity and velocity over the bulk mixed layer in the melt parameterisation, then inserts freshwater in the bulk mixed layer as a volume flux (which can later be entrained in the interior ocean layers, Hallberg, 2003). The magnitude of melting is likely to be sensitive to these choices, as well as to the vertical resolution (Gwyther et al., 2020). Melting is set to zero when the ocean column is less than 10 m thick. The friction velocity  $u_*$  is calculated from the velocities in the uppermost model layer (the top half of the bulk mixed layer, approximately 5 m thick). This vertical resolution is insufficient to resolve the structure of the ice shelf–ocean boundary layer, though the uppermost layers exhibit cooling and freshening in response to melting.

L433–437: Tracers and the velocities for the friction velocity and melt parameterisation are sampled over the uppermost 20 m layer (Losch, 2008), which generally covers more than one vertical grid cell. Meltwater is represented as a virtual salt flux rather than a volume flux, distributed over the same 20 m layer. As in MOM6, this vertical resolution is insufficient to resolve the structure of the ice shelf–ocean boundary layer.

4. In the Pine Island Glacier cavity modelling the authors chose to tune Cd to give a good comparison with HJ99-neutral. Why did they not choose one of the satellite-derived melt rates? I guess it depends on the purpose of the experiment, but again, it seems like a free choice, and a result closer to what we think of as reality might have been more useful. Just a question. A sentence in the text to explain the choice would be good.

Thank you for the observation, which combined with the other reviewers suggestions, prompted us to rerun the Pine Island Glacier simulations with a more useful tuning. In the revised

manuscript, we tune melt rates to satellite-derived Adusumilli et al. (2020) results, and provide a better explanation for the choice.

5. (A bit of a detail. In the fit to derive the StratFeedback parameterisation (from Figure 1) why do the authors adopt natural logarithms in the formula? No problem, but it seems to be a free choice, and they are plotted using logs to base 10.)

There was no particular reason for this choice, but we agree that given the plot is in  $\log_{10}$  we should express our parameterisation with the same base. We have computed the equivalent expressions in base 10 and replaced them in the text.

**Specific comments**

Reviewer 3 has provided comments in a PDF of our preprint, and for the sake of time and space do not repeat all of these comments point-by-point. We have implemented all the minor grammatical and phrasing suggestions: thank you for these improvements to the presentation of the manuscript. Below, we respond to the larger comments suggested by Reviewer 3 in the PDF.

L77: I think the authors can say that the idealized simulations and lab studies have demonstrated the possibility of double-diffusive convection, but I disagree that they have demonstrated its presence beneath ice shelves.

We have replaced with "possibility" as suggested.

L157: Please explain what is meant by this (boundary layer flow profile). Is it the vertical profile of water speed?

Yes, vertical profile of water speed. We have replaced "boundary layer flow profile" with "vertical profile of water speed".

L201: "regime transitions of heat and salt transport may occur at different L+" That's not completely obvious to me. Does this come from V&T 2019? Can you explain it? (or am I being a bit dull?). Is it perhaps due to different molecular diffusivities?

Thanks for the question. Our motivation for choosing the same transition is that we cannot tell from the data where exactly the transition should occur, and therefore the data doesn't necessarily tell us that the transition is the same for salt and temperature. We have rephrased the sentence to clarify it.

L249–251: This fit is chosen for simplicity since it is not possible to determine from the data (Fig. 3) at which exact  $L^+$  the regime transitions of heat and salt transport occur.

L205: On the neglection of the conductive heat flux: This is fine, but can you give an indication of how big the error is? I usually think of this as around 10%, so smallish, Neglecting it for simplicity makes sense to me - it can always be incorporated later, but the reader might like an indication of how big an effect it is.

We ran brief tests with the conductive flux (the recommended linearised advection-diffusion form of Holland and Jenkins (1999)) turned on in MOM6, and mean melt rates were less than 10% smaller in accordance with Holland and Jenkins (1999) findings (Fig. R10)

We have added the following comment

L257–262: Note that we also neglect the conductive heat flux term of Eqn. 2; although the conductive heat flux may be an important term in some ice shelf cavity conditions (Holland and Jenkins, 1999; Arzeno et al., 2014; Washam et al., 2020; Wiskandt and Jourdain, in review), melt rates are not expected to decrease by more than 10% (Holland and Jenkins, 1999). Thus,

Figure R10: Effect of heat conductive flux in MOM6 ISOMIP+ with insulating ice (no heat flow into ice,  $Q_I^T = 0$ ) in blue and conducting ice (with the linearised advection-diffusion form of Holland and Jenkins (1999) (option C of Wiskandt and Jourdain, in review) in orange. Left panel shows the warm experiment and right the cold. The StratFeedback parameterisation is used with ISOMIP+ protocol of prescribed tidal velocity (i.e.  $U_{tide} = 0.01 \,\mathrm{m\,s^{-1}}$ ).

we do not expect qualitatively different conclusions when we omit the conductive heat flux term

L211: I think the reader might appreciate a brief discussion of how the drag coefficient is considered a feature of the flow field rather than of the ice base. It would certainly help me.

Thanks for the suggestion. We have expanded on this in the drag coefficient section of 2.1:

L191: However, the drag coefficient, representing the scales of turbulent velocities compared to far-field flow speeds, is...

and also in the near the line suggested:

L267–269: Monin-Obukhov theory expects that under a stabilising buoyancy flux, the drag coefficient is also reduced as the friction velocity is suppressed relative to a fixed far-field velocity (the drag coefficient is defined as the ratio of these speeds)..

The drag coefficient is a feature of the flow field but the ice base can also be used to quantify it (e.g. Washam et al., 2023; Lawrence et al., 2023) so to avoid confusion we would prefer not to state that the ice base doesn't affect the drag coefficient.

L226: I think this should be  $p_b$ . It won't make a lot of difference in warmer cavities, but it's important in their cold counterparts.

Thanks, done. We have also changed the equation of state constants to be different letters  $(\lambda_1, \lambda_2, \lambda_3)$  in case there was confusion with pressure contribution to the freezing point cp and the specific heat capacity  $c_p$ .

Fig2 caption: "phase diagram" is a bit of a reserved term. Why "phase"?

We have replaced phase diagram with parameter space diagram.

Fig2 caption: I'm confused. Should these be "broken"?

Thanks for finding this error from a previous iteration of the figure. We have fixed it.

Fig2c caption: Could this point of separation be highlighted with a white solid line? It would look a bit like a parabola.

Added to Fig 2c. See Fig. R11.

L236: This is fine, but it's quite extreme. It might be worth stating that. Typical speeds of 1 cm/s are very, very low.

We have added this.

L293–294: At a thermal driving of  $T^* = 2^{\circ}\text{C}$  and  $u_* = 0.001 \text{ m s}^{-1}$  (far-field velocities of  $\sim 2 \text{ cm/s}$ , on the lower end of observed speeds; Table B1), StratFeedback predicts 30%...

L257: I would prefer "might be" here: we don't yet know that small L+ is a sufficient condition for diffusive convection.

Thanks, added as suggested.

L264: I don't think this makes a lot of sense. A lack of observations is not a good argument.

That is fair, we have removed the sentence.

L287: 10 degrees seems a bit arbitrary. I understand that this is the limit of applicability for MK18, but 10 degrees is a huge basal slope for an ice shelf. A sentence explaining the choice would perhaps help.

Thanks for the comment. Also partly in response to reviewer 1, we have added a paragraph about the small-scale features observed beneath ice shelves which may have large basal slopes where MK18 may be applicable (e.g. Washam et al., 2023, see features with angles up to 83°). We agree that in the coarse ocean models and datasets used, 10 degrees is a very large basal slope. We have tried to reframe the idea of slopes throughout the paper to discuss both the fact that larger slopes exist and that they cannot generally be resolved.

L102-118: It is important to highlight the many spatial scales involved in ice shelf basal melting. Considering vertical resolution, the processes within the ice shelf ocean boundary layer can be less than  $\mathcal{O}(10^{-3})$  m in size, hence the need for basal melt parameterisations in ocean models. Horizontally, the ice shelf base and bottom topography have significant spatial variability on scales between  $\mathcal{O}(10^{-1}-10^3)$  m, with melt rate varying correspondingly (Nicholls et al., 2006; Dutrieux et al., 2014; Alley et al., 2016; Watkins et al., 2021; Schmidt et al., 2023; Washam et al., 2023; Wåhlin et al., 2024). For example, in an ice base crevasse, melt rates can be enhanced at the terrace side-walls, with freezing by buoyant, supercooled water at the top of the crevasse (Washam et al., 2023), indicating multiple physical drivers of melt within a small distance. A variety of ice features such as scallops and terraces can form depending on the ice melt regime (Washam et al., 2023; Wåhlin et al., 2024). Though idealised and process models have simulated some of these small-scale features (Jordan et al., 2014; Zhou and Hattermann, 2020; Wilson et al., 2023), and some high-resolution regional models may capture part of the spatial variability (Nakayama et al., 2019, 2021; Shrestha et al., 2024), large-scale ocean models generally have horizontal grid sizes greater than  $\mathcal{O}(10^3)$  m and vertical resolutions  $\mathcal{O}(10^1)$  m and cannot resolve ice base variability at the required scales, nor do commonly-used bathymetry and ice base forcing products (Morlighem et al., 2020). Therefore, although there are known regions of significant ice shelf base variability and high slopes (Washam et al., 2023; Schmidt et al., 2023; Wåhlin et al., 2024), much of Antarctic ice shelves are represented in ocean models as weakly sloped (

Figure R11: Thermal driving – friction velocity parameter space diagram indicating melt rates calculated as a function of far-field temperature, salinity and pressure (which are set to S =34.5 psu and p = 500 dbar) and friction velocity. The melt rates are solved for a variety of parameterisation options in (a) and (b). The orange, dotted lines in panel (a) are the Holland and Jenkins (1999) formulation with the  $\eta_*$  stratification parameter (McPhee, 1981). A constant transfer coefficient formulation (ConstCoeff) is in the blue solid lines (using the maximal values of Vreugdenhil and Taylor (2019), which is similar to Jenkins et al. (2010) in the light yellow solid lines), and the stratification feedback (StratFeedback) parameterisation we develop here is shown in white dashed lines. In panel (b), we add in the constant melt rates obtained from McConnochie and Kerr (2018) with a slope angle of  $\theta = 10^{\circ}$  from the horizontal in the pink dotted line, and the combination of the StratFeedback+MK18 limit in the red dash-dot line. Panel (c) shows the viscous Obukhov scale  $L^+$  derived from the stratification feedback parameterisation, and where the white  $L^+ = 1 \times 10^4$  indicates where the white dashed lines (StratFeedback) and blue line (ConstCoeff) transition from having the same melt rate to the right and different to the left. Panel (d) also shows this in the ratio of the StratFeedback to ConstCoeff melt rates, with stippling indicating where they are equal.

Also partly in response to reviewer 1, we have added Fig A2 to the appendix which shows the same plot with three other angles (including a smaller angle of 2°, Fig. R12).

Figure R12: Thermal driving – friction velocity parameter space diagram indicating melt rates calculated as a function of far-field temperature, salinity and pressure (which are set to S=34.5 psu and p=500 dbar) and friction velocity. The melt rates are solved for a variety of parameterisation options: ConstCoeff is in the blue solid lines, and StratFeedback is shown in white dashed lines. Constant melt rates obtained from slope-dependent McConnochie and Kerr (2018) convective parameterisation are in the pink dotted lines, and the combination of the StratFeedback+MK18 limit is in the red dash-dot line. The three panels show different slope angle choices to Fig. 2b, with angles from the horizontal of (a)  $2^{\circ}$ , (b)  $45^{\circ}$  and (c)  $80^{\circ}$ .

And we have added a line where suggested to motivate 10 degrees:

L347–349: ...the MK18 limit at a given ice base angle of 10°, a value within the limits of that observed by Wåhlin et al. (2024), though equivalent plots with different angles are provided in Appendix A).

L348: Is this the same as the "bulk mixed layer"? Or is that divided into other multiple layers? A more detailed description of the nature of the way the upper part of the water column is discretized in the models would be helpful.

The bulk mixed layer we use consists of two layers. We have added:

L424–425: The friction velocity  $u_*$  is calculated from the velocities in the uppermost model layer (the top half of the bulk mixed layer, approximately 5 m thick)

L390: So this is new? It isn't in Table 2.

Thanks for the pick up, we have now added these tuned parameterisations into Table 2.

L424: Fine, but a bit obvious?

We would like to keep this statement of the strong feedback between melt and circulation because it explains the large change between StratFeedback and ConstCoeff simulation circulation.

L455: I assume this is because the value selected for T\* is from a greater depth? And therefore a higher temperature? Perhaps a half sentence to help the reader?

That is correct, we have rewritten:

L549–553: The different magnitude of melt between models may be explained chiefly by the different vertical coordinates (Gwyther et al., 2020), where the z-level coordinates of MITgcm result in a coarser vertical resolution near the ice, and therefore a stronger thermal driving

since the far-field temperature is sampled at a greater depth, but may also be associated with other model choices such as the vertical mixing scheme.

L509: Is there a particular reason for not using one of the satellite-derived datasets for tuning? A statement about why the HJ99-neutral experiment was selected would be useful.

There was no particular reason other than simplicity! As also suggested by other reviewers, we have rerun the Pine Island Glacier simulations, tuning to Adusumilli et al. (2020) melt rates. The revised section 4.4 is provided at the end of this document.

L607: I'm not sure what the authors are saying here. It clearly isn't relevant to the ice base in ice sheet models, if these are grounded ice sheets. Perhaps they mean that the parameterisation is relevant to ice shelf cavities that might be included in ice sheet models? If so, then I think the sentence can be deleted, as I think the reader appreciates that that will be the ultimate goal.

In this sentence we were referring to basal melt parameterisations for standalone ice sheet models, which are generally distinct from ocean models as they do not include 3D ocean circulation beneath ice shelves. These parameterisations tend to be even simpler than the three-equation parameterisation and almost certainly do not include the stratification feedback of melt. We have made this more explicit:

L727–730: This work has focused on basal melt parameterisations in ocean models, but there is also scope to translate the effect of stratified melt rate feedbacks into basal melt parameterisations for stand-alone ice sheet models, where ice shelf basal melt parameterisations tend to be even further simplified than in ocean models (e.g. Burgard et al., 2022)

**Revised Section 4.4**

**Sec 4.4: Realistic Pine Island Glacier Simulation**

To assess the parameterisation in a realistic situation where circulation is more complex and the results can be compared with observations, we use the MITgcm Pine Island Glacier setup of Nakayama et al. (2021) (model details in Section 3.2) with its drag coefficient tuned to achieve melt rates similar to the Adusumilli et al. (2020) satellite melt rate product. After 20 days of simulation, area-averaged melt rates are approximately equilibrated and of a similar magnitude of  $\sim 17 \,\mathrm{m/yr}$  (as a result of the tuning, Table 2, noting that this rate refers to the whole simulated cavity average rather than masked tuning melt rate in Sect. 3.2). We compare the melt rate distributions for three different parameterisation choices averaged over days 20-50. The simulation run with the Holland and Jenkins (1999) parameterisation and McPhee (1981)  $\eta_*$  stability parameter set to 1, hereafter HJ99-neutral, requires the lowest tuning drag coefficient ( $C_d = 0.004$ ), corresponding to the largest average melt rate if tuning is not performed (i.e. using  $C_d = 0.0015$  gives an average melt rate of  $11.3 \,\mathrm{m/yr}$ , Supplementary Fig. S2a). The StratFeedback parameterisation without tuning yielded a melt rate of 4 m/yr (Fig. S2b) and required a larger tuning drag coefficient of  $C_d = 0.0073$ . This implies that much of the Pine Island Glacier ice shelf is in the stratified regime. Furthermore, when we include the MK18 low-velocity limit in the untuned simulations, the melt rates increase to an average of 8 m/yr (Fig. S2c). This melt rate is larger than the untuned StratFeedback simulation because the relatively large ice base slopes (up to 30°) contribute substantial melting via the MK18 parameterisation. Because the untuned StratFeedback simulation has the weakest melting, the tuned StratFeedback simulation has the largest drag coefficient of the three tuned simulations so that the same mean melt rate is achieved. By using the tuned simulations, we can more

Figure R13: Fig 9: MITgcm Pine Island Glacier melt rates averaged over days 20-50 of the simulation, for (a) the tuned HJ99-neutral basal melt parameterisation used in Nakayama et al. (2021) (with a drag coefficient of  $C_d = 0.004$ ); (b) the anomaly of the tuned stratification feedback (StratFeedback) parameterisation (with a drag coefficient of  $C_d = 0.0073$ ) and (c) the anomaly of the tuned stratification feedback parameterisation with MK18 limit (with a drag coefficient of  $C_d = 0.0057$ )). Both anomalies in (b) and (c) are with respect to (a). The melt rates quoted are calculated over the whole simulated ice shelf area and differ from the tuning melt rate, which was only over the region where Adusumilli et al. (2020) data is present (Fig. D1b) and only south of 74.8°S. The friction velocity, thermal driving and viscous Obukhov scales of the stratification feedback parameterisation with tuned drag coefficient (b) are shown in panels (d), (e) and (f).

easily compare spatial distributions of melt rate and the parameterisation's effect on ocean properties. Note the tuned drag coefficients ( $C_d = 0.004$  for tuned HJ99-neutral, 0.0073 for tuned StratFeedback and 0.0057 for tuned StratFeedback+MK18) all lie between the value  $C_d = 0.0015$  used in the original simulation and the value  $C_d = 0.0097$  suggested by Jenkins et al. (2010) (see Sec. 2.1 for more observational estimates of drag coefficients).

In the tuned HJ99-neutral simulation, melt is enhanced near the grounding line (Fig. 9a), and reaches the observed melt rates of up to  $200\,\mathrm{m/yr}$  in this region (Shean et al., 2019; Zinck et al., in review, see probability distribution in Fig. D1). Melt is also enhanced at the ice shelf keels (Fig. 9a), as in Shean et al. (2019). Unlike observations which suggest low melt rates in the northern part of the ice shelf, simulated melt rates reach  $\sim 50\,\mathrm{m/yr}$  in this region (compare Fig. 9a and Figs. D1b,c). The difference suggests there may be differences between the simulated and real pathways of water masses into the northern section of the Pine Island Glacier ice shelf cavity.

In the tuned StratFeedback simulation, the melt rates are increased relative to the tuned HJ99-neutral simulation in some regions, such as near the Pine Island Glacier grounding line and in the ice shelf keels (and some channels with high velocities), and decreased elsewhere (Fig. 9b). The regions where the tuned StratFeedback simulation enhances melt correspond to regions with large friction velocities (Fig. 9d) and melt decreases in regions with low friction velocities, including some regions near the grounding line. In the large friction velocity regions,  $L^+$  is also large (Fig. 9f), indicating melting in the well-mixed regime, whereas regions with lower friction velocities have lower  $L^+$  and are simulated to be in the stratified and diffusive-convective regimes. The StratFeedback parameterisation therefore enhances the spatial variability in melt beneath Pine Island Glacier. Ocean properties and circulation respond to this modified melt rate, leading to fresher, colder water in regions with more melting (see Supplementary Figs. S3,S4,S5).

The tuned StratFeedback+MK18 simulation has a similar melt rate anomaly pattern to the tuned StratFeedback simulation despite the different drag coefficients. However, in addition to having enhanced melt in regions with large friction velocity compared to the HJ99-neutral experiment, melt is also enhanced at the sloped ice shelf front and near the grounding line, the latter where the thermal driving is large (Fig. 9e). Both the tuned StratFeedback and StratFeedback+MK18 experiments have a larger area of the ice shelf with melting greater than 50 m/yr compared with the tuned HJ99-neutral simulation (Fig. D1), and align better with the high-resolution observational products near the grounding line (Zinck et al., in review; Shean et al., 2019). However, the missing data and coarser resolution of Adusumilli et al. (2020) make quantitative comparison challenging. Since the drag coefficient also affects the momentum equation's drag law, the less aggressive tuning required by MK18, combined with generally higher melt rates near the grounding line, indicate that the MK18 lower limit choice may be more appropriate choice. However, this assessment should be extended to other ice shelves. The general similarity between StratFeedback and StratFeedback+MK18 also indicates that large parts of the tuned Pine Island Glacier simulations are not in the low-velocity regime (with the diffusive-convective regime as a guide in Fig. 9f).

The difference in the spatial distribution of melt rates between the original simulation and that with the StratFeedback parameterisation highlights the spatial heterogeneity in melt rate regimes within individual ice shelves. All three regimes, well-mixed shear-driven, stratified and diffusive-convective, were observed in the tuned simulations (Fig. 9f). Analysis of borehole observations from Pine Island Glacier yielded a shear-driven  $L^+$  of  $1.1\times10^4$  (Fig. 3b), which was taken in one of the channels approximately halfway between the ice front and grounding zone (Stanton et al., 2013). Without the precise location in the ice shelf (and noting differences in time of simulation and observation), it is difficult to determine if the simulated channels'  $L^+$  agree with the observation. However, keels and some channels are generally simulated to be in the shear-driven regime, potentially in agreement with Stanton et al. (2013). Nevertheless, the need for significantly different drag coefficients between tuned simulations demonstrates the sensitivity of regional ice shelf models' basal melting and melt regimes to parameterisations.

**Appendix D1: Pine Island Glacier melt rate distributions compared to observations**

Fig. D1a compares the distribution of melt rates between the three tested parameterisations, as well as melt rates computed from the Adusumilli et al. (2020) and Zinck et al. (in review) satellite-derived melt rate products. Whilst not directly comparable, due to different resolutions and ice shelf area due to missing data (e.g. at the grounding line, where the grounding line is taken from Morlighem et al. (2020), see Figs. D1b,c), the tuned StratFeedback parameterisation (blue colours) and StratFeedback+MK18 (yellow colours) have larger positive melt rate tails than the HJ99-neutral experiment (pink colours), more similar to the large (~ 200 m/yr) melt rates observed near the grounding line in high-resolution satellite products (grey colours, Zinck et al., in review), and Shean et al. (2019). Note the two satellite products here differ significantly,

Figure R14: Fig D1: (a) Melt rate statistical distributions in Pine Island Glacier, for the MITgcm simulation with three different basal melt parameterisations compared with Adusumilli Adusumilli et al.) and Zinck et al. (in review) (data: et al. (2020) (data: 2024). Note that the Adusumilli et al. (2020) product is coarser-resolution (500 m) than the MITgcm model (200 m) and is missing data whilst Zinck et al. (in review) is finer resolution (50 m). The y-axis is logarithmically scaled. MITgcm data is averaged over simulation days 20-50. (b) Adusumilli et al. (2020) melt rate and (c) Zinck et al. (in review) melt rates at Pine Island Glacier, with the same colourbar as Fig. 10a-c, but note it is rotated with the Antarctic Ice Sheet at the top of the figure and ocean at the bottom. The Bedmachine V3 surface elevation (Morlighem et al., 2020) (data: Morlighem, 2022) is shown in grey and the associated ice shelf mask is outlined with a black contour. The model domain is outlined with a grey dashed contour. Data from Adusumilli et al. is licensed under CC BY 4.0 (https: //creativecommons.org/licenses/by/4.0/) and Zinck et al. (2024) is licensed under CC BY-SA 4.0 (https://creativecommons.org/licenses/by-sa/4.0/) and have been adapted in this Figure.

highlighting the uncertainty in satellite-derived melt rates. The time periods of the satellite products and model run also differ.

**References**

Adusumilli, S., Fricker, H. A., Medley, B. C., Padman, L., and Siegfried, M. R.: Data from: Interannual variations in meltwater input to the Southern Ocean from Antarctic ice shelves., UC San Diego Library Digital Collections., https://doi.org/10.6075/J04Q7SHT.

Adusumilli, S., Fricker, H. A., Medley, B., Padman, L., and Siegfried, M. R.: Interannual variations in meltwater input to the Southern Ocean from Antarctic ice shelves, Nature Geoscience, 13, 616–620, https://doi.org/10.1038/s41561-020-0616-z, 2020.

Alley, K. E., Scambos, T. A., Siegfried, M. R., and Fricker, H. A.: Impacts of warm water on Antarctic ice shelf stability through basal channel formation, Nature Geoscience, 9, 290–293, https://doi.org/10.1038/ngeo2675, 2016.

Arzeno, I. B., Beardsley, R. C., Limeburner, R., Owens, B., Padman, L., Springer, S. R., Stewart, C. L., and Williams, M. J.: Ocean variability contributing to basal melt rate near

- the ice front of Ross Ice Shelf, Antarctica, Journal of Geophysical Research: Oceans, 119, 4214–4233, https://doi.org/10.1002/2014JC009792, 2014.
- Asay-Davis, X. S., Cornford, S. L., Durand, G., Galton-Fenzi, B. K., Gladstone, R. M., Gudmundsson, G. H., Hattermann, T., Holland, D. M., Holland, D., Holland, P. R., et al.: Experimental design for three interrelated marine ice sheet and ocean model intercomparison projects: MISMIP v. 3 (MISMIP+), ISOMIP v. 2 (ISOMIP+) and MIS-OMIP v. 1 (MISOMIP1), Geoscientific Model Development, 9, 2471–2497, https://doi.org/10.5194/gmd-9-2471-2016, 2016.
- Begeman, C. B., Tulaczyk, S. M., Marsh, O. J., Mikucki, J. A., Stanton, T. P., Hodson, T. O., Siegfried, M. R., Powell, R. D., Christianson, K., and King, M. A.: Ocean stratification and low melt rates at the Ross Ice Shelf grounding zone, Journal of Geophysical Research: Oceans, 123, 7438–7452, https://doi.org/10.1029/2018JC013987, 2018.
- Burgard, C., Jourdain, N. C., Reese, R., Jenkins, A., and Mathiot, P.: An assessment of basal melt parameterisations for Antarctic ice shelves, The Cryosphere, 16, 4931–4975, https://doi.org/10.5194/tc-16-4931-2022, 2022.
- Darelius, E., Makinson, K., Daae, K., Fer, I., Holland, P. R., and Nicholls, K. W.: Hydrography and circulation in the Filchner depression, Weddell Sea, Antarctica, Journal of Geophysical Research: Oceans, 119, 5797–5814, https://doi.org/10.1002/2014JC010225, 2014.
- Davis, P. E. and Nicholls, K. W.: Turbulence observations beneath Larsen C ice shelf, Antarctica, Journal of Geophysical Research: Oceans, 124, 5529–5550, https://doi.org/10.1029/2019JC015164, 2019.
- Davis, P. E., Nicholls, K. W., Holland, D. M., Schmidt, B. E., Washam, P., Riverman, K. L., Arthern, R. J., Vaňková, I., Eayrs, C., Smith, J. A., et al.: Suppressed basal melting in the eastern Thwaites Glacier grounding zone, Nature, 614, 479–485, https://doi.org/10.1038/s41586-022-05586-0, 2023.
- Dutrieux, P., Stewart, C., Jenkins, A., Nicholls, K. W., Corr, H. F., Rignot, E., and Steffen, K.: Basal terraces on melting ice shelves, Geophysical Research Letters, 41, 5506–5513, https://doi.org/10.1002/2014GL060618, 2014.
- Gwyther, D. E., Cougnon, E. A., Galton-Fenzi, B. K., Roberts, J. L., Hunter, J. R., and Dinniman, M. S.: Modelling the response of ice shelf basal melting to different ocean cavity environmental regimes, Annals of Glaciology, 57, 131–141, https://doi.org/10.1017/aog.2016. 31, 2016.
- Gwyther, D. E., Kusahara, K., Asay-Davis, X. S., Dinniman, M. S., and Galton-Fenzi, B. K.: Vertical processes and resolution impact ice shelf basal melting: A multi-model study, Ocean Modelling, 147, 101 569, https://doi.org/10.1016/j.ocemod.2020.101569, 2020.
- Hallberg, R.: The ability of large-scale ocean models to accept parameterizations of boundary mixing, and a description of a refined bulk mixed-layer model, in: Internal Gravity Waves and Small-Scale Turbulence: Proc.'Aha Huliko 'a Hawaiian Winter Workshop, pp. 187–203, https://www.soest.hawaii.edu/PubServices/2003pdfs/Hallberg.pdf, 2003.
- Hellmer, H. H. and Olbers, D. J.: A two-dimensional model for the thermohaline circulation under an ice shelf, Antarctic Science, 1, 325–336, https://doi.org/10.1017/S0954102089000490, 1989.
- Holland, D. M. and Jenkins, A.: Modeling thermodynamic ice-ocean interactions at the base

- of an ice shelf, Journal of Physical Oceanography, 29, 1787–1800, https://doi.org/10.1175/1520-0485(1999)029 $\langle$ 1787:MTIOIA $\rangle$ 2.0.CO;2, 1999.
- Hyogo, S., Nakayama, Y., and Mensah, V.: Modeling ocean circulation and ice shelf melt in the Bellingshausen Sea, Journal of Geophysical Research: Oceans, 129, e2022JC019275, https://doi.org/10.1029/2022JC019275, 2024.
- Jackson, L., Hallberg, R., and Legg, S.: A parameterization of shear-driven turbulence for ocean climate models, Journal of Physical Oceanography, 38, 1033–1053, https://doi.org/10.1175/2007JPO3779.1, 2008.
- Jenkins, A.: A simple model of the ice shelf-ocean boundary layer and current, Journal of Physical Oceanography, 46, 1785–1803, 2016.
- Jenkins, A.: Shear, stability, and mixing within the ice shelf-ocean boundary current, Journal of Physical Oceanography, 51, 2129–2148, 2021.
- Jenkins, A., Nicholls, K. W., and Corr, H. F.: Observation and parameterization of ablation at the base of Ronne Ice Shelf, Antarctica, Journal of Physical Oceanography, 40, 2298–2312, https://doi.org/10.1175/2010JPO4317.1, 2010.
- Jordan, J. R., Holland, P. R., Jenkins, A., Piggott, M. D., and Kimura, S.: Modeling ice-ocean interaction in ice-shelf crevasses, Journal of Geophysical Research: Oceans, 119, 995–1008, https://doi.org/10.1002/2013JC009208, 2014.
- Jourdain, N. C., Molines, J.-M., Le Sommer, J., Mathiot, P., Chanut, J., de Lavergne, C., and Madec, G.: Simulating or prescribing the influence of tides on the Amundsen Sea ice shelves, Ocean Modelling, 133, 44–55, https://doi.org/10.1016/j.ocemod.2018.11.001, 2019.
- Kerr, R. C. and McConnochie, C. D.: Dissolution of a vertical solid surface by turbulent compositional convection, Journal of Fluid Mechanics, 765, 211–228, https://doi.org/10.1017/jfm.2014.722, 2015.
- Kimura, S., Nicholls, K. W., and Venables, E.: Estimation of ice shelf melt rate in the presence of a thermohaline staircase, Journal of Physical Oceanography, 45, 133–148, https://doi.org/10.1175/JPO-D-14-0106.1, 2015.
- Kraus, E. and Turner, J.: A one-dimensional model of the seasonal thermocline II. The general theory and its consequences, Tellus, 19, 98–106, https://doi.org/10.3402/tellusa.v19i1.9753, 1967.
- Lawrence, J., Washam, P., Stevens, C., Hulbe, C., Horgan, H., Dunbar, G., Calkin, T., Stewart, C., Robinson, N., Mullen, A., et al.: Crevasse refreezing and signatures of retreat observed at Kamb Ice Stream grounding zone, Nature Geoscience, 16, 238–243, https://doi.org//10.1038/s41561-023-01129-y, 2023.
- Losch, M.: Modeling ice shelf cavities in a z coordinate ocean general circulation model, Journal of Geophysical Research: Oceans, 113, https://doi.org/10.1029/2007JC004368, 2008.
- Makinson, K., Schröder, M., and Østerhus, S.: Effect of critical latitude and seasonal stratification on tidal current profiles along Ronne Ice Front, Antarctica, Journal of Geophysical Research: Oceans, 111, https://doi.org/10.1029/2005JC003062, 2006.
- Malyarenko, A., Wells, A. J., Langhorne, P. J., Robinson, N. J., Williams, M. J., and Nicholls, K. W.: A synthesis of thermodynamic ablation at ice—ocean interfaces from theory, observations and models, Ocean Modelling, 154, 101692, https://doi.org/10.1016/j.ocemod.2020. 101692, 2020.

- McConnochie, C. and Kerr, R.: Testing a common ice-ocean parameterization with laboratory experiments, Journal of Geophysical Research: Oceans, 122, 5905–5915, https://doi.org/10.1002/2017JC012918, 2017.
- McConnochie, C. D. and Kerr, R. C.: Dissolution of a sloping solid surface by turbulent compositional convection, Journal of Fluid Mechanics, 846, 563–577, https://doi.org/10.1017/jfm.2018.282, 2018.
- McPhee, M.: Air-ice-ocean interaction: Turbulent ocean boundary layer exchange processes, Springer Science & Business Media, 2008.
- McPhee, M. G.: An analytic similarity theory for the planetary boundary layer stabilized by surface buoyancy, Boundary-Layer Meteorology, 21, 325–339, https://doi.org/10.1007/BF00119277, 1981.
- McPhee, M. G., Maykut, G. A., and Morison, J. H.: Dynamics and thermodynamics of the ice/upper ocean system in the marginal ice zone of the Greenland Sea, Journal of Geophysical Research: Oceans, 92, 7017–7031, https://doi.org/10.1029/JC092iC07p07017, 1987.
- Middleton, L., Davis, P., Taylor, J., and Nicholls, K.: Double diffusion as a driver of turbulence in the stratified boundary layer beneath George VI Ice Shelf, Geophysical Research Letters, 49, e2021GL096119, https://doi.org/10.1029/2021GL096119, 2022.
- Mondal, M., Gayen, B., Griffiths, R. W., and Kerr, R. C.: Ablation of sloping ice faces into polar seawater, Journal of Fluid Mechanics, 863, 545–571, https://doi.org/10.1017/jfm.2018.970, 2019.
- Morlighem, M.: MEaSUREs BedMachine Antarctica, Version 3., https://doi.org/10.5067/FPSU0V1MWUB6, accessed 22 October 2024, 2022.
- Morlighem, M., Rignot, E., Binder, T., Blankenship, D., Drews, R., Eagles, G., Eisen, O., Ferraccioli, F., Forsberg, R., Fretwell, P., et al.: Deep glacial troughs and stabilizing ridges unveiled beneath the margins of the Antarctic ice sheet, Nature Geoscience, 13, 132–137, https://doi.org/10.1038/s41561-019-0510-8, 2020.
- Mueller, R., Padman, L., Dinniman, M. S., Erofeeva, S., Fricker, H. A., and King, M.: Impact of tide-topography interactions on basal melting of Larsen C Ice Shelf, Antarctica, Journal of Geophysical Research: Oceans, 117, https://doi.org/10.1029/2011JC007263, 2012.
- Nakayama, Y., Menemenlis, D., Zhang, H., Schodlok, M., and Rignot, E.: Origin of Circumpolar Deep Water intruding onto the Amundsen and Bellingshausen Sea continental shelves, Nature Communications, 9, 1–9, https://doi.org/10.1038/s41467-018-05813-1, 2018.
- Nakayama, Y., Manucharyan, G., Zhang, H., Dutrieux, P., Torres, H. S., Klein, P., Seroussi, H., Schodlok, M., Rignot, E., and Menemenlis, D.: Pathways of ocean heat towards Pine Island and Thwaites grounding lines, Scientific Reports, 9, 16649, https://doi.org/10.1038/s41598-019-53190-6, 2019.
- Nakayama, Y., Cai, C., and Seroussi, H.: Impact of subglacial freshwater discharge on Pine Island Ice Shelf, Geophysical Research Letters, 48, e2021GL093923, https://doi.org/10.1029/2021GL093923, 2021.
- Nicholls, K., Abrahamsen, E., Buck, J., Dodd, P., Goldblatt, C., Griffiths, G., Heywood, K., Hughes, N., Kaletzky, A., Lane-Serff, G., et al.: Measurements beneath an Antarctic ice shelf using an autonomous underwater vehicle, Geophysical Research Letters, 33, https://doi.org/10.1029/2006GL025998, 2006.

- Nicholls, K. W., Makinson, K., and Østerhus, S.: Circulation and water masses beneath the northern Ronne Ice Shelf, Antarctica, Journal of Geophysical Research: Oceans, 109, https://doi.org/10.1029/2004JC002302, 2004.
- Niiler, P. and Kraus, E.: One-dimensional models of the upper ocean., in: E.B. Kraus (Editor), Modelling and Prediction of the Upper Layers of the Ocean., pp. 143–172, Pergamon Press, 1977.
- Orlanski, I.: A simple boundary condition for unbounded hyperbolic flows, Journal of Computational Physics, 21, 251–269, https://doi.org/10.1016/0021-9991(76)90023-1, 1976.
- Paolo, F. S., Fricker, H. A., and Padman, L.: Volume loss from Antarctic ice shelves is accelerating, Science, 348, 327–331, https://doi.org/10.1126/science.aaa0940, 2015.
- Pope, S. B.: Turbulent flows, Measurement Science and Technology, 12, 2020–2021, 2001.
- Rignot, E., Mouginot, J., Scheuchl, B., Van Den Broeke, M., Van Wessem, M. J., and Morlighem, M.: Four decades of Antarctic Ice Sheet mass balance from 1979–2017, Proceedings of the National Academy of Sciences, 116, 1095–1103, https://doi.org/10.1073/pnas.1812883116, 2019.
- Rosevear, M. G., Galton-Fenzi, B., and Stevens, C.: Evaluation of basal melting parameterisations using in situ ocean and melting observations from the Amery Ice Shelf, East Antarctica, Ocean Science, 18, 1109–1130, https://doi.org/0.5194/os-18-1109-2022, 2022a.
- Rosevear, M. G., Gayen, B., and Galton-Fenzi, B. K.: Regimes and transitions in the basal melting of Antarctic ice shelves, Journal of Physical Oceanography, 52, 2589–2608, https://doi.org/10.1175/JPO-D-21-0317.1, 2022b.
- Rosevear, M. G., Gayen, B., Vreugdenhil, C. A., and Galton-Fenzi, B. K.: How Does the Ocean Melt Antarctic Ice Shelves?, Annual Review of Marine Science, 17, https://doi.org/10.1146/annurev-marine-040323-074354, 2024.
- Schmidt, B. E., Washam, P., Davis, P. E., Nicholls, K. W., Holland, D. M., Lawrence, J. D., Riverman, K. L., Smith, J. A., Spears, A., Dichek, D., et al.: Heterogeneous melting near the Thwaites Glacier grounding line, Nature, 614, 471–478, https://doi.org/10.1038/s41586-022-05691-0, 2023.
- Schulz, K., Nguyen, A., and Pillar, H.: An Improved and Observationally-Constrained Melt Rate Parameterization for Vertical Ice Fronts of Marine Terminating Glaciers, Geophysical Research Letters, 49, e2022GL100654, https://doi.org/10.1029/2022GL100654, 2022.
- Shean, D. E., Joughin, I. R., Dutrieux, P., Smith, B. E., and Berthier, E.: Ice shelf basal melt rates from a high-resolution digital elevation model (DEM) record for Pine Island Glacier, Antarctica, The Cryosphere, 13, 2633–2656, https://doi.org/10.5194/tc-13-2633-2019, 2019.
- Shrestha, K., Manucharyan, G. E., and Nakayama, Y.: Submesoscale variability and basal melting in ice shelf cavities of the Amundsen Sea, Geophysical Research Letters, 51, e2023GL107029, https://doi.org/10.1029/2023GL107029, 2024.
- Stanton, T. P., Shaw, W., Truffer, M., Corr, H., Peters, L., Riverman, K., Bindschadler, R., Holland, D., and Anandakrishnan, S.: Channelized ice melting in the ocean boundary layer beneath Pine Island Glacier, Antarctica, Science, 341, 1236–1239, https://doi.org/10.1126/science.1239373, 2013.
- Stewart, C. L.: Ice-ocean interactions beneath the north-western Ross Ice Shelf, Antarctica, Ph.D. thesis, https://doi.org/10.17863/CAM.21483, 2018.

- Vreugdenhil, C. A. and Taylor, J. R.: Stratification effects in the turbulent boundary layer beneath a melting ice shelf: Insights from resolved large-eddy simulations, Journal of Physical Oceanography, 49, 1905–1925, https://doi.org/10.1175/JPO-D-18-0252.1, 2019.
- Wåhlin, A., Alley, K. E., Begeman, C., Hegrenæs, Ø., Yuan, X., Graham, A. G., Hogan, K., Davis, P. E., Dotto, T. S., Eayrs, C., et al.: Swirls and scoops: Ice base melt revealed by multibeam imagery of an Antarctic ice shelf, Science Advances, 10, eadn9188, https://doi.org/10.1126/sciadv.adn9188, 2024.
- Washam, P., Nicholls, K. W., Münchow, A., and Padman, L.: Tidal modulation of buoyant flow and basal melt beneath Petermann Gletscher Ice Shelf, Greenland, Journal of Geophysical Research: Oceans, 125, e2020JC016427, https://doi.org/10.1029/2020JC016427, 2020.
- Washam, P., Lawrence, J. D., Stevens, C. L., Hulbe, C. L., Horgan, H. J., Robinson, N. J., Stewart, C. L., Spears, A., Quartini, E., Hurwitz, B., et al.: Direct observations of melting, freezing, and ocean circulation in an ice shelf basal crevasse, Science Advances, 9, eadi7638, https://doi.org/10.1126/sciadv.adi7638, 2023.
- Watkins, R. H., Bassis, J. N., and Thouless, M.: Roughness of ice shelves is correlated with basal melt rates, Geophysical Research Letters, 48, e2021GL094743, https://doi.org/10.1029/2021GL094743, 2021.
- Wilson, N. J., Vreugdenhil, C. A., Gayen, B., and Hester, E. W.: Double-Diffusive Layer and Meltwater Plume Effects on Ice Face Scalloping in Phase-Change Simulations, Geophysical Research Letters, 50, e2023GL104396, https://doi.org/10.1029/2023GL104396, 2023.
- Wiskandt, J. and Jourdain, N.: Brief Communication: Representation of heat conduction into the ice in marine ice shelf melt modeling, EGUsphere, 2024, 1–10, https://doi.org/10.5194/egusphere-2024-2239, in review.
- Zhao, K. X., Skyllingstad, E. D., and Nash, J. D.: Improved parameterizations of vertical ice-ocean boundary layers and melt rates, Geophysical Research Letters, 51, e2023GL105862, https://doi.org/10.1029/2023GL105862, 2024.
- Zhou, Q. and Hattermann, T.: Modeling ice shelf cavities in the unstructured-grid, Finite Volume Community Ocean Model: Implementation and effects of resolving small-scale topography, Ocean Modelling, 146, 101 536, 2020.
- Zinck, A.-S. P., Lhermitte, S., Wearing, M., and Wouters, B.: Dataset belonging to the article: Exposure to Underestimated Channelized Melt in Antarctic Ice Shelves, https://doi.org/10.4121/4e2ba9a9-7b1b-4837-b52d-036f8c876e67.v1, 2024.
- Zinck, A.-S. P., Lhermitte, S., Wearing, M., and Wouters, B.: Exposure to Underestimated Channelized Melt in Antarctic Ice Shelves, https://doi.org/10.21203/rs.3.rs-4806463/v1, in review.

---

## Referee Report (RR1)

**Review:**

**Egusphere-2024-3513: Stratified suppression of turbulence in an ice shelf basal melt parameterisation**

**Overall Statement:**

The team of authors have submitted a much-improved version of this bear of a manuscript that tackles the difficult, but important problem of including stratification in ice shelf-ocean models. Overall, the manuscript is quite good, with improved clarity and readability throughout. It is a big, chunky manuscript with a lot of detail that will inspire further work. Apart from some parts of the text needing improved writing structure and some improvement of the Methods Section, I feel that this manuscript will only need a minor review to be ready for publication. I have provided an assortment of comments below that are trivial and should not take much time to address. Great job turning around a solidly-revised manuscript!

-Peter Washam

**Larger Comments:**

Part of Methods section is still unclear – see comments below.

I don't think it is appropriate to cite manuscripts in review.

**Specific Revisions:**

**Abstract:**

- Li 2 15: This now fits within the 250 word limit of The Cryosphere. Still, it is quite a long abstract, so consider shortening it somewhat.
- Li 4-5: This sentence is hard to follow. Please rewrite improve readability.
- Li 7: rewrite to "stratification by accumulation of buoyant meltwater beneath a flat ice interface"

**Introduction:**

- Li 19-21: Insert somewhere in here that ice shelves have already entered the ocean and displaced sea level.
- Li 25 26: add "in the ocean models that produce melt rate projections (IPCC, 2023; Bennetts et al., 2024)."

- Li 28: As someone who does observations, I appreciate it when modelers define a relative scale for models to place them into context. Can you express what scale this "large-scale ocean, climate and earth system models" are operating on? Is this referring to global or circum-Antarctic?
- Li 52: You have not defined the sections of the ice-ocean boundary layer yet, so change this to "the far-field flow below the ice-ocean boundary layer"
- Li 57 60: This is a quite verbose sentence with poor structure. Please rewrite to improve readability and be sure to mention that meltwater accumulation and stratification inhibits melting on ice that is flat or has low slopes. This then will lead into the next sentence.
- Li 62: The Washam et al., (2023) reference is fine if you'd like to cite it, but I think the Schimdt et al., (2023) reference is more appropriate.
- Li 67 68: This is an incomplete sentence.
- Li 73: "Vertical discretisation of the basal melt parameterisation" is a little heavy on the jargon. Please rewrite this to make it more clear what is trying to be communicated.
- Li 80: Not sure about citing a paper in review unless it will be out before this one.
- Li 81: Specify what you mean by "future ice shelf regime changes."
- Li 86 92: This sentence is incredibly hard to understand. I understand that the authors are trying to summarize the important missing processes in models, but please rewrite this with improved sentence structure to improve readability.
- Li 99: State the scale of "large-scale." Is this regional scale (1 large ice shelf or several small ones) or circum-Antarctic?
- Li 100: The previous statements on scale should help the reader better understand the goal of the paper now.
- Li 106 109: I think these sentences also refer to Schmidt et al., (2023).
- Li 112: What is "large-scale" here? Is this a regional model or a GCM? I would be amazed if a GCM had this sort of resolution.
- Li 119: Start this sentence with: "In this paper,"
- Li 134 142: Nice summary of the upcoming contents of the paper.
- Li 161: "Independent of these three unknowns" is vague. Please change to "is known" or something similar.

**Melt Parameterisation Design and Validation: The Three-Equation Melt Parameterisation and Transfer Coefficients**

Li 145 - 146: Correct me if I am wrong here, but the problem is not the three-equation melt parameterisation, it is the inability to resolve the fluxes all the way up to the viscous sublayer, where then heat and salt diffuse at the molecular rate. The wording of these two sentences makes it seem like there is an inherent problem with the three equations that are: 1. Freezing point at ice base, 2. Conservation of heat, 3. Conservation of salt. Please clarify this.

Li 186 – 187: Correct me if I'm wrong here, but I think the take home message from Schmidt et al. (2023) was that while maybe missing the physics, the unstratified shear-driven parameterisation performed closest to the observations under steep slopes.

Li 194: Did Davis & Nicholls (2019) explicitly state that the ice base of Larsen C was smooth? I would expect there to be scallops or ripples from the strong turbulence there.

Li 196: See Table 1 of Washam et al. (2023) for a summary of observed C\_D beneath ice shelves, including what was observed in the Ross Ice Shelf crevasse.

**Stratification Feedback on Turbulence – Insights from Large Eddy Simulations:**

Li 221: Love the specification on Beta/alpha

**Stratification Feedback Parameterisation Design:**

Li 246: This is a small point, but does \Gamma\_{T,S} refer to the combined Gamma for the two-equation parameterisation or the individual Gammas for the three-equation parameterisation? I have not seen anything about a two-equation formulation at this point in the text.

Li 260: Davis et al. (2023) and Schmidt et al. (2023) also published using a heat conductive flux term. I would check through all the values from the published papers before downplaying the importance of heat conduction.

Li 281 - 284: This is not correct – In a stratified setting,  $S_b < S_M$  and therefore  $T_b > T_M$ . So, by choosing a fully mixed setting in (9), you are artificially raising the thermal driving. This can be quite significant in stratified settings – take a look at the Washam et al. (2020) and Schmidt et al. (2023) estimates of  $T_b$  and  $S_b$ .

Li 276 – 296: I realize that the authors attempted to clarify this paragraph to make it more digestible, but I am still having a hard time sifting through the unclear presentation of what has been done here. I realize that in the actual work, they are solving the 3 equations, but here it is a 2 equation formulation with no salt gradient through the boundary layer. Please rewrite this section of remove it, as it is quite confusing.

**Comparison to Observations:**

Li 298 – 300: Consistent with the above comment, I do not understand if StratFeedback here is considering salt flux or a 2 equation formulation, as I interpreted the above section to state. This

is so important for the reader to understand, since this paper is all about boundary layer stratification.

Li 298 - 321: This is a nice section that could be in the results after clarifying the above comment. Although, I understand that it is still motivation for the approach that will then be applied to the model.

**Limiting to a Velocity-Independent Parameterisation:**

**Model Configurations**

**ISOMIP+ Setup and Modifications**

Li 407: Can you please provide a sentence that defends your selection of the standard value for C D.

**Idealised MOM6 Configuration**

**Idealised MITgcm Configuration**

**Idealised Explicit Tidal Forcing:**

**Pine Island Glacier Configuration:**

Li 460: Perhaps it would be worthwhile to mention here that subglacial discharge could interact with the ice-ocean boundary layer in ways to alter the StratFeddback parameterization in locations of the ice shelf.

**Results:**

**Idealised ISOMIP+ Results:**

**Sensitivity to the Low-Velocity Limit:**

**Energetic Ice Shelf Cavity Regimes:**

**Realistic Pine Island Glacier Simulation:**

Li 589: Please note the weakness of satellite-derived melt rates here. While they provide excellent coverage, they are only a first order estimate of what the true melt rate is (See Vankova & Nicholls, 2022 Fig. 8 for comparison with ApRES obs).

Li 587 – 642: This section is much improved!

**Discussion:**

Li 669: Was Davis et al. (2023) in the diffusive convective regime? I think it is worth double checking and also taking a look at Davis et al. (2025): "Lateral Fluxes Drive Basal Melting Beneath

Thwaites Eastern Ice Shelf, West Antarctica." This paper should be cited somewhere in the manuscript.

**Figures:**

There is a labeling convention switch for velocity units from m/s to ms^-1 in the figures. Please change them to be consistent throughout. Also, make m/yr or m yr-1 consistent throughout the manuscript. I suggest to make it consistent with the notation from the text.

- Fig. 1d: Totally optional, but it might be worthwhile to plot the Washam et al. (2023) means from Table 1 in here as grey lines, as well.
- Fig. 5: there is a space missing in the m s^-1 labels on this figure
- Fig. 6: there is a space missing in the m s^-1 labels on this figure
- Fig. 7: there is a space missing in the m s^-1 labels on this figure

**References:**

Davis et al. (2025): "Lateral Fluxes Drive Basal Melting Beneath Thwaites Eastern Ice Shelf, West Antarctica."

---

## Author Response (AR2)

**Response to reviewer comments for the manuscript Stratified suppression of turbulence in an ice shelf basal melt parameterisation**

Second Round of Review EGUsphere-2024-3513 C.K. Yung, M.G. Rosevear, A.K. Morrison, A.McC. Hogg & Y. Nakayama July 23, 2025

We thank the reviewers, Peter Washam and Carolyn Begeman, for their supportive comments on our manuscript, as well as their thorough reading of our revised manuscript. We have endeavoured to address their minor and technical comments in our second revised manuscript. We have also addressed some minor errors found during the preparation of the revised manuscript, reported at the end of this response.

Below we respond to each comment in turn (with our responses indicated in blue) and changes that we have made to the manuscript. Edits to the text are written in purple, and line numbers refer to the revised manuscript.

**Reviewer 1**

**Overall Statement:**

The team of authors have submitted a much-improved version of this bear of a manuscript that tackles the difficult, but important problem of including stratification in ice shelf-ocean models. Overall, the manuscript is quite good, with improved clarity and readability throughout. It is a big, chunky manuscript with a lot of detail that will inspire further work. Apart from some parts of the text needing improved writing structure and some improvement of the Methods Section, I feel that this manuscript will only need a minor review to be ready for publication. I have provided an assortment of comments below that are trivial and should not take much time to address. Great job turning around a solidly-revised manuscript!

-Peter Washam

We thank Peter Washam for your encouraging feedback, we appreciate your thorough reading of our manuscript.

Larger Comments:

Some poorly written sections that can benefit from: https://writersdiet.com/writing-test/ and https://www.amazon.com/Elements-Style-Fourth-William-Strunk/dp/020530902X .

Thank you for sharing these references. We have edited the text for improved sentence structure and hope the revised manuscript is improved.

Part of Methods section is still unclear – see comments below.

We have addressed these concerns below.

I don't think it is appropriate to cite manuscripts in review.

The Cryosphere policy accepts articles in review when available as preprints with a doi. All four of the articles that we have cited and are still in review satisfy this criteria. However, given our reliance on the Zinck et al. (in review) Pine Island Glacier melt rate data we agree that it would be ideal if the Zinck et al. (in review) preprint was accepted and published before the

present manuscript. We have been in contact with the authors of this manuscript and expect that their paper will be published before the present paper, and have confirmed that their melt rate product has not changed in the review process.

Specific Revisions:

Abstract:

Li 2-15: This now fits within the 250 word limit of The Cryosphere. Still, it is quite a long abstract, so consider shortening it somewhat.

The abstract is now shortened to 190 words.

Li 4-5: This sentence is hard to follow. Please rewrite improve readability

**Rewritten and shortened as**

L3-4: Basal melting is controlled by small-scale processes, therefore ice shelf-ocean models rely on parameterisations to predict basal melt.

Li 7: rewrite to "stratification by accumulation of buoyant meltwater beneath a flat ice interface"

Done, but rephrased to "flat and weakly sloped ice interfaces.

Introduction:

Li 19 - 21: Insert somewhere in here that ice shelves have already entered the ocean and displaced sea level.

L15-17: Antarctic ice shelves are the floating extensions of the Antarctic Ice Sheet, and therefore have already displaced sea level. However, they buttress the ice sheet and slow its flow towards the ocean. Ice shelves melt...

Li 25 - 26: add "in the ocean models that produce melt rate projections (IPCC, 2023; Bennetts et al., 2024)."

Done.

Li 28: As someone who does observations, I appreciate it when modelers define a relative scale for models to place them into context. Can you express what scale this "large-scale ocean, climate and earth system models" are operating on? Is this referring to global or circum-Antarctic?

It could be a variety of scales, so we have replaced with 'regional and global':

L24-25: Antarctic ice shelf melting is controlled by ice shelf-ocean boundary layer processes, which occur on scales that are too small to resolve in regional and global ocean, climate and earth system models (Rosevear et al., 2025).

Li 52: You have not defined the sections of the ice-ocean boundary layer yet, so change this to "the far-field flow below the ice-ocean boundary layer"

Sorry, we are unsure what is being referred to, since in both the tracked changes and updated manuscript Line 52 is after the paragraph which defines the ice-ocean boundary layer.

Li 57 - 60: This is a quite verbose sentence with poor structure. Please rewrite to improve readability and be sure to mention that meltwater accumulation and stratification inhibits melting on ice that is flat or has low slopes. This then will lead into the next sentence.

We have rewritten it as

L54-57: In these warmer conditions, and beneath flat and weakly sloping ice shelves, the ice shelf-ocean boundary layer is stratified by buoyant meltwater. The stratification suppresses turbulence and creates a feedback on heat and salt transport, but this feedback is not captured by a constant transfer coefficient (Vreugdenhil and Taylor, 2019; Rosevear et al., 2022b).

Li 62: The Washam et al., (2023) reference is fine if you'd like to cite it, but I think the Schimdt et al., (2023) reference is more appropriate.

Added Schmidt et al. (2023).

Li 67 - 68: This is an incomplete sentence.

If we understand correctly, you are referring to this sentence (parentheses removed): "Other simulations use varying choices of basal melt parameterisations." It is complete, though short.

Li 73: "Vertical discretisation of the basal melt parameterisation" is a little heavy on the jargon. Please rewrite this to make it more clear what is trying to be communicated.

**Rewritten as**

L71-73: The biases could also be related to choices made in the vertical discretisation of the basal melt parameterisation (Gwyther et al., 2020), such as the sampling distance of the far-field flow conditions. However, it is difficult to determine sources of biases with a lack of observations.

Li 80: Not sure about citing a paper in review unless it will be out before this one.

See response on page 1.

Li 81: Specify what you mean by "future ice shelf regime changes."

Replaced with "future cold-warm ice shelf regime shifts"

Li 86 – 92: This sentence is incredibly hard to understand. I understand that the authors are trying to summarize the important missing processes in models, but please rewrite this with improved sentence structure to improve readability.

**We have rewritten it in several sentences as**

L83-90: For instance, studies have used idealised simulations and laboratory setups to explore melt-induced convective plumes (Gayen et al., 2016; Mondal et al., 2019; Zhao et al., 2024; Anselin et al., 2024; Kerr and McConnochie, 2015; McConnochie and Kerr, 2018). Idealised studies have also demonstrated the possibility of double-diffusive convection (Rosevear et al., 2021; Middleton et al., 2021), including the feedback of double-diffusive layers on vertical ice shape (Wilson et al., 2023; Sweetman et al., 2024; Guo and Yang, 2025). Other Large Eddy Simulation studies demonstrate the effect of stratification of melting (Vreugdenhil and Taylor, 2019; Rosevear et al., 2022b; Begeman et al., 2022). The effect of vertical resolution on boundary layer structure in turbulence-permitting ice-ocean melt simulations has also been studied (Patmore et al., 2023; Burchard et al., 2022).

Li 99: State the scale of "large-scale." Is this regional scale (1 large ice shelf or several small ones) or circum-Antarctic?

**Rewritten as**

L97-99: However, thus far, these parameterisations have not been implemented nor tested in realistic ocean models. In this work, we aim to bridge this gap between the insights created by

idealised process studies, and the large-scale regional and global ocean models used in climate and sea level projections.

Li 100: The previous statements on scale should help the reader better understand the goal of the paper now.

See above.

Li 106 – 109: I think these sentences also refer to Schmidt et al., (2023).

Added alongside the existing references.

Li 112: What is "large-scale" here? Is this a regional model or a GCM? I would be amazed if a GCM had this sort of resolution.

There is a large range in possible resolutions, hence the "greater than" statement, but we have rewritten it as

L111-113: However, regional ocean models generally have horizontal grid sizes greater than  $\mathcal{O}(10^3)$  m (and global models are even coarser) and vertical resolutions  $\mathcal{O}(10^1)$  m and cannot resolve ice base variability at the required scales, nor do commonly-used bathymetry and ice base forcing products (Morlighem et al., 2020).

Li 119: Start this sentence with: "In this paper,"

Done.

Li 134 – 142: Nice summary of the upcoming contents of the paper.

Thank you.

Li 161: "Independent of these three unknowns" is vague. Please change to "is known" or something similar.

Technically when we make the transfer coefficient a function of  $L^+$  and therefore the buoyancy forcing (and hence  $T_b$  and  $S_b$ ), the system is no longer quadratic even though the transfer coefficient is known. It is only a quadratic system if the transfer coefficient is independent of m,  $T_b$  and  $S_b$ . Likewise, the Holland and Jenkins (1999) parameterisation with McPhee (1981) stability term does not reduce to a quadratic equation since it includes a dependence on L, the Obukhov length, which depends on the buoyancy forcing. We have rewritten it as

L159-160: Assuming the transfer coefficient is constant or only depends on known values, this system of equations reduces to a quadratic equation.

Melt Parameterisation Design and Validation:

The Three-Equation Melt Parameterisation and Transfer Coefficients

Li 145 – 146: Correct me if I am wrong here, but the problem is not the three-equation melt parameterisation, it is the inability to resolve the fluxes all the way up to the viscous sublayer, where then heat and salt diffuse at the molecular rate. The wording of these two sentences makes it seem like there is an inherent problem with the three equations that are: 1. Freezing point at ice base, 2. Conservation of heat, 3. Conservation of salt. Please clarify this.

You are correct. We have rewritten it as

L142-144: Ice shelf cavity-scale ocean models cannot resolve the turbulent fluxes within the ice shelf-ocean boundary layer. To address this issue, models generally employ the three-equation basal melt parameterisation (Hellmer and Olbers, 1989; Holland and Jenkins, 1999).

Li 186 - 187: Correct me if I'm wrong here, but I think the take home message from Schmidt et al. (2023) was that while maybe missing the physics, the unstratified shear-driven parameterisation performed closest to the observations under steep slopes.

We have revised the sentence in question so that the citations support the sentence:

L185-186: Significant basal slopes, such as those observed by Schmidt et al. (2023) and Washam et al. (2023), are also expected to contribute to deviations from the shear-driven J10 and HJ99-M81 parameterisations (McConnochie and Kerr, 2017, 2018).

Li 194: Did Davis & Nicholls (2019) explicitly state that the ice base of Larsen C was smooth? I would expect there to be scallops or ripples from the strong turbulence there.

Yes, Davis and Nicholls (2019) explicitly state that the underside is smooth. They obtain their drag coefficient from a velocity profile and state that "the ice shelf base at this location appears to be smooth over wide area." However they also predict based on the derived roughness length that "this suggests that the roughness elements (such as scallops) at the base of Larsen C ice shelf have a vertical extent of 1.3 cm."

Li 196: See Table 1 of Washam et al. (2023) for a summary of observed  $C_D$  beneath ice shelves, including what was observed in the Ross Ice Shelf crevasse

Note that Washam et al. (2023) is cited a few sentences down, with values from the Washam et al. (2023) Supplementary Material Table S1 used in Fig. 2. As requested, we have rewritten the sentence to include the Washam et al. (2023) Table 1 range, which encompasses the Lawrence et al. (2023) value.

L191-194: Most suggested values range from 0.0015 (Holland and Jenkins, 1999) to 0.0097 (Jenkins et al., 2010), with a value of 0.0022 estimated from turbulence measurements beneath the smooth underside of Larsen C Ice Shelf (Davis and Nicholls, 2019) and values between 0.0023 and 0.0068 estimated from basal ice morphology beneath the crevassed Ross Ice Shelf grounding zone (Lawrence et al., 2023; Washam et al., 2023).

Stratification Feedback on Turbulence – Insights from Large Eddy Simulations:

Li 221: Love the specification on Beta/alpha

Thanks.

Stratification Feedback Parameterisation Design:

Li 246: This is a small point, but does  $\Gamma_{T,S}$  refer to the combined Gamma for the two-equation parameterisation or the individual Gammas for the three-equation parameterisation? I have not seen anything about a two-equation formulation at this point in the text.

They refer to individual gammas, we have rewritten them separately as " $\Gamma_T$  and  $\Gamma_S$ " to make it clearer.

Li 260: Davis et al. (2023) and Schmidt et al. (2023) also published using a heat conductive flux term. I would check through all the values from the published papers before downplaying the importance of heat conduction.

We have added these references. Note that Schmidt et al. (2023)'s 12% best fit for ice heat conduction to ocean heat transport ratio is not dissimilar to the Holland and Jenkins (1999) value, and we have relaxed the strength of our statement by adding "order 10%". We want to emphasise that we are not saying that heat conduction doesn't matter, rather we are saying that its inclusion (which would be done through a somewhat simplified advection-diffusion model)

would not qualitatively change the conclusions of our paper. To clarify this intent, we have revised some statements:

L255-260: Note that we also neglect the conductive heat flux term of Eqn. 2; although the conductive heat flux may be an important term in some ice shelf cavity conditions (Holland and Jenkins, 1999; Arzeno et al., 2014; Washam et al., 2020; Schmidt et al., 2023; Washam et al., 2023; Wiskandt and Jourdain, in review), melt rates are not expected to decrease by more than order 10% (Holland and Jenkins, 1999). Thus, we do not expect qualitatively different conclusions in the comparison of transfer coefficient parameterisations when we omit the conductive heat flux term.

Appendix A3, L821-822: We also neglect the heat capacity term with the ice conduction, as we have done with the shear-driven parameterisation (Section 2.3), as it is unlikely to qualitatively change the results when comparing melt parameterisations.

Li 281 – 284: This is not correct – In a stratified setting,  $S_b < S_M$  and therefore  $T_b > T_M$ . So, by choosing a fully mixed setting in (9), you are artificially raising the thermal driving. This can be quite significant in stratified settings – take a look at the Washam et al. (2020) and Schmidt et al. (2023) estimates of  $T_b$  and  $S_b$ .

Thanks for pointing this out, this was a mistake made in revisions. As stated earlier in the paragraph,  $T_M - T_{fr}(S_M)$  quantifies the maximum heat available for melting and is likely *larger* than the real ocean heat transport. We have fixed this and added the references too.

L281-284: Note this thermal driving may be larger than the actual temperature difference delivering heat from the ocean for melting that we computed using the three-equation parameterisation ( $T_M - T_b$  in Eqn. 2), as observed in stratified conditions (Schmidt et al., 2023; Washam et al., 2023). However, this thermal driving definition is independent of transfer coefficient parameterisations and therefore more appropriate when comparing parameterisations.

Li 276 – 296: I realize that the authors attempted to clarify this paragraph to make it more digestible, but I am still having a hard time sifting through the unclear presentation of what has been done here. I realize that in the actual work, they are solving the 3 equations, but here it is a 2 equation formulation with no salt gradient through the boundary layer. Please rewrite this section of remove it, as it is quite confusing.

Here we are explaining the definition of the y-axis in Fig. 2 (and similar plots elsewhere). All melt rates are calculated with the three-equation parameterisation. Our parameter space explores both  $T_M$  and  $u_*$ , but we elected to plot the corresponding thermal driving  $T_*$  for each  $T_M$  as a more useful measure of heat above the freezing point. We could not use the true temperature above the freezing point because  $T_M - T_{fr}(S_b)$  depends on  $S_b$  and therefore the transfer coefficient choice, which would not be a fair comparison across parameterisations on the same plot. We prefer to keep the plot as is, but have clarified what was done.

L275-284: We vary the far-field temperature  $T_M$  and friction velocity,  $u_*$ , and compute melt rates across this parameter space with the StratFeedback, ConstCoeff, J10 and HJ99-M81 transfer coefficients, assuming  $S_M = 34.5$  psu and a pressure  $p_b$  of 500 dbar ( $\sim 500$  m depth). Rather than plot the far-field temperature on the y-axis, we instead plot the corresponding thermal driving,

$$T^* = T_M - T_{fr}(S_M) ,$$

which quantifies the maximum heat available for melting (where  $T_{fr}(S) = \lambda_1 S + \lambda_2 + \lambda_3 p_b$  is the local freezing point as in Eqn. 1). Note this thermal driving may be larger than the actual temperature difference delivering heat from the ocean for melting that we computed

using the three-equation parameterisation ( $T_M - T_b$  in Eqn. 2), as observed in stratified conditions (Schmidt et al., 2023; Washam et al., 2023). However, this thermal driving definition is independent of transfer coefficient parameterisations and therefore more appropriate when comparing parameterisations.

Comparison to Observations:

Li 298 – 300: Consistent with the above comment, I do not understand if StratFeedback here is considering salt flux or a 2 equation formulation, as I interpreted the above section to state. This is so important for the reader to understand, since this paper is all about boundary layer stratification.

We use the same plotting convention as before, so solve the three equation parameterisation but plot the corresponding  $T^*$  given  $T_M$  and  $S_M$ . This is consistent with Rosevear et al. (2022b). We believe the rewritten paragraph (included in previous point response) clarifies this. We have also added "three-equation parameterisation" to this section and added an explicit link to the  $T^*$  definition in Eqn. 9 in the Figure caption.

L299-300: Following Rosevear et al. (2022a), we compare the melt rate produced by the Strat-Feedback and ConstCoeff three-equation melt parameterisations...

Fig3 caption: Thermal driving  $T^*$  (Eqn. 9) – friction velocity regime (b) updated from Rosevear et al. (2022b)

Li 298 – 321: This is a nice section that could be in the results after clarifying the above comment. Although, I understand that it is still motivation for the approach that will then be applied to the model.

We received a similar comment from the previous round of reviews, but feel that given the large amount of model experiments discussed in the Results, it is simpler to leave it here as a motivation.

Model Configurations

ISOMIP+ Setup and Modifications

Li 407: Can you please provide a sentence that defends your selection of the standard value for  $C_D$ .

We choose it to follow the ISOMIP+ protocol and enable easier cross-study comparison.

L410-412: In all ISOMIP+ simulations, the drag coefficient  $C_d = 0.0025$  is used for the melt parameterisation and top and bottom boundary conditions for momentum, consistent with Asay-Davis et al. (2016) and Gwyther et al. (2020).

Pine Island Glacier Configuration:

Li 460: Perhaps it would be worthwhile to mention here that subglacial discharge could interact with the ice-ocean boundary layer in ways to alter the StratFeddback parameterization in locations of the ice shelf.

**Added:**

L465-466: However, subglacial discharge could modify the ice shelf-ocean boundary layer, thereby altering the effect of the StratFeedback parameterisation.

Results:

Realistic Pine Island Glacier Simulation:

Li 589: Please note the weakness of satellite-derived melt rates here. While they provide excellent coverage, they are only a first order estimate of what the true melt rate is (See Vankova & Nicholls, 2022 Fig. 8 for comparison with ApRES obs).

L596-698: To assess the parameterisation in a realistic situation where circulation is more complex and the results can be compared with observations, we use the MITgcm Pine Island Glacier setup of Nakayama et al. (2021) (model details in Section 3.2). We tune the drag coefficient to achieve melt rates similar to the Adusumilli et al. (2020) satellite melt rate product, though we acknowledge that satellite melt rates contain uncertainties and can differ from *in situ* ApRES measurements Vaňková and Nicholls (2022); Lindbäck et al. (2025).

Li 587 – 642: This section is much improved!

**Thanks!**

**Discussion:**

Li 669: Was Davis et al. (2023) in the diffusive convective regime? I think it is worth double checking and also taking a look at Davis et al. (2025): "Lateral Fluxes Drive Basal Melting Beneath Thwaites Eastern Ice Shelf, West Antarctica." This paper should be cited somewhere in the manuscript.

Davis et al. (2023) results are in the diffusive regime according to our definitions in Fig. 3, but in their paper they classify the observed conditions as being within the stratified turbulent regime.

We have removed the Davis et al. (2023) reference in the sentence in question and replaced it with a reference to Fig. 3.

L675-677: The StratFeedback parameterisation, though designed for the stratified regime, suppresses melt rates in the diffusive-convective regime and better matches observations and simulations in this regime (Fig. 3, Begeman et al., 2018), but is still an extrapolation in these low-velocity conditions.

Earlier, we also mention the difference between the Davis et al. (2023) classification and ours

L309-311: The original studies may also classify their ice shelf regimes differently, for example, Davis et al. (2023) categorise their observed Thwaites Ice Shelf conditions as stratified turbulence, whereas our definitions place it in the diffusive-convective regime (Fig. 3).

We have also added Davis et al. (2025).

L725-726: Davis et al. (2025) also recently demonstrated the importance of lateral processes beneath warm-cavity ice shelves, which ice shelf basal melt parameterisations do not include.

**Figures:**

There is a labeling convention switch for velocity units from m/s to  $ms^-1$  in the figures. Please change them to be consistent throughout. Also, make m/yr or m yr-1 consistent throughout the manuscript. I suggest to make it consistent with the notation from the text.

We have made the units consistently  $m s^{-1}$  for velocity and m/yr for melt rate throughout the manuscript.

Fig. 1d: Totally optional, but it might be worthwhile to plot the Washam et al. (2023) means from Table 1 in here as grey lines, as well.

We have added these lines.

Fig. 5: there is a space missing in the m  $s^{-1}$  labels on this figure

Fixed.

Fig. 6: there is a space missing in the m  $s^{-1}$  labels on this figure

Fixed

Fig. 7: there is a space missing in the m  $s^{-1}$  labels on this figure

Fixed.

References:

Davis et al. (2025): "Lateral Fluxes Drive Basal Melting Beneath Thwaites Eastern Ice Shelf, West Antarctica."

**Reviewer 2**

**Summary**

Thank you for your thorough response to my review. Overall, I think the changes you made help the clarity of the manuscript, I just noted below a few places where the changes you made seem to me to make the manuscript less clear. I consider all of my suggested changes minor, and I do not have a need to re-review.

We thank Carolyn Begeman for your feedback. We appreciate your thorough reading of our manuscript.

Major comments:

L107: "It is important to highlight..." This paragraph feels like too large a detour from the flow of the introduction. I know you added it on request from reviewers. I think just deleting the sentence beginning "Therefore, although" could help the flow, and the content of that sentence doesn't seem crucial to me.

We have deleted the "therefore, although" sentence but kept the rest of the paragraph to satisfy feedback from other reviewers.

L693: "despite the different drag coefficients" I would delete this phrase. Since the drag coefficient is spatially uniform, I wouldn't expect it to lead to a different anomaly pattern.

**Deleted as suggested.**

L697: "align better with observational products near the grounding line" which predict melt rates near the grounding line of what magnitude? Is the maximum melt rate simulated by HJ99 less than the maximum observed value? I really like Figure D1. I think you may want to add a sentence to the main text indicating that the upper tail of the distribution is increased for your StratFeedback parameterization, bringing it in closer agreement with Zinck et al.

L633-637: Both the tuned StratFeedback and StratFeedback+MK18 experiments have a larger area of the ice shelf with melting greater than 50 m/yr compared with the tuned HJ99-neutral simulation (Fig. D1), and align better with the order 100 m/yr melt rates seen in high-resolution observational products near the grounding line (Zinck et al., in review; Shean et al., 2019). This improvement is demonstrated by the upper tail of the melt rate statistical distribution (Fig. D1) increasing in area with the StratFeedback parameterisation, bringing the distribution closer to the Zinck et al. (in review) product.

L699: "less aggressive tuning" Since only one coefficient is tuned in both and both are in the observational range, I would say the tuning is not more aggressive in one versus the other. I would say the same for L775 "required less drag coefficient tuning" I don't think there's anything special about the HJ99 value that means a parameterization that is closer to that value is better than one that is farther from it.

**We have removed these sentences.**

L748: "the strat feedback parameterization affected melt rates..." I found this sentence confusing. When I look at Figure 6, it seems that the values for the cold, most energetic cavity are in the range observed in Figure 3's WGZ. I think it could be clearer to talk about what regime space you could not access even with a wide range of parameter combinations.

Thank you for your suggested reframing of this statement. Indeed, the tide experiments can access the parameter space of several of the ice shelf borehole observations. We have rewritten this sentence as

L669-673: However, even with explicit tides, our ISOMIP+ experiments could not achieve the thermal driving and friction velocity conditions observed at George VI and Pine Island Glacier ice shelves, nor Stewart (2018)'s summer Ross Ice Shelf observations (compare Figs. 3 and 6). This result suggests that idealised ocean models should be used with caution when assessing melt parameterisations or other ice shelf boundary layer physics, or indeed other aspects of ice shelf cavity circulation.

L784: "may not simulate true ice shelf melt regimes" this seems overly general to me.

**Removed this statement. The sentence now reads**

L696-699: Ocean models, particularly coarse-resolution models, may lack the small-scale flow variability observed at high frequencies beneath ice shelves, either through not resolving these scales of motion (through both horizontal and vertical resolution), not simulating tidal motion, or having anomalously smooth bathymetry and ice base shape.

L795: "exercise caution around the simulated velocities" Unclear what this means. Can you put it in practical terms like what kind of inferences we would make? I don't think you've demonstrated that u\* or L+ in your realistic simulation is clearly biased.

**We have removed this part of the sentence.**

L809: "are extremely sensitive to these ... parameters" Have you shown this? I thought that you only tested one prescribed tidal velocity and one minimum friction velocity? Or do you mean in comparison with one another?

We did some preliminary tests varying the strength of these parameters. However, since we are not showing them, we have rewritten as

L720: However, both melting and circulation are likely to be sensitive to these unconstrained parameters.

**Technical comments:**

L46: Do you mean "Multiple physical processes in the ice shelf—ocean boundary layer contribute to melting beneath ice shelves"?

**Yes, rewritten as suggested.**

L305: I think there is a duplicated "is"

**Addressed.**

L334: "might" >> "may"

Done.

L395: To me, it is unclear which entry in Table 2 corresponds to "a fixed transfer coefficient parameterisation choice"

We have rewritten this sentence to clarify our intent, which was to signpost what is coming (comparison of low-velocity limits with each of the ConstCoeff and StratFeedback parameterisation). We also updated Table 2 to clarify what ConstCoeff is.

L381-382: In this study, we assess the sensitivity to the choice of low-velocity limits with the transfer coefficient parameterisation choices, ConstCoeff and StratFeedback.

L645: "masked tuning melt rate" >> "the masked area over which melt rates were tuned"?

Done.

L649 and 655: "tuning drag coefficient" >> "tuned drag coefficient"

Done.

**Additional modifications to manuscript**

In addition to the modifications made in response to the reviewers, during the review process we identified two errors that we have since addressed.

- 1. Firstly, Figure 7 has been updated to reflect the correct averaging time period (last 180 days of simulations), with minor changes to the melt rates and stated percentages.
- 2. Secondly, we have made the borehole observation temperatures and salinities consistent. The original studies quote their temperature and salinity using a variety of thermodynamic quantities. We choose to use conservative temperature and absolute salinity to match the most recent studies Begeman et al. (2018), Rosevear et al. (2022a) and Davis et al. (2023), and therefore convert potential temperatures to conservative temperature and practical salinity to absolute salinity. This is a choice made for consistency and so that quoted units are correct. Fig 3 has minor changes as a result, as do the numbers in Table B1, and extra information is provided in Appendix B to explain how thermodynamic quantities were converted. In the main text in Sect 2.4 we have added

L304-309: Note that the studies that originally presented this data may have used slightly different melt parameterisations in their comparisons (e.g. Jenkins et al., 2010; Davis and Nicholls, 2019, where different drag coefficients and transfer coefficients were used) and recall we ignore heat conduction into the ice. Additionally, the studies may use different thermodynamic variables – here we use conservative temperature and absolute salinity with conversions performed using the Gibbs Seawater Oceanographic Toolbox (McDougall and Barker, 2011, Appendix B).

**In Appendix B we added**

L853-858: Where data was not reported as conservative temperature or absolute salinity, we have converted the values using the Gibbs Seawater Oceanographic Toolbox (McDougall and Barker, 2011). This choice was made for consistency to match the most recent studies presented (Begeman et al., 2018; Rosevear et al., 2022a; Davis et al., 2023), noting the variety and evolution in thermodynamic variables used previously. However, the choice does not significantly impact results. The thermodynamic variables used in the observational comparison differ from

- those used in the models, potential temperature and practical salinity, but we use the same linear freezing point equation of state coefficients throughout the study (Table 1).
- 3. We have also edited the text, rearranging some sentences for improved structure and readability as requested by Reviewer 1. These changes are mostly in the Introduction. There were also a few typos addressed throughout the manuscript, and some recent, relevant references added: Guo and Yang (2025), Lindbäck et al. (2025), Couston et al. (2021) and Yung et al. (in review). Please see the tracked changes for further details.
- 4. Shortly after resubmission of the second round of review on 20 July 2025, an error was found in two of the simulation experiments. The Pine Island Glacier MITgcm simulations with StratFeedback+MK18 parameterisations originally used a set of old parameters in the melt parameterisation. Specifically, these two experiments were run with slightly different constants  $A_T$ ,  $n_T$ ,  $A_S$  and  $n_S$  (see Table 1 in the manuscript) than what is quoted in the manuscript and used in other experiments. All other experiments were correct.

We re-ran the experiments and there were only very minor changes to the plots as a result (barely noticeable in the plots, and small changes to total melt rates on the order 0.1%). The plots and open model code/data on zenodo have been updated to reflect the correct simulations.

**References**

- Adusumilli, S., Fricker, H. A., Medley, B., Padman, L., and Siegfried, M. R.: Interannual variations in meltwater input to the Southern Ocean from Antarctic ice shelves, Nature Geoscience, 13, 616–620, https://doi.org/10.1038/s41561-020-0616-z, 2020.
- Anselin, J., Holland, P., Jenkins, A., and Taylor, J.: Ice base slope effects on the turbulent ice shelf-ocean boundary current, Journal of Physical Oceanography, pp. 1545–1562, https://doi.org/10.1175/JPO-D-23-0256.1, 2024.
- Arzeno, I. B., Beardsley, R. C., Limeburner, R., Owens, B., Padman, L., Springer, S. R., Stewart, C. L., and Williams, M. J.: Ocean variability contributing to basal melt rate near the ice front of Ross Ice Shelf, Antarctica, Journal of Geophysical Research: Oceans, 119, 4214–4233, https://doi.org/10.1002/2014JC009792, 2014.
- Asay-Davis, X. S., Cornford, S. L., Durand, G., Galton-Fenzi, B. K., Gladstone, R. M., Gudmundsson, G. H., Hattermann, T., Holland, D. M., Holland, D., Holland, P. R., et al.: Experimental design for three interrelated marine ice sheet and ocean model intercomparison projects: MISMIP v. 3 (MISMIP+), ISOMIP v. 2 (ISOMIP+) and MISOMIP v. 1 (MISOMIP1), Geoscientific Model Development, 9, 2471–2497, https://doi.org/10.5194/gmd-9-2471-2016, 2016.
- Begeman, C. B., Tulaczyk, S. M., Marsh, O. J., Mikucki, J. A., Stanton, T. P., Hodson, T. O., Siegfried, M. R., Powell, R. D., Christianson, K., and King, M. A.: Ocean stratification and low melt rates at the Ross Ice Shelf grounding zone, Journal of Geophysical Research: Oceans, 123, 7438–7452, https://doi.org/10.1029/2018JC013987, 2018.
- Begeman, C. B., Asay-Davis, X., and Van Roekel, L.: Ice-shelf ocean boundary layer dynamics from large-eddy simulations, The Cryosphere, 16, 277–295, https://doi.org/10.5194/tc-16-277-2022, 2022.
- Burchard, H., Bolding, K., Jenkins, A., Losch, M., Reinert, M., and Umlauf, L.: The vertical

- structure and entrainment of subglacial melt water plumes, Journal of Advances in Modeling Earth Systems, 14, e2021MS002925, https://doi.org/10.1029/2021MS002925, 2022.
- Couston, L.-A., Hester, E., Favier, B., Taylor, J. R., Holland, P. R., and Jenkins, A.: Topography generation by melting and freezing in a turbulent shear flow, Journal of Fluid Mechanics, 911, A44, https://doi.org/10.1017/jfm.2020.1064, 2021.
- Davis, P. E. and Nicholls, K. W.: Turbulence observations beneath Larsen C ice shelf, Antarctica, Journal of Geophysical Research: Oceans, 124, 5529–5550, https://doi.org/10.1029/2019JC015164, 2019.
- Davis, P. E., Nicholls, K. W., Holland, D. M., Schmidt, B. E., Washam, P., Riverman, K. L., Arthern, R. J., Vaňková, I., Eayrs, C., Smith, J. A., et al.: Suppressed basal melting in the eastern Thwaites Glacier grounding zone, Nature, 614, 479–485, https://doi.org/10.1038/s41586-022-05586-0, 2023.
- Davis, P. E., Nicholls, K. W., Holland, D. M., Schmidt, B. E., Washam, P., Castro, B. F., Riverman, K. L., Smith, J. A., Anker, P. G., Mullen, A. D., et al.: Lateral Fluxes Drive Basal Melting Beneath Thwaites Eastern Ice Shelf, West Antarctica, Geophysical Research Letters, 52, e2024GL111873, https://doi.org/10.1029/2024GL111873, 2025.
- Gayen, B., Griffiths, R. W., and Kerr, R. C.: Simulation of convection at a vertical ice face dissolving into saline water, Journal of Fluid Mechanics, 798, 284–298, https://doi.org/10.1017/jfm.2016.315, 2016.
- Guo, R. and Yang, Y.: The effects of double diffusive convection on the basal melting of solid ice in seawater, Journal of Fluid Mechanics, 1013, A24, https://doi.org/10.1017/jfm.2025.10256, 2025.
- Gwyther, D. E., Kusahara, K., Asay-Davis, X. S., Dinniman, M. S., and Galton-Fenzi, B. K.: Vertical processes and resolution impact ice shelf basal melting: A multi-model study, Ocean Modelling, 147, 101 569, https://doi.org/10.1016/j.ocemod.2020.101569, 2020.
- Hellmer, H. H. and Olbers, D. J.: A two-dimensional model for the thermohaline circulation under an ice shelf, Antarctic Science, 1, 325–336, https://doi.org/10.1017/S0954102089000490, 1989.
- Holland, D. M. and Jenkins, A.: Modeling thermodynamic ice—ocean interactions at the base of an ice shelf, Journal of Physical Oceanography, 29, 1787–1800, https://doi.org/10.1175/1520-0485(1999)029\langle1787:MTIOIA\rangle2.0.CO;2, 1999.
- Jenkins, A., Nicholls, K. W., and Corr, H. F.: Observation and parameterization of ablation at the base of Ronne Ice Shelf, Antarctica, Journal of Physical Oceanography, 40, 2298–2312, https://doi.org/10.1175/2010JPO4317.1, 2010.
- Kerr, R. C. and McConnochie, C. D.: Dissolution of a vertical solid surface by turbulent compositional convection, Journal of Fluid Mechanics, 765, 211–228, https://doi.org/10.1017/jfm.2014.722, 2015.
- Lawrence, J., Washam, P., Stevens, C., Hulbe, C., Horgan, H., Dunbar, G., Calkin, T., Stewart, C., Robinson, N., Mullen, A., et al.: Crevasse refreezing and signatures of retreat observed at Kamb Ice Stream grounding zone, Nature Geoscience, 16, 238–243, https://doi.org//10.1038/s41561-023-01129-y, 2023.
- Lindbäck, K., Darelius, E., Moholdt, G., Vaňková, I., Hattermann, T., Lauber, J., and de Steur, L.: Basal melting and oceanic observations beneath central Fimbulisen, East Antarctica,

- Journal of Geophysical Research: Oceans, 130, e2023JC020506, https://doi.org/10.1029/2023JC020506, 2025.
- McConnochie, C. and Kerr, R.: Testing a common ice-ocean parameterization with laboratory experiments, Journal of Geophysical Research: Oceans, 122, 5905–5915, https://doi.org/10.1002/2017JC012918, 2017.
- McConnochie, C. D. and Kerr, R. C.: Dissolution of a sloping solid surface by turbulent compositional convection, Journal of Fluid Mechanics, 846, 563–577, https://doi.org/10.1017/jfm.2018.282, 2018.
- McDougall, T. J. and Barker, P. M.: Getting started with TEOS-10 and the Gibbs Seawater (GSW) oceanographic toolbox, Scor/iapso WG, 127, 1–28, http://www.teos-10.org/, 2011.
- McPhee, M. G.: An analytic similarity theory for the planetary boundary layer stabilized by surface buoyancy, Boundary-Layer Meteorology, 21, 325–339, https://doi.org/10.1007/BF00119277, 1981.
- Middleton, L., Vreugdenhil, C. A., Holland, P. R., and Taylor, J. R.: Numerical simulations of melt-driven double-diffusive fluxes in a turbulent boundary layer beneath an ice shelf, Journal of Physical Oceanography, 51, 403–418, https://doi.org/10.1175/JPO-D-20-0114.1, 2021.
- Mondal, M., Gayen, B., Griffiths, R. W., and Kerr, R. C.: Ablation of sloping ice faces into polar seawater, Journal of Fluid Mechanics, 863, 545–571, https://doi.org/10.1017/jfm.2018.970, 2019.
- Morlighem, M., Rignot, E., Binder, T., Blankenship, D., Drews, R., Eagles, G., Eisen, O., Ferraccioli, F., Forsberg, R., Fretwell, P., et al.: Deep glacial troughs and stabilizing ridges unveiled beneath the margins of the Antarctic ice sheet, Nature Geoscience, 13, 132–137, https://doi.org/10.1038/s41561-019-0510-8, 2020.
- Nakayama, Y., Cai, C., and Seroussi, H.: Impact of subglacial freshwater discharge on Pine Island Ice Shelf, Geophysical Research Letters, 48, e2021GL093923, https://doi.org/10.1029/2021GL093923, 2021.
- Patmore, R. D., Holland, P. R., Vreugdenhil, C. A., Jenkins, A., and Taylor, J. R.: Turbulence in the ice shelf-ocean boundary current and its sensitivity to model resolution, Journal of Physical Oceanography, 53, 613–633, https://doi.org/10.1175/JPO-D-22-0034.1, 2023.
- Rosevear, M. G., Gayen, B., and Galton-Fenzi, B. K.: The role of double-diffusive convection in basal melting of Antarctic ice shelves, Proceedings of the National Academy of Sciences, 118, https://doi.org/10.1073/pnas.2007541118, 2021.
- Rosevear, M. G., Galton-Fenzi, B., and Stevens, C.: Evaluation of basal melting parameterisations using in situ ocean and melting observations from the Amery Ice Shelf, East Antarctica, Ocean Science, 18, 1109–1130, https://doi.org/0.5194/os-18-1109-2022, 2022a.
- Rosevear, M. G., Gayen, B., and Galton-Fenzi, B. K.: Regimes and transitions in the basal melting of Antarctic ice shelves, Journal of Physical Oceanography, 52, 2589–2608, https://doi.org/10.1175/JPO-D-21-0317.1, 2022b.
- Schmidt, B. E., Washam, P., Davis, P. E., Nicholls, K. W., Holland, D. M., Lawrence, J. D., Riverman, K. L., Smith, J. A., Spears, A., Dichek, D., et al.: Heterogeneous melting near the Thwaites Glacier grounding line, Nature, 614, 471–478, https://doi.org/10.1038/s41586-022-05691-0, 2023.

- Shean, D. E., Joughin, I. R., Dutrieux, P., Smith, B. E., and Berthier, E.: Ice shelf basal melt rates from a high-resolution digital elevation model (DEM) record for Pine Island Glacier, Antarctica, The Cryosphere, 13, 2633–2656, https://doi.org/10.5194/tc-13-2633-2019, 2019.
- Stewart, C. L.: Ice-ocean interactions beneath the north-western Ross Ice Shelf, Antarctica, Ph.D. thesis, https://doi.org/10.17863/CAM.21483, 2018.
- Sweetman, J. K., Shakespeare, C. J., Stewart, K. D., and McConnochie, C. D.: Laboratory experiments of melting ice in warm salt-stratified environments, Journal of Fluid Mechanics, 984, A42, https://doi.org/10.1017/jfm.2024.201, 2024.
- Vaňková, I. and Nicholls, K. W.: Ocean variability beneath the Filchner-Ronne ice shelf inferred from basal melt rate time series, Journal of Geophysical Research: Oceans, 127, e2022JC018879, https://doi.org/10.1029/2022JC018879, 2022.
- Vreugdenhil, C. A. and Taylor, J. R.: Stratification effects in the turbulent boundary layer beneath a melting ice shelf: Insights from resolved large-eddy simulations, Journal of Physical Oceanography, 49, 1905–1925, https://doi.org/10.1175/JPO-D-18-0252.1, 2019.
- Washam, P., Nicholls, K. W., Münchow, A., and Padman, L.: Tidal modulation of buoyant flow and basal melt beneath Petermann Gletscher Ice Shelf, Greenland, Journal of Geophysical Research: Oceans, 125, e2020JC016427, https://doi.org/10.1029/2020JC016427, 2020.
- Washam, P., Lawrence, J. D., Stevens, C. L., Hulbe, C. L., Horgan, H. J., Robinson, N. J., Stewart, C. L., Spears, A., Quartini, E., Hurwitz, B., et al.: Direct observations of melting, freezing, and ocean circulation in an ice shelf basal crevasse, Science Advances, 9, eadi7638, https://doi.org/10.1126/sciadv.adi7638, 2023.
- Wilson, N. J., Vreugdenhil, C. A., Gayen, B., and Hester, E. W.: Double-Diffusive Layer and Meltwater Plume Effects on Ice Face Scalloping in Phase-Change Simulations, Geophysical Research Letters, 50, e2023GL104396, https://doi.org/10.1029/2023GL104396, 2023.
- Wiskandt, J. and Jourdain, N.: Brief Communication: Representation of heat conduction into the ice in marine ice shelf melt modeling, EGUsphere, 2024, 1–10, https://doi.org/10.5194/egusphere-2024-2239, in review.
- Yung, C. K., Asay-Davis, X. S., Adcroft, A., Bull, C. Y. S., De Rydt, J., Dinniman, M. S., Galton-Fenzi, B. K., Goldberg, D., Gwyther, D. E., Hallberg, R., Harrison, M., Hattermann, T., Holland, D. M., Holland, D., Holland, P. R., Jordan, J. R., Jourdain, N. C., Kusahara, K., Marques, G., Mathiot, P., Menemenlis, D., Morrison, A. K., Nakayama, Y., Sergienko, O., Smith, R. S., Stern, A., Timmermann, R., and Zhou, Q.: Results of the second Ice Shelf Ocean Model Intercomparison Project (ISOMIP+), EGUsphere, 2025, 1–50, https://doi.org/10.5194/egusphere-2025-1942, in review.
- Zhao, K. X., Skyllingstad, E. D., and Nash, J. D.: Improved parameterizations of vertical iceocean boundary layers and melt rates, Geophysical Research Letters, 51, e2023GL105862, https://doi.org/10.1029/2023GL105862, 2024.
- Zinck, A.-S. P., Lhermitte, S., Wearing, M., and Wouters, B.: Exposure to Underestimated Channelized Melt in Antarctic Ice Shelves, https://doi.org/10.21203/rs.3.rs-4806463/v1, in review.